# Learning Hamiltonian Flow Maps:
# Mean Flow Consistency for Large-Timestep Molecular Dynamics

**Winfried Ripken** [* 1 2]  **Michael Plainer** [* 3 4 1 2]  **Gregor Lied** [* 1]  **J. Thorben Frank** [* 1 2]
**Oliver T. Unke** [5]  **Stefan Chmiela** [1 2]  **Frank Noé** [3 6 7]  **Klaus-Robert Müller** [1 2 5 8 9]

## Abstract

Simulating the long-time evolution of Hamiltonian systems is limited by the small timesteps required for stable numerical integration. To overcome this constraint, we introduce a framework to learn *Hamiltonian Flow Maps* by predicting the *mean* phase-space evolution over a chosen time span $\Delta t$, enabling stable large-timestep updates far beyond the stability limits of classical integrators. To this end, we impose a *Mean Flow* consistency condition for time-averaged Hamiltonian dynamics. Unlike prior approaches, this allows training on independent phase-space samples without access to future states, avoiding expensive trajectory generation. Validated across diverse Hamiltonian systems, our method in particular improves upon molecular dynamics simulations using machine-learned force fields (MLFF). Our models maintain comparable training and inference cost, but support significantly larger integration timesteps while trained directly on widely-available *trajectory-free* MLFF datasets. Our code, model weights, and self-contained JAX and PyTorch notebooks are available at `https://ml4molsim.github.io/hamiltonian-flow-maps`.

## 1. Introduction

Solving Hamilton's equations of motion is central to modeling physical systems, with molecular dynamics (MD) simulations as a prominent application, where atomic motion is governed by interatomic forces (Hollingsworth & Dror, 2018; Goldstein et al., 2001). In practice, numerical integration of these equations requires small timesteps $\Delta t$ to ensure stability, rendering long-time simulations computationally expensive (Hairer et al., 2006). To reduce cost, forces are commonly computed using fast but approximate force fields (Frenkel & Smit, 2002) as a replacement for highly accurate but computationally prohibitive quantum mechanical (QM) methods (Tuckerman & Martyna, 2000). While machine-learned force fields (MLFFs) have started to bridge the gap between QM accuracy and computational cost (Noé et al., 2020; Keith et al., 2021; Unke et al., 2021b; Bonneau et al., 2026), the integration bottleneck remains, limiting the accessible time scales (Wang et al., 2025).

As a result, accelerating MLFFs is an active area of research, focusing on efficient architectures (Xie et al., 2023; Frank et al., 2024) and improved implementations (Pelaez et al., 2024; Park et al., 2024). An orthogonal line of work bypasses integration by predicting future states directly (Bigi et al., 2025a; Thiemann et al., 2026; Thompson et al., 2026; Diez et al., 2026). However, these approaches rely on reference trajectories for training, which are prohibitively expensive to generate with *ab-initio* QM methods. To mitigate this, Bigi et al. (2025a) propose training an MLFF on *ab-initio* data and then using it as a teacher to generate trajectories for distillation. While effective, this strategy remains costly for chemically diverse datasets, introduces teacher biases, and requires retraining whenever the teacher changes.

This work proposes an alternative approach: Rather than regressing future states from reference trajectories, we learn the cumulative dynamics directly from a single phase-space configuration and its instantaneous time derivative. Since Hamilton's equations are deterministic, this information is sufficient to learn the correct *Hamiltonian flow map*. We reframe learning accurate large-timestep dynamics as optimizing a consistency condition: Inspired by the seminal work of *Flow Maps* for generative modeling (Boffi et al., 2025b), we adapt the mathematical formalism of *Mean Flows* (Geng et al., 2025a), originally designed for stochastic generative modeling, to the deterministic integration of

---

[*]Equal contribution [1]Technical University Berlin [2]BIFOLD Berlin [3]Free University of Berlin [4]Zuse School ELIZA [5]Google DeepMind [6]Rice University [7]Microsoft Research AI4Science [8]MPI for Informatics, Saarbrücken [9]Department of Artificial Intelligence, Korea University. Correspondence to: Thorben Frank <thorben.frank@tu-berlin.de>, Frank Noé <franknoe@microsoft.com>, Klaus-Robert Müller <klaus-robert.mueller@tu-berlin.de>.

*Proceedings of the $43^{rd}$ International Conference on Machine Learning*, Seoul, South Korea. PMLR 306, 2026. Copyright 2026 by the author(s).

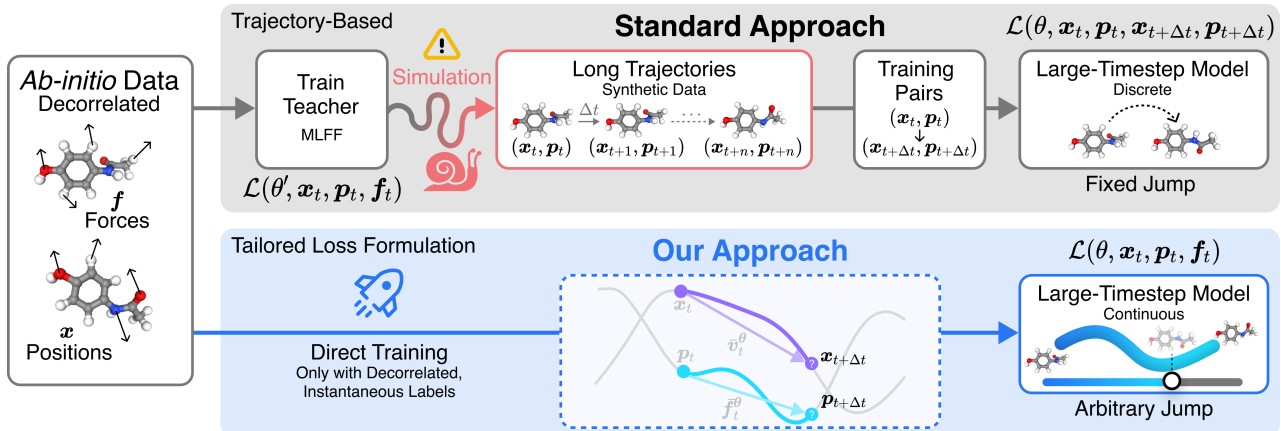

*Figure 1.* Hamiltonian Flow Maps (HFMs) for large timesteps in phase space. **Top row**: Existing approaches rely on trajectory data, typically generated using a teacher MLFF through sequential simulation with small timesteps. While this enables training large-timestep models via direct regression, the resulting models are limited to a fixed set of predefined timesteps that cannot be adjusted after training. **Bottom row**: Our approach learns continuous-time, large-timestep dynamics directly from decorrelated *ab-initio* samples, without requiring trajectories. The model supports arbitrary timesteps at inference. Our tailored loss combines force matching with a consistency constraint that enforces agreement of the predicted flow across different time horizons (see Section 4.2).

Hamilton's equations. This objective requires *neither numerical integration nor time-ordered data*. We can therefore train directly on standard MLFF datasets, which typically consist of decorrelated molecular geometries with force labels (Eastman et al., 2023; Ganscha et al., 2025; Levine et al., 2025), without relying on a teacher MLFF or trajectory generation. The same objective recovers instantaneous forces as well as their time-averaged counterparts, yielding a single model that functions both as a conventional force field and as a large-timestep integrator. Consequently, our models rely purely on *ab-initio* data (see Figure 1) and can be trained with a compute budget comparable to standard MLFFs (Batzner et al., 2022).

Our contributions can be summarized as follows:

- We derive a trajectory-free training objective to learn large-timestep Hamiltonian flow maps from single-time phase-space samples via a consistency condition.
- We show that our method can be trained on widely available MLFF datasets, which typically contain independent snapshots rather than equilibrium samples.
- We apply our method to molecular dynamics, enabling stable rollouts at timesteps far beyond the stability limits of standard MLFFs while maintaining high accuracy.

## 2. Background

### 2.1. Hamiltonian Mechanics

Hamiltonian mechanics (Goldstein et al., 2001) provide a unifying description for a wide range of physical systems, including harmonic oscillators, rigid-body dynamics, gravitational $N$-body systems, and molecular systems governed by interatomic potentials. In all cases, the system state evolves in phase space according to an energy function, the *Hamiltonian* $\mathcal{H}$, where the time evolution is governed by

$$\boldsymbol{v} := \dot{\boldsymbol{x}} = \frac{\partial \mathcal{H}(\boldsymbol{x}, \boldsymbol{p})}{\partial \boldsymbol{p}}, \qquad \boldsymbol{f} := \dot{\boldsymbol{p}} = -\frac{\partial \mathcal{H}(\boldsymbol{x}, \boldsymbol{p})}{\partial \boldsymbol{x}}, \quad (1)$$

with $\boldsymbol{x}$ and $\boldsymbol{p}$ denoting positions and momenta, and $\boldsymbol{v}$ and $\boldsymbol{f}$ the corresponding instantaneous velocities and forces. These dynamics are fully coupled, meaning that changes in positions affect the evolution of momenta and vice versa.

The state at a later time $t^*$ can be obtained by integrating these instantaneous quantities over a time interval $[t, t^*]$,

$$\begin{pmatrix} \boldsymbol{x}_{t^*} \\ \boldsymbol{p}_{t^*} \end{pmatrix} = \begin{pmatrix} \boldsymbol{x}_t \\ \boldsymbol{p}_t \end{pmatrix} + \int_t^{t^*} \begin{pmatrix} \boldsymbol{v}_\tau \\ \boldsymbol{f}_\tau \end{pmatrix} \mathrm{d}\tau, \quad (2)$$

with $\boldsymbol{v}_\tau$ and $\boldsymbol{f}_\tau$ denote the instantaneous velocity and force along the trajectory at time $\tau$. Equation (2) is an exact identity, where classical numerical integrators approximate this integral by evaluating $\boldsymbol{v}_\tau$ and $\boldsymbol{f}_\tau$ at discrete timesteps.

### 2.2. Molecular Dynamics

Molecular dynamics (MD) are a concrete instantiation of Hamiltonian mechanics in which the Hamiltonian $\mathcal{H}$ is assumed to be separable and takes the form

$$\mathcal{H}(\boldsymbol{x}, \boldsymbol{p}) = T(\boldsymbol{p}) + V(\boldsymbol{x}), \quad (3)$$

with kinetic energy $T(\boldsymbol{p}) = \sum_{i=1}^N \frac{\|\boldsymbol{p}^{(i)}\|_2^2}{2\boldsymbol{m}^{(i)}}$, where $\boldsymbol{p}^{(i)}$ and $\boldsymbol{m}^{(i)}$ denote the momentum and mass of particle $i$, and the potential energy $V(\boldsymbol{x})$ depends only on the positions $\boldsymbol{x}$. Under this assumption, the equations of motion reduce to

$$\boldsymbol{v} := \frac{\partial T(\boldsymbol{p})}{\partial \boldsymbol{p}} = \frac{\boldsymbol{p}}{\boldsymbol{m}}, \qquad \boldsymbol{f} := -\frac{\partial V(\boldsymbol{x})}{\partial \boldsymbol{x}}. \quad (4)$$

Conventional MD simulations model particle interactions using an empirical *force field* that parameterizes the potential energy $V(\boldsymbol{x})$, and integrate the resulting equations of motion with explicit numerical schemes, such as Velocity Verlet, advancing the system through many small timesteps of size $\Delta t$ (Frenkel & Smit, 2002).

## 3. Related Work

**Designing faster MLFFs.** MLFFs have emerged as a powerful tool to obtain cheap but QM-accurate MD simulations (Behler & Parrinello, 2007; Chmiela et al., 2017; Schütt et al., 2018; Batzner et al., 2022). Although much faster than QM methods, simulations with MLFFs remain limited by computational cost, restricting sampling and applications to large systems (Wang et al., 2025). Recent work has focused on improving MLFF efficiency via architectural advances (Xie et al., 2023; Frank et al., 2024), kernel optimization (Pelaez et al., 2024), or GPU parallelization (Musaelian et al., 2023; Park et al., 2024). In contrast, our approach accelerates simulations by increasing the integration timestep instead of speeding up models.

**Learning larger timesteps from trajectories.** A complementary line of work accelerates simulations by learning from trajectory data to enable larger integration steps (Xu et al., 2024; Jing et al., 2024), typically in two regimes.

The first operates in the *deterministic* setting, where the current state uniquely determines the future state. Similar to our approach, these methods aim to predict long integration steps for Hamiltonian dynamics (Bigi et al., 2025a; Thiemann et al., 2026), but they require reference trajectories, whose generation at *ab-initio* accuracy is prohibitively expensive (Thompson et al., 2026). Using pre-trained MLFFs surrogates (Bigi et al., 2025a) alleviates this cost but remains computationally expensive and can introduce teacher-induced biases. In contrast, our method trains directly on *ab-initio* force data without trajectories or surrogate teachers. Moreover, while prior approaches are usually trained for a fixed timestep $\Delta t$, our model can be evaluated for any $\Delta t \in [0, \Delta t_{\max}]$, unifying instantaneous predictions and large-timestep propagation within a single model.

The second regime goes beyond the deterministic limit and reframes long-time propagation as a *stochastic sampling problem* (Noé et al., 2019; Lewis et al., 2025), typically addressed with generative models (Klein et al., 2023; Schreiner et al., 2023; Diez et al., 2026; Olsson, 2026). As the timestep increases, systems increasingly behave like an equilibrium sampler, shifting emphasis from temporal evolution to matching stationary statistics. These methods target this equilibrium-sampling limit, but sacrifice fine-grained kinetic information and typically restrict dynamics to a fixed thermodynamic ensemble.

**Interpolation for Hamiltonian dynamics.** Other methods exploit time-invertibility to interpolate between states in phase space and recover ground truth dynamics with high accuracy (Winkler et al., 2022; Wang et al., 2023).

**Similarities with few-step generative models.** Our problem shares similarities with accelerated sampling in diffusion models (Sohl-Dickstein et al., 2015; Ho et al., 2020; Song et al., 2021, see also Appendix D), where recent works employ few-step generation to bypass expensive ODE integration (Song et al., 2023; Song & Dhariwal, 2024; Lu & Song, 2025). In contrast to those probabilistic methods, our training objective is entirely deterministic and purely motivated by physics without any noise injection.

Another key distinction lies in the training signal: Flow-based generative models interpolate between known source and target distributions over a fixed time interval $t \in [0, 1]$. In physical simulations without trajectory data, neither the target samples nor the path are known, precluding trajectory-matching objectives (Salimans & Ho, 2022; Zheng et al., 2023). Instead, we train solely on instantaneous forces over an unbounded time horizon, adapting Mean Flow-style formulations (Geng et al., 2025a; 2026; Boffi et al., 2025a; Sabour et al., 2025) to Hamiltonian dynamics.

**Similarities with PINNs.** Our objective can be viewed as minimizing the residual of the Liouville equation. From this perspective it resembles a PDE loss found in Physics-Informed Neural Networks (PINNs) (Raissi et al., 2019).

In contrast to PINNs, our method uses self-distillation to learn an operator capable of advancing any phase-space state. Consequently, unlike standard PINNs that fit specific solutions to fixed boundary conditions, our framework dynamically bootstraps its own targets and trains exclusively on instantaneous labels.

## 4. Method

In this work, we present a framework that accelerates Hamiltonian dynamics by shifting the focus from *faster force evaluation* to *faster integration*. In the following, we introduce a trajectory-free training objective for a model capable of recovering instantaneous forces while supporting accurate predictions over large timesteps, reducing integration steps.

### 4.1. Hamiltonian Flow Maps

The core idea is to model the Hamiltonian evolution directly in phase space over a finite time interval. Concretely, we learn a *Hamiltonian flow map* $u_{t \to t^*}$ that advances positions and momenta from time $t$ to $t^*$ according to Equation (2),

$$\begin{pmatrix} \boldsymbol{x}_t \\ \boldsymbol{p}_t \end{pmatrix} \mapsto \begin{pmatrix} \boldsymbol{x}_{t^*} \\ \boldsymbol{p}_{t^*} \end{pmatrix} = u_{t \to t^*}(\boldsymbol{x}_t, \boldsymbol{p}_t). \tag{5}$$

A direct way to learn this mapping is to regress paired initial $(\boldsymbol{x}_t, \boldsymbol{p}_t)$ and final states $(\boldsymbol{x}_{t^*}, \boldsymbol{p}_{t^*})$ from trajectories (Klein et al., 2023; Thiemann et al., 2026; Bigi et al., 2025a), or to unroll a numerical integrator during training (Žugec et al., 2025). However, both strategies are inherently sequential, leading to high computational cost and limited scalability. Instead, we seek a formulation that avoids simulating intermediate states during training and learns large-step predictions directly from *single-time phase-space samples*.

To achieve this, we build on recent advances in few-step generative modeling, in particular the *Mean Flow* framework of Geng et al. (2025a) and the related self-distillation approach of Boffi et al. (2025a), which reduce the number of integration steps required during sampling in flow matching and diffusion models (Lipman et al., 2023). While conceptually related, we do not train a flow matching model. Instead, we adapt the underlying consistency identity to deterministic Hamiltonian phase-space dynamics and to a training regime in which only instantaneous force labels are available. In Appendix D, we discuss this in more detail.

**Mean displacement field.** Motivated by this view, we define the startpoint-conditioned mean displacement field

$$\bar{u}(\boldsymbol{x}_t, \boldsymbol{p}_t, t^* - t) := \frac{1}{t^* - t} \int_t^{t^*} \begin{pmatrix} \boldsymbol{v}_\tau \\ \boldsymbol{f}_\tau \end{pmatrix} d\tau, \qquad (6)$$

which represents the time-averaged velocity and force accumulated along the trajectory over the interval $[t, t^*]$. This quantity captures the non-trivial, integration-dependent component of the flow map in Equation (2) that we seek to approximate with a neural network. Formulating the dynamics in terms of this mean displacement normalizes small and large time intervals to the same scale and ensures a well-defined limit as $\Delta t = t^* - t$ approaches 0, in which $\bar{u}$ recovers the instantaneous velocity and force. In the following, we show how $\bar{u}$ can be learned efficiently and subsequently used to reconstruct the Hamiltonian flow map $u_{t \to t^*}$.

**Trajectory-free consistency equation.** The parameterization of the average displacement field $\bar{u}$ suffers from the same computational drawbacks as the full flow map since it still involves an explicit time integral. Differentiating with respect to time yields an equivalent formulation that no longer involves an explicit integral. To this end, we multiply Equation (6) by $(t^* - t)$ and take the time derivative $d/dt$,

$$\frac{d}{dt}[(t^* - t)\bar{u}(\boldsymbol{x}_t, \boldsymbol{p}_t, t^* - t)] = \frac{d}{dt}\int_t^{t^*} \begin{pmatrix} \boldsymbol{v}_\tau \\ \boldsymbol{f}_\tau \end{pmatrix} d\tau. \quad (7)$$

We use the product rule on the left, the fundamental theorem of calculus on the right, and multiply with $-1$ to get

$$\bar{u}(\boldsymbol{x}_t, \boldsymbol{p}_t, t^*-t) - (t^*-t)\frac{d}{dt}\bar{u}(\boldsymbol{x}_t, \boldsymbol{p}_t, t^*-t) = \begin{pmatrix} \boldsymbol{v}_t \\ \boldsymbol{f}_t \end{pmatrix}. \quad (8)$$

The total derivative $\frac{d}{dt}\bar{u}$ follows from the chain rule,

$$\frac{d}{dt}\bar{u}(\boldsymbol{x}_t, \boldsymbol{p}_t, t^* - t) =$$
$$= \frac{d\boldsymbol{x}_t}{dt}\partial_{\boldsymbol{x}_t}\bar{u} + \frac{d\boldsymbol{p}_t}{dt}\partial_{\boldsymbol{p}_t}\bar{u} + \frac{dt^* - t}{dt}\partial_{t^*-t}\bar{u} \quad (9)$$
$$= (\boldsymbol{v}_t \cdot \partial_{\boldsymbol{x}_t})\bar{u} + (\boldsymbol{f}_t \cdot \partial_{\boldsymbol{p}_t})\bar{u} - \partial_{t^*-t}\bar{u},$$

where we used Hamilton's equations to replace the time derivatives of position $\boldsymbol{x}$ and momentum $\boldsymbol{p}$ with velocity $\boldsymbol{v}$ and force $\boldsymbol{f}$, respectively.

Together, Equations (8) and (9) define a necessary integral-free consistency condition that any mean displacement field must satisfy under Hamiltonian dynamics.

### 4.2. Computational Approach

This consistency condition directly induces a regression objective for learning the displacement field. Concretely, this allows us to train a neural network $\bar{u}^{\boldsymbol{\theta}}$ by minimizing

$$\mathcal{L}(\boldsymbol{\theta}, \boldsymbol{x}, \boldsymbol{p}, \boldsymbol{v}, \boldsymbol{f}) = \mathbb{E}_{\Delta t}\left[\left\|\bar{u}^{\boldsymbol{\theta}}(\boldsymbol{x}, \boldsymbol{p}, \Delta t) - \bar{u}_{\text{tgt}}\right\|_2^2\right]$$

$$\bar{u}_{\text{tgt}} = \begin{pmatrix} \boldsymbol{v} \\ \boldsymbol{f} \end{pmatrix} + \Delta t\left[(\boldsymbol{v} \cdot \partial_{\boldsymbol{x}})\bar{u}^{\boldsymbol{\theta}} + (\boldsymbol{f} \cdot \partial_{\boldsymbol{p}})\bar{u}^{\boldsymbol{\theta}} - \partial_{\Delta t}\bar{u}^{\boldsymbol{\theta}}\right],$$
$$(10)$$

where the time interval is denoted by $\Delta t = t^* - t$. For a system with $N$ particles and $d$ dimensions, the neural network is a map $\bar{u}^{\boldsymbol{\theta}} : \mathbb{R}^{2dN+1} \to \mathbb{R}^{2dN}$, predicting mean forces and velocities $\bar{\boldsymbol{f}}, \bar{\boldsymbol{v}} \in \mathbb{R}^{dN}$ from positions $\boldsymbol{x} \in \mathbb{R}^{dN}$, momenta $\boldsymbol{p} \in \mathbb{R}^{dN}$ and time interval $\Delta t \in [0, \Delta t_{\text{max}}]$.

In Appendix C, we show that minimizing this loss is sufficient to learn Hamiltonian flow maps. Importantly, the objective uses only instantaneous labels and *does not* require time integrals or any explicit simulation during training.

**Training data and sampling.** Our training objective requires phase-space samples $(\boldsymbol{x}_t, \boldsymbol{p}_t, \boldsymbol{v}_t, \boldsymbol{f}_t)$ together with a randomly sampled time interval $\Delta t$. Because we restrict to time-independent Hamiltonians, the phase-space state is fully determined by $(\boldsymbol{x}, \boldsymbol{p})$ and is independent of absolute time $t$. Training can therefore use arbitrary instantaneous tuples $(\boldsymbol{x}, \boldsymbol{p}, \boldsymbol{v}, \boldsymbol{f})$ that do not need to come from trajectories. For the systems we study, we consider the common case of *separable* Hamiltonians, so that $\boldsymbol{v} = \boldsymbol{p}/\boldsymbol{m}$. In molecular

---

**Algorithm 1** Efficiently Learning Hamiltonian Flow Maps

**Require:** Neural network $\bar{u}^{\boldsymbol{\theta}}$, sample $(\boldsymbol{x}, \boldsymbol{p}, \boldsymbol{p}/\boldsymbol{m}, \boldsymbol{f})$, distribution $q(\tau)$, maximal timestep $\Delta t_{\text{max}}$
1: $\Delta t \leftarrow \tau \cdot \Delta t_{\text{max}}, \ \tau \sim q(\tau)$
2: $(\bar{u}, \frac{d}{dt}\bar{u}) \leftarrow \texttt{jvp}(\bar{u}^{\boldsymbol{\theta}}, (\boldsymbol{x}, \boldsymbol{p}, \Delta t), (\boldsymbol{p}/\boldsymbol{m}, \boldsymbol{f}, -1))$
3: $\bar{u}_{\text{tgt}} \leftarrow \texttt{stack}([\boldsymbol{p}/\boldsymbol{m}, \boldsymbol{f}]) + \Delta t \cdot \frac{d}{dt}\bar{u}$
4: $\mathcal{L}_{\boldsymbol{\theta}} \leftarrow \|\bar{u} - \texttt{stopgrad}(\bar{u}_{\text{tgt}})\|_2^2$

---

systems, however, most MLFF datasets provide only $(\boldsymbol{x}, \boldsymbol{f})$ and no $\boldsymbol{p}$ labels. We therefore sample $\boldsymbol{p}$ independently from Maxwell–Boltzmann distributions at varying temperatures, resulting in the training sample $(\boldsymbol{x}, \boldsymbol{p}, \boldsymbol{p}/\boldsymbol{m}, \boldsymbol{f})$. Details on the sampling of $\Delta t$ and $\boldsymbol{p}$ are given in Appendix B.

**Efficient implementation.** The loss in Equation (10) can be evaluated efficiently using Jacobian–vector products (jvp) available in standard deep learning frameworks. It requires only a single forward pass of the network and one backward pass. Optimizing the full objective, including gradients through the target $\bar{u}_{\text{tgt}}$, can yield the best results (You et al., 2026) but incurs substantial computational cost due to higher-order derivatives. In practice, we therefore adopt a *stop-gradient* formulation, which treats the right-hand side as a fixed regression target and avoids these higher-order terms. This strategy is well established (Song et al., 2023; Song & Dhariwal, 2024; Geng et al., 2025a; He et al., 2026) and reduces the training overhead to approximately 30% compared to standard MLFF training on a single GPU. The full training procedure is given in Algorithm 1.

**Force matching and consistency across time.** In the case where $\Delta t = 0$, the force component of the loss reduces to

$$\mathcal{L}(\boldsymbol{\theta}, \boldsymbol{x}, \boldsymbol{p}, \boldsymbol{v}, \boldsymbol{f})_{dN+1:} = \left\| \bar{u}^{\boldsymbol{\theta}}(\boldsymbol{x}, \boldsymbol{p}, \Delta t = 0)_{dN+1:} - \boldsymbol{f} \right\|_2^2, \tag{11}$$

which recovers the standard loss of MLFFs (Unke et al., 2021b) and therefore the network learns to predict the interatomic forces. For $\Delta t > 0$, our loss encourages consistency between the predicted displacement field and the time-integrated instantaneous quantities.

Intuitively, this view amounts to a form of self-distillation across time: instantaneous force predictions act as a base signal, while predictions at larger $\Delta t$ are constrained to match the accumulation of these local dynamics (see bottom row of Figure 1). Zhang et al. (2026) analyze these two components of the *Mean Flow* loss in the generative modeling setting, which is useful for understanding our loss as well. Accurate learning at $\Delta t = 0$ is essential, as it anchors the consistency condition for larger timesteps. Therefore, we sample $\Delta t = 0$ in 75% of training cases, balancing stable learning of instantaneous quantities with gradual enforcement of long-timescale consistency (see Appendix G.3).

**Recovering the Hamiltonian flow map.** The proposed loss learns the *average displacement* Hamiltonian flow $\bar{u}$, which does not directly predict the next state. However, by construction, this field recovers the Hamiltonian flow map $u_{t \to t^*}$ (compare Equation (2)) by multiplying with $\Delta t$

$$u_{t \to t^*}(\boldsymbol{x}_t, \boldsymbol{p}_t) = \begin{pmatrix} \boldsymbol{x}_t \\ \boldsymbol{p}_t \end{pmatrix} + \Delta t \cdot \bar{u}(\boldsymbol{x}_t, \boldsymbol{p}_t, \Delta t). \tag{12}$$

Simulations can be obtained by repeating this update, effectively reducing the number of materialized states.

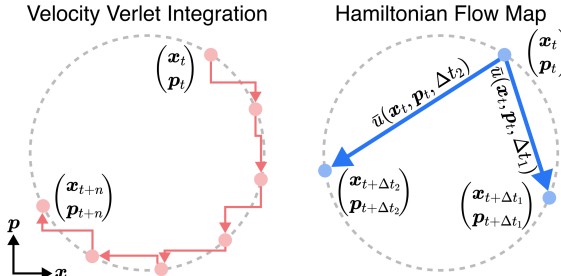

*Figure 2.* HFMs vs. classical integration: Symplectic integrators such as Velocity Verlet (VV) advance the system through many local half-steps (left). HFMs instead predict the phase-space displacement over the interval directly (right). By modeling the mean velocity and force $(\bar{\boldsymbol{v}}, \bar{\boldsymbol{f}})$ over the interval, HFMs apply a single large update (blue arrows) from the current state.

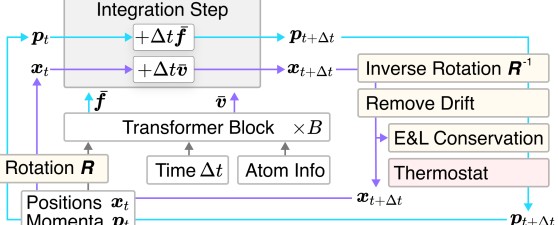

*Figure 3.* Simulation with a trained HFM. The model predicts mean forces $\bar{\boldsymbol{f}}$ and mean velocities $\bar{\boldsymbol{v}}$, conditioned on the current phase-space state $(\boldsymbol{x}_t, \boldsymbol{p}_t)$, the time interval $\Delta t$, and atom information (types and masses). The predictions advance the system by one integration step. To ensure stable rollouts, we refine the updated state using three filters (right): removal of global translation drift, coupled conservation of energy and angular momentum (E&L), and random rotation for approximate rotation equivariance.

**Example: The 1D harmonic oscillator.** We illustrate the proposed method using an analytically tractable toy system: the one-dimensional harmonic oscillator, which models, e.g., a frictionless spring. The trajectories for this system are circular orbits in phase space. Figure 2 schematically compares Velocity Verlet integration with our approach.

In contrast to classical integrators, such as Velocity Verlet, which rely on local half-step corrections, Hamiltonian flow maps directly apply finite displacements in phase space. We define these displacements through mean velocities and mean forces, obtained by learning the averaged instantaneous vector field over a continuous time interval.

### 4.3. Model Architecture and Inference Filters

We adopt a recent transformer for 3D geometries proposed by Frank et al. (2025) to predict mean velocities and mean forces (see Appendix E). Directly predicting these quantities yields non-conservative models with energy drift potentially causing instability (Bigi et al., 2025b), which is critical in NVE simulations (Thiemann et al., 2026; Bigi et al., 2026) and can persist in NVT ensembles despite thermostats (Bigi et al., 2025a). In our experiments, NVT rollouts remain stable even without inference filters, but still show some

thermodynamic drift (see Appendix G.11).

To mitigate these problems, previous work has introduced inference filters to preserve physical constraints (Bigi et al., 2025a; Thiemann et al., 2026). However, conservation of total angular momentum and total energy have so far been looked at only separately. We solve a constrained optimization problem in closed form, to minimally adjust the momenta after each update step, such that both total energy and angular momentum are preserved (see Appendices B.2 and B.3). Compared to sequential or decoupled correction schemes, this coupled formulation crucially avoids interference between the two constraints. To apply this filter, we need to evaluate the change in total energy that is caused by our model to correct it: for the potential energy part, we employ an additional prediction head in our model, or use a separately trained MLFF. Note, that these filters do not correct fundamental integration errors and do not enable larger timesteps on their own (see Appendix G.11).

In summary, we employ three lightweight filters during simulation (see Figure 3):

- **Random rotation.** Reduce effects of non-equivariance by applying a random global rotation before each step.
- **Remove drift.** Remove total momentum of the system to prevent flying-ice-cube instabilities (Harvey et al., 1998).
- **Energy and angular momentum conservation.** Enforce coupled conservation of total energy and angular momentum by modifying and rescaling the momenta.

## 5. Experiments

In this section, we train Hamiltonian Flow Maps (HFMs) with our proposed approach and evaluate them. Our code and model weights are available at:

```
https://ml4molsim.github.io/
hamiltonian-flow-maps.
```

### 5.1. Classical Mechanics

We first consider three systems described by classical mechanics, including two simple single-particle problems and a more complex gravitational 100-body system.

**Single particle systems.** We begin with two classic Hamiltonian systems, the Barbanis potential and the spring pendulum, and stress-test integration stability at large timesteps. For simplicity, we use a dimensionless coordinate system. For each system, we train HFMs on temporally uncorrelated phase-space samples by uniformly sampling positions and momenta at fixed total energy and computing forces from the ground-truth potential (see Appendix F.1 for details).

Once trained, we simulate dynamics either with Velocity Verlet (VV) using the ground-truth potential, or with the learned HFM at increasing timestep $\Delta t$ (Figure 4). Even at

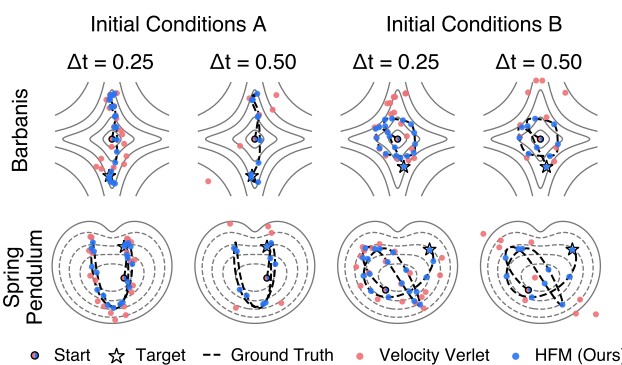

*Figure 4.* Generated trajectories for two potentials (Barbanis, spring pendulum) from two initial conditions (left/right). We compare the ground truth (dashed), VV (red), and HFM (blue) at two large timesteps. VV deviates even with a small timestep increase, while HFM stays aligned at larger steps.

$\Delta t = 0.25$, VV no longer matches the reference trajectory and exhibits phase and amplitude errors as $\Delta t$ increases. In contrast, the learned HFM remains stable for larger $\Delta t$ and closely follows the reference dynamics. This demonstrates that HFMs trained with our objective can take timesteps well beyond the regime in which VV is accurate.

**Gravitational $N$-body system.** To consider a more complex setting, we use the 100-particle gravitational dataset from Brandstetter et al. (2022). Although the dataset contains trajectories, we ignore temporal ordering and train only on individual force labels, sampling momenta independently for each configuration. We again omit units for simplicity.

Following the protocol in Brandstetter et al. (2022), we assess extrapolation over the interval from $t = 3$ to $t^* = 4$ (details in Appendix F.2). We vary the number of integration steps $n$ used to bridge this interval, which sets the step size to $\Delta t = (t^* - t)/n$. We compare HFMs to VV integration of the ground-truth analytical potential in Figure 5. HFMs significantly outperform VV at large step sizes (small $n$), where the classical integrator becomes unstable. As $\Delta t$ decreases, the task approaches instantaneous force prediction, and performance saturates due to inherent model error.

### 5.2. Molecular Dynamics

In the following, we show that our framework enables long-time MD simulations with fewer integration steps while faithfully reproducing the target equilibrium statistics.

We train and analyze HFM models on a selection of the MD17 and MD22 datasets (Chmiela et al., 2017; 2023), alanine dipeptide in implicit solvent (Köhler et al., 2021) and coarse-grained proteins (Lindorff-Larsen et al., 2011). We use experiment-dependent training splits with details in Appendices F.3–F.5. As baselines, we train SO(3)-equivariant MLFFs and use the force outputs for numerical integration

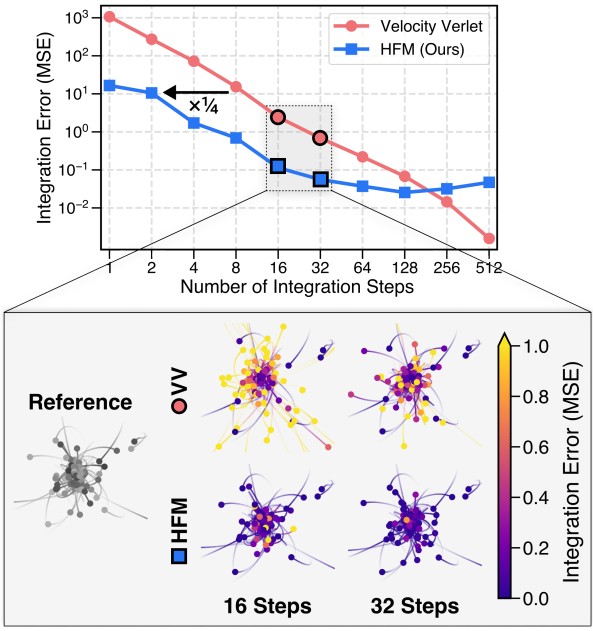

*Figure 5.* Gravitational $N$-body rollout ($t = 3$, $t^* = 4$). **Top:** MSE as a function of the number of integration steps $n$, averaged over the test set. The HFM remains accurate at coarse discretization, while VV diverges rapidly. Overall, the HFM requires $4\times$ fewer steps than VV to reach the same accuracy. **Bottom:** Particle trajectories for 16 and 32 integration steps, colored by per-particle deviation from the ground truth. The HFM yields physically consistent rollouts, whereas VV leads to unphysical scattering.

within VV with $\Delta t = 0.5$ fs. All MLFF models use the same data splits as the HFM models.

**NVE simulations.** Before performing realistic constant temperature NVT simulations, we analyze our learned HFM in the NVE setting. This poses an absolute stress test for physical validity, as thermostats in an NVT can absorb numerical integration errors. Choosing the right timestep $\Delta t$ is a trade-off between faster state space exploration and simulation accuracy, which can be addressed with our learned HFM. To test this, we compare the Ramachandran plots of the dihedral angles in paracetamol for HFM ($\Delta t = 9$ fs) and the VV baseline given a fixed budget of integration steps. We find that the learned HFM explores the conformational space significantly faster, discovering relevant modes and uncorrelated samples more efficiently than the baseline

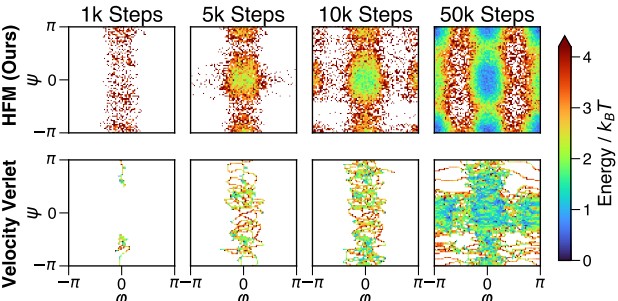

*Figure 6.* State space exploration in an NVE simulation of the paracetamol molecule for an increasing number of integration steps. **Top**: Our HFM with $\Delta t = 9$ fs. **Bottom**: MLFF + VV integrator with $\Delta t = 0.5$ fs. Since our HFM allows for a larger integration timestep, it covers the state space much faster.

(Figure 6). Additional ablations testing energy and angular momentum conservation can be found in Appendix G.1.

**Structural observables.** We then transition to the realistic setting of NVT simulations and perform 300 ps MD simulations for four organic molecules using a Langevin thermostat. To asses the structural accuracy and stability of our simulations, we follow Fu et al. (2023) and calculate the distribution $h(r)$ over interatomic distances $r$ and report the MAE w.r.t. *ab-initio* reference data. Our learned HFM produces accurate simulation statistics for all test molecules (see Table 1). At $\Delta t = 0.5$ fs, it achieves distance distribution accuracies comparable to the MLFF baseline. As expected, the error increases in $\Delta t$, but the HFM model maintains decent accuracy and stability up to $\Delta t = 9$ fs for most molecules. Our results demonstrate robustness close to the training horizon $\Delta t_{\max} = 10$ fs and well beyond standard integration limits. Appendices G.2, G.3 and G.5 contain further metrics for wall clock time comparison, force MAE and simulation stability. We further ran simulations with other thermostats (Appendix G.4), where we observe that a global and deterministic Nosé-Hoover thermostat (Martyna et al., 1996) can lead to simulation artifacts. Similar effects have been observed in other long-timestep models (Bigi et al., 2025a) and have been attributed to failure of the thermostat to maintain kinetic energy equipartition across atom types.

**Temporal observables.** Although distributions are well suited to validate structural statistics, they do not probe temporal dynamics. To do so, we perform an NVT simulation

*Table 1.* Interatomic distances $h(r)$ MAE [unitless] for 300 ps of MD simulation in the NVT ensemble using a Langevin thermostat w.r.t. the reference data from *ab-initio* calculations. For each system, we compare a single HFM model with $\Delta t_{\max} = 10$ fs at increasing timesteps $\Delta t \in \{0.5, 1, 3, 5, 7, 9\}$ fs against an MLFF baseline with $\Delta t = 0.5$ fs. We run the simulation for 5 different initial conditions and report the average performance with standard deviations shown in parentheses. Lower values indicate better performance.

| Dataset | MLFF | Hamiltonian Flow Map | | | | | |
| | 0.5 fs | 0.5 fs | 1 fs | 3 fs | 5 fs | 7 fs | 9 fs |
|---|---|---|---|---|---|---|---|
| Aspirin | 0.030 (0.002) | 0.026 (0.001) | 0.027 (0.001) | 0.035 (0.001) | 0.038 (0.001) | 0.042 (0.001) | 0.046 (0.001) |
| Ethanol | 0.073 (0.001) | 0.075 (0.001) | 0.076 (0.002) | 0.083 (0.001) | 0.093 (0.001) | 0.089 (0.001) | 0.115 (0.001) |
| Naphthalene | 0.043 (0.000) | 0.044 (0.000) | 0.047 (0.000) | 0.053 (0.000) | 0.051 (0.000) | 0.052 (0.000) | 0.063 (0.000) |
| Salicylic Acid | 0.035 (0.000) | 0.035 (0.000) | 0.039 (0.000) | 0.042 (0.000) | 0.051 (0.001) | 0.052 (0.000) | 0.058 (0.003) |

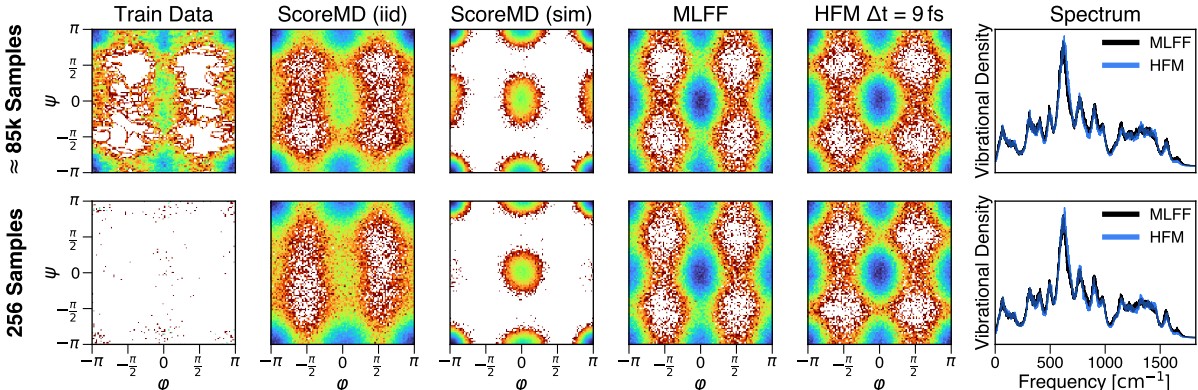

*Figure 7.* Data efficiency for Paracetamol in NVT simulation (3 ns). We compare models trained on $\approx 85$k samples (**top**) and on only 256 samples (**bottom**). We show the free energy surface projected onto the dihedral angles $\varphi, \psi$. Both the MLFF and the HFM recover the reference distribution despite the biased training distribution. In contrast, ScoreMD, like many other generative approaches, assumes converged training samples and therefore reproduces the biased training distribution in both settings. The HFM additionally yields accurate large-timestep dynamics, as indicated by a vibrational spectrum that closely matches the small-timestep MLFF.

of paracetamol and calculate the power spectrum from the atomic velocities (right panel of Figure 7). Close alignment between HFM and the MLFF baseline indicates that our model describes the underlying dynamics with high fidelity.

**Training data distribution.** Next, we train two HFMs on sparse (256 samples) and saturated (85k samples) paracetamol data (bottom and top row in Figure 7). Within NVT simulations, both HFMs accurately describe the conformational space (Ramachandran plots) and the temporal dynamics (Power spectra) compared to the MLFF baseline, demonstrating that our HFMs capture the essential flow topology even from sparse data and potentially biased distributions.

In contrast, many generative approaches assume specific training data, e.g., samples from a converged MD simulation. To highlight this, we also compare our HFM with ScoreMD (Plainer et al., 2025), a recent diffusion-based approach that can generate independent samples (iid) by denoising but can also be used for simulation (sim). On the respective splits, it reproduces the training distribution rather than yielding converged distributions (Figure 7), which in practice requires computationally expensive reweighting (Klein & Noé, 2024). Further, many generative approaches cannot recover dynamics such as spectra.

**Long, stable peptide dynamics.** To demonstrate scalability to systems with complex metastable states, we simulate alanine dipeptide. Parts A and B of Figure 8 show the Ramachandran plot and marginal distributions of the backbone dihedral angles $(\varphi, \psi)$. We generate a total trajectory length of $1\,\mu$s by running 10 parallel simulations of 100 ns each with the HFM at varying timesteps. The learned integrator captures transitions between metastable states and reproduces the characteristic free energy landscape.

**Stepsize.** For alanine dipeptide, we train the HFM with $\Delta t_{\max} = 15$ fs and obtain accurate free energy surfaces

with $\Delta t = 12$ fs. This is practically relevant because standard MD simulations often require additional stabilization beyond $\Delta t = 4$ fs (Hopkins et al., 2015), with $\Delta t \approx 6$ fs being achievable in principle (Izaguirre et al., 1999), which is still only about half of the step size we use here. Such regularization techniques may also further improve stability for our method, but are beyond the scope of this work.

Across experiments and hyperparameter choices, we observe that long rollouts around $\Delta t \approx 10$ fs can become unstable for alanine dipeptide. This instability does not consistently correlate with one-step prediction error, suggesting that it arises from the interaction between the learned map and the simulation setup rather than from degraded local accuracy. We report free energy surfaces and compute errors for a wider range of timesteps in Appendix G.12.

**Inference-time Extrapolation.** We test extrapolation with HFMs beyond incomplete data by training on MD22 Ac-Ala3-NHMe (Chmiela et al., 2023), where a conformational mode is completely missing. The HFM extrapolates during inference to recover the missing mode, producing the same free energy landscape like an MLFF (Figure 9).

**Coarse-grained systems.** To evaluate robustness on larger macromolecules, we study our model on coarse-grained (CG) representations of Chignolin and BBA, retaining only the $C_\alpha$ atoms. Instead of direct ab initio data, we generate force labels by querying ScoreMD (Plainer et al., 2025), a pre-trained coarse-grained machine learning force field for fast folding proteins (Lindorff-Larsen et al., 2011). We distill this model into large-timestep dynamics using our HFM framework. To account for any potential model bias, we additionally train an MLFF for comparison on the same CG data. As shown in C and D of Figure 8, HFM maintains stable simulations and captures the correct ensemble statistics across complex landscapes and large timesteps.

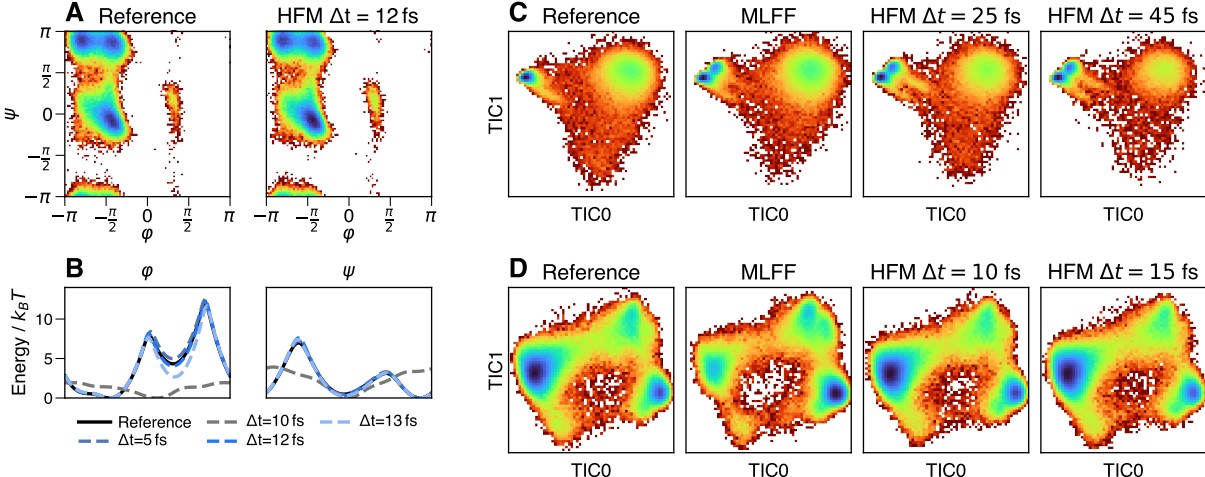

*Figure 8.* Long-running NVT simulations. **A** Ramachandran plot of the backbone dihedral angles $\phi$ and $\psi$ for alanine dipeptide, comparing the reference simulation to the HFM evaluated at $\Delta t = 12$ fs. **B** Free energy projections for alanine dipeptide across different step sizes $\Delta t$ compared to the reference distribution. Across most step sizes, the HFM correctly samples the metastable basins; as $\Delta t$ increases, the systematic deviation grows. **C** Time-lagged independent component analysis (TICA) projections for Chignolin simulations (10 ns, 10 replicas) comparing the all-atom reference simulation, rollouts using a CG MLFF and HFM trained on CG data. The HFM accurately captures the conformational landscape even at large time steps up to $\Delta t = 45$ fs compared to the reference MLFF simulation ($\Delta t = 0.5$ fs). **D** TICA projections for BBA simulations (10 ns, 10 replicas). HFM converges significantly faster with $\Delta t = 10/15$ fs.

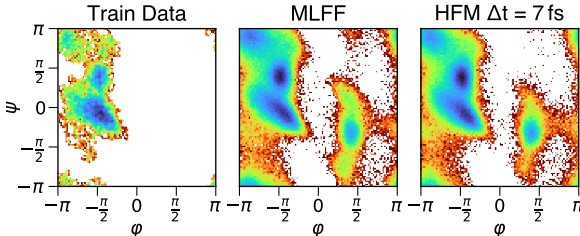

*Figure 9.* Ac-Ala3-NHMe NVT simulations. When trained on biased data where a mode is completely missing (left), our HFM recovers the hidden mode (3ns rollouts, 10 replicas) and stays comparable to an MLFF reference (right).

For Chignolin, all of our models overrepresent a second folded state. However, since this mode is also present in the MLFF, we attribute this to the distillation of ScoreMD, where errors can easily accumulate. Our model remains highly stable and accurate at a step size of $\Delta t = 45$ fs ($\Delta t_{\max} = 50$ fs), exceeding a speedup of $20\times$ compared to the underlying reference force field while recovering correct transition rates (see Appendix G.15).

For BBA, our HFM rollouts stay closer to the reference distribution and better recover the respective weights between states. Overall, we found our HFM model to be more stable during inference (compare Appendix G.16). We show stable rollouts up to $\Delta t = 15$ fs steps ($\Delta t_{\max} = 20$ fs).

## 6. Conclusion and Limitations

A central challenge in computational physics is the need for small integration steps to ensure numerical stability.

This constraint often prevents simulations from reaching the timescales of interest and has motivated data-driven approaches to accelerate time integration. Existing methods, however, rely on expensive pre-computed trajectories.

We address this limitation by introducing a trajectory-free training objective for Hamiltonian Flow Maps (HFMs) from instantaneous supervision (forces and positions). Across systems from single-particle dynamics to multi-body molecular systems, we show that HFMs enable stable large-timestep simulation beyond the limits of classical integrators.

The proposed maps approximate the underlying dynamics and therefore only approximately preserve physical structure such as symplecticity, energy conservation, or equivariance. In practice, we combine the model with inference filters and local stochastic thermostats to improve long-rollout stability. Furthermore, we observe that optimizing the training objective becomes increasingly challenging as the timestep grows, particularly in strongly chaotic regimes.

Overall, we show that useful dynamical maps can be learned without access to trajectories. Our HFMs are trained on the same data as classical MLFFs while maintaining similar training and inference costs. Beyond this, they combine learning instantaneous forces and large-timestep dynamics simultaneously. We hope this perspective broadens the set of practical tools for long-time simulation, encouraging practitioners to augment standard MLFFs with HFMs to obtain stable, large-timestep simulations.

## Impact Statement

This work develops a machine learning framework for accelerating Hamiltonian dynamics simulations, with a focus on molecular dynamics. By enabling larger integration timesteps without requiring trajectory data during training, the proposed method can substantially reduce the computational cost of atomistic simulations. This may lower barriers to studying long-timescale processes in chemistry, materials science, and biophysics, and support applications such as drug design. We do not anticipate direct negative societal impacts; the method is intended to complement, not replace, first-principles simulations, and does not introduce ethical concerns beyond those common in computational modeling in the physical sciences.

## Acknowledgements

We would like to thank Klara Bonneau, Jonas Köhler, Hartmut Maennel, Tim Ebert, Khaled Kahouli and Martin Michajlow for fruitful discussions and their helpful input. JTF, WR, KRM, and SC acknowledge support by the German Federal Ministry of Research, Technology and Space (BMFTR) under Grants BIFOLD24B, BIFOLD25B, 01IS18037A, 01IS18025A, and 01IS24087C. MP is supported by the Konrad Zuse School of Excellence in Learning and Intelligent Systems (ELIZA) through the DAAD programme Konrad Zuse Schools of Excellence in Artificial Intelligence, sponsored by the Federal Ministry of Education and Research. Further, this work was in part supported by the BMBF under Grants 01IS14013A-E, 01GQ1115, 01GQ0850, 01IS18025A, 031L0207D, and 01IS18037A. KRM was partly supported by the Institute of Information & Communications Technology Planning & Evaluation (IITP) grants funded by the Korea government (MSIT) (No.2019-0-00079, Artificial Intelligence Graduate School Program, Korea University and No. 2022-0-00984, Development of Artificial Intelligence Technology for Personalized Plug-and-Play Explanation and Verification of Explanation). Moreover, we gratefully acknowledge support by the Deutsche Forschungsgemeinschaft (SFB1114, Projects No. A04 and No. B08) and the Berlin Mathematics center MATH+ (AA1-10, AA2-20, and AA-Health).

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

## A. Notation

For better readability, we will use $\square_i$ instead of $\square^{(i)}$ to index particles throughout the appendix. For the state $(\boldsymbol{x}_t, \boldsymbol{f}_t, \boldsymbol{p}_t)$ of an $N$-particle system at time $t$, with $\boldsymbol{x}_t, \boldsymbol{f}_t, \boldsymbol{p}_t \in \mathbb{R}^{dN}$, we write $\boldsymbol{x}_{t,i}, \boldsymbol{f}_{t,i}, \boldsymbol{p}_{t,i} \in \mathbb{R}^d$ for the position, force, and momentum of particle $i$, respectively. In contrast, for an independent sample $(\boldsymbol{x}, \boldsymbol{f}, \boldsymbol{p})$, with $\boldsymbol{x}, \boldsymbol{f}, \boldsymbol{p} \in \mathbb{R}^{dN}$, we write $\boldsymbol{x}_i, \boldsymbol{f}_i, \boldsymbol{p}_i \in \mathbb{R}^d$. To avoid ambiguity, we reserve $i, j$ exclusively for particle indices. All other single symbol subscripts denote time labels.

| Description | Symbol | Identities |
|---|---|---|
| Hamiltonian | $\mathcal{H}(\boldsymbol{x}, \boldsymbol{p})$ | |
| Position | $\boldsymbol{x}$ | |
| Momentum | $\boldsymbol{p}$ | $\boldsymbol{p} = m\boldsymbol{v}$ |
| Velocity | $\boldsymbol{v}$ | $\boldsymbol{v} = \dot{\boldsymbol{x}}$ |
| Force | $\boldsymbol{f}$ | $\boldsymbol{f} = m\boldsymbol{a} = \dot{\boldsymbol{p}}$ |
| Mass | $m$ | |
| Timestep | $\Delta t$ | |
| Start point | $t$ | $t^* - t$ |
| End point | $t^*$ | |
| Time derivative | $\dot{\square}$ | $\frac{\mathrm{d}}{\mathrm{d}t}\square$ |

*Table 2.* Notation and symbols used in the paper.

## B. Implementation Details

In this section, we will discuss additional implementation details for our approach.

### B.1. Training Hamiltonian Flow Maps

**Training objective.** We implement the loss in Equation (10) for a sample $(\boldsymbol{x}, \boldsymbol{p}, \boldsymbol{v}, \boldsymbol{f})$ and timestep $\Delta t$ as a weighted sum:

$$\mathcal{L}(\boldsymbol{\theta}, \boldsymbol{x}, \boldsymbol{p}, \boldsymbol{v}, \boldsymbol{f}, \Delta t) = \lambda_v \mathcal{L}_v(\boldsymbol{\theta}, \boldsymbol{x}, \boldsymbol{p}, \boldsymbol{v}, \boldsymbol{f}, \Delta t) + \lambda_f \mathcal{L}_f(\boldsymbol{\theta}, \boldsymbol{x}, \boldsymbol{p}, \boldsymbol{v}, \boldsymbol{f}, \Delta t). \tag{13}$$

The individual loss terms, i.e., the velocity loss term $\mathcal{L}_v(\boldsymbol{\theta}, \boldsymbol{x}, \boldsymbol{p}, \boldsymbol{f}, \boldsymbol{v})$ and force loss term $\mathcal{L}_f(\boldsymbol{\theta}, \boldsymbol{x}, \boldsymbol{p}, \boldsymbol{f}, \boldsymbol{v})$, are defined as

$$\mathcal{L}_v(\boldsymbol{\theta}, \boldsymbol{x}, \boldsymbol{p}, \boldsymbol{f}, \boldsymbol{v}, \Delta t) = w_v \cdot \frac{1}{3N} \sum_{i=1}^{N} m_i \cdot \left\| \bar{\boldsymbol{v}}^{\boldsymbol{\theta}}(\boldsymbol{x}, \boldsymbol{p}, \Delta t)_i - \mathrm{sg}(\bar{\boldsymbol{v}}_{\mathrm{tgt}})_i \right\|_2^2, \tag{14}$$

$$\mathcal{L}_f(\boldsymbol{\theta}, \boldsymbol{x}, \boldsymbol{p}, \boldsymbol{f}, \boldsymbol{v}, \Delta t) = w_f \cdot \frac{1}{3N} \sum_{i=1}^{N} \left\| \bar{\boldsymbol{f}}^{\boldsymbol{\theta}}(\boldsymbol{x}, \boldsymbol{p}, \Delta t)_i - \mathrm{sg}(\bar{\boldsymbol{f}}_{\mathrm{tgt}})_i \right\|_2^2, \tag{15}$$

where $N$ denotes the number of particles in the system, $\bar{\boldsymbol{v}}^{\boldsymbol{\theta}}(\boldsymbol{x}, \boldsymbol{p}, \Delta t)_i, \bar{\boldsymbol{f}}^{\boldsymbol{\theta}}(\boldsymbol{x}, \boldsymbol{p}, \Delta t)_i \in \mathbb{R}^d$ are the predicted mean velocities and forces for the $i$-th particle, $\mathrm{sg}(\bar{\boldsymbol{v}}_{\mathrm{tgt}})_i, \mathrm{sg}(\bar{\boldsymbol{f}}_{\mathrm{tgt}})_i \in \mathbb{R}^d$ are the regression targets, and $\mathrm{sg}(\cdot)$ is a stop-gradient operator (see Section 4). In general, the targets $\bar{\boldsymbol{v}}_{\mathrm{tgt}}, \bar{\boldsymbol{f}}_{\mathrm{tgt}} \in \mathbb{R}^{dN}$ are defined as

$$\bar{\boldsymbol{v}}_{\mathrm{tgt}} = \boldsymbol{v} + \Delta t \left[ (\boldsymbol{v} \cdot \partial_x) \bar{\boldsymbol{v}}^{\boldsymbol{\theta}} + (\boldsymbol{f} \cdot \partial_p) \bar{\boldsymbol{v}}^{\boldsymbol{\theta}} - \partial_{\Delta t} \bar{\boldsymbol{v}}^{\boldsymbol{\theta}} \right], \tag{16}$$

$$\bar{\boldsymbol{f}}_{\mathrm{tgt}} = \boldsymbol{f} + \Delta t \left[ (\boldsymbol{v} \cdot \partial_x) \bar{\boldsymbol{f}}^{\boldsymbol{\theta}} + (\boldsymbol{f} \cdot \partial_p) \bar{\boldsymbol{f}}^{\boldsymbol{\theta}} - \partial_{\Delta t} \bar{\boldsymbol{f}}^{\boldsymbol{\theta}} \right], \tag{17}$$

and can be efficiently computed using Jacobian–vector products (`jvp`) available in `jax` (Bradbury et al., 2018) with minimal computational overhead (see Section 4). The scalars $w_v, w_f \in \mathbb{R}$ in Equations (14) and (15) are adaptive loss weights (Geng et al., 2025b) with

$$w_v = \left( \mathrm{sg} \left( \frac{1}{3N} \sum_{i=1}^{N} m_i \cdot \left\| \bar{\boldsymbol{v}}^{\boldsymbol{\theta}}(\boldsymbol{x}, \boldsymbol{p}, \Delta t)_i - \mathrm{sg}(\bar{\boldsymbol{v}}_{\mathrm{tgt}})_i \right\|_2^2 \right) + c \right)^{-p}, \tag{18}$$

$$w_f = \left( \mathrm{sg} \left( \frac{1}{3N} \sum_{i=1}^{N} \left\| \bar{\boldsymbol{f}}^{\boldsymbol{\theta}}(\boldsymbol{x}, \boldsymbol{p}, \Delta t)_i - \mathrm{sg}(\bar{\boldsymbol{f}}_{\mathrm{tgt}})_i \right\|_2^2 \right) + c \right)^{-p}. \tag{19}$$

In our experiments, we set $c = 10^{-3}$ and $p = 0.5$, yielding a weighting similar to the Pseudo-Huber loss (Song & Dhariwal, 2024), which we find to improve the training stability. Additionally, in Equation (14), we weight the contribution of the $i$-th particle to the velocity loss by its mass $\boldsymbol{m}_i$. We apply mass-weighted loss scaling, rather than predicting mass-scaled quantities (Zheng et al., 2021; Bigi et al., 2025a), as doing so would violate the underlying physics in our loss formulation. The networks $\bar{\boldsymbol{v}}^{\boldsymbol{\theta}}$ and $\bar{\boldsymbol{f}}^{\boldsymbol{\theta}}$ share the same backbone. We discuss the architectures used throughout this work in Appendix E.

**Rotation augmentation.** In this work, we learn Hamiltonian flow maps using network architectures that do not transform equivariantly under rotation of the input positions. As a consequence, we apply random rotations during training to enforce approximate rotational equivariance via data augmentation (Abramson et al., 2024; Frank et al., 2025; Plainer et al., 2025). Specifically, we randomly sample rotation matrices $\boldsymbol{R} \in \mathbb{R}^{3 \times 3}$ (orthogonal matrices with determinant +1) and apply them as

$$\text{ApplyRotation}(\boldsymbol{R}, \boldsymbol{x})_i = \boldsymbol{R}\boldsymbol{x}_i, \quad \text{ApplyRotation}(\boldsymbol{R}, \boldsymbol{f})_i = \boldsymbol{R}\boldsymbol{f}_i, \quad \text{ApplyRotation}(\boldsymbol{R}, \boldsymbol{p})_i = \boldsymbol{R}\boldsymbol{p}_i, \qquad (20)$$

where $\boldsymbol{x}_i, \boldsymbol{f}_i, \boldsymbol{p}_i \in \mathbb{R}^3$ denote the position, force and momentum of the $i$-th particle, respectively. Based on Equation (20), we can adapt the original training algorithm in Algorithm 1 with minimal changes to allow for training with rotation augmentation. The modified training algorithm is summarized in Algorithm 2.

---

**Algorithm 2** Efficiently Learning Hamiltonian Flow Maps with Rotation Augmentation

---

**Require:** Neural network $\bar{u}^{\boldsymbol{\theta}}$, training sample $(\boldsymbol{x}, \boldsymbol{p}, \boldsymbol{p}/\boldsymbol{m}, \boldsymbol{f})$, distribution $q(\tau)$, maximal timestep $\Delta t_{\max}$
1: $\tau \sim q(\tau)$
2: $\boldsymbol{R} \sim \text{SO}(3)$
3: $\Delta t \leftarrow \tau \cdot \Delta t_{\max}$
4: $\boldsymbol{x} \leftarrow \text{ApplyRotation}(\boldsymbol{R}, \boldsymbol{x})$
5: $\boldsymbol{f} \leftarrow \text{ApplyRotation}(\boldsymbol{R}, \boldsymbol{f})$
6: $\boldsymbol{p} \leftarrow \text{ApplyRotation}(\boldsymbol{R}, \boldsymbol{p})$
7: $(\bar{u}, \frac{\mathrm{d}}{\mathrm{d}t}\bar{u}) \leftarrow \text{jvp}(\bar{u}^{\boldsymbol{\theta}}, (\boldsymbol{x}, \boldsymbol{p}, \Delta t), (\boldsymbol{p}/\boldsymbol{m}, \boldsymbol{f}, -1))$
8: $\bar{u}_{\text{tgt}} \leftarrow \text{stack}([\boldsymbol{p}/\boldsymbol{m}, \boldsymbol{f}]) + \Delta t \cdot \frac{\mathrm{d}}{\mathrm{d}t}\bar{u}$
9: $\mathcal{L}_{\boldsymbol{\theta}} \leftarrow \|\bar{u} - \text{stopgrad}(\bar{u}_{\text{tgt}})\|_2^2$

---

**Sampling timesteps.** The choice of the base distribution $q(\tau)$ for the timestep $\Delta t$ has a significant impact on the structure of the loss landscape and thus the training stability of our method (Zhang et al., 2026). We experiment with three different choices: (i) the uniform distribution, (ii) the absolute difference of two logit-normal random variables, and (iii) our proposed mixture distribution. In all cases, $q(\tau)$ is defined over $[0, 1]$. During training, we sample $\tau \sim q(\tau)$ and set

$$\Delta t = \tau \cdot \Delta t_{\max}, \qquad (21)$$

where $\Delta t_{\max}$ is the maximal timestep used during training. First, we draw $\tau \sim \mathcal{U}(0, 1)$ as a simple baseline. Second, inspired by Geng et al. (2025a), we sample $\tau$ as the absolute difference of two i.i.d. logit-normal random variables (Esser et al., 2024). Specifically, we sample $z_1, z_2 \sim \mathcal{N}(-0.4, 1.0)$, map them to $(0, 1)$ via the sigmoid function $\sigma(\cdot)$, and set

$$\tau = \big|\sigma(z_1) - \sigma(z_2)\big|. \qquad (22)$$

In our experiments, we find that emphasizing smaller timesteps can help the training stability for some systems. However, this distribution has low density for $[0.8, 1.0]$, which results in an increased integration error for timesteps $\Delta t \to \Delta t_{\max}$ (see Appendix G.6). To mitigate this issue, we propose a mixture distribution, which allows learning for larger $\Delta t$, while maintaining the stability benefits of right-skewed distributions. Specifically, we define

$$q(\tau) = 0.98\mathcal{B}(1, 2) + 0.02\mathcal{U}(0, 1), \qquad (23)$$

where $\mathcal{B}(\cdot, \cdot)$ is a beta distribution and $\mathcal{U}(\cdot, \cdot)$ is a uniform distribution. We visualize all three distributions in Figure 10.

**Force matching ratio.** As discussed in Section 4, the accurate learning at $\Delta t = 0$ is essential, as it anchors the consistency condition for timesteps $\Delta t > 0$. However, for any continuous distribution $q(\tau)$ over $[0, 1]$, it holds $\mathbb{P}(\tau = 0) = 0$. Therefore, with probability $q_{\Delta t = 0}$, we set $\Delta t = 0$ during training. Otherwise, we draw $\tau \sim q(\tau)$ and set $\Delta t = \tau \cdot \Delta t_{\max}$.

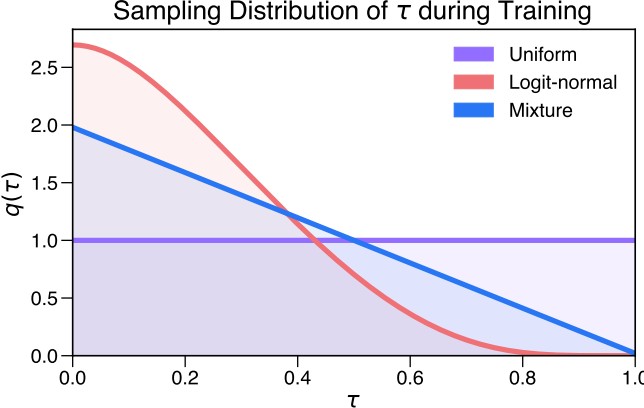

*Figure 10.* Sampling distributions for $\tau$. We experiment with three different $\tau$-sampling distributions in this work: (i) a uniform distribution, (ii) the logit-normal difference distribution inspired by Geng et al. (2025a), and (iii) our proposed mixture distribution.

**Sampling momenta.** Standard MLFF datasets (Eastman et al., 2023; Ganscha et al., 2025; Levine et al., 2025) typically consist of single point calculations with molecular geometries and force labels $(\boldsymbol{x}, \boldsymbol{f})$ but without $\boldsymbol{p}$ labels. For each training example, we therefore draw momenta $\boldsymbol{p}$ independently from Maxwell–Boltzmann distributions at varying temperatures. Specifically, we sample $T \sim \mathcal{N}(\mu_T, \sigma_T^2)$ and ensure non-negative values via clipping $\tilde{T} = \max\{0, T\}$. We then draw

$$\boldsymbol{p}_i \sim \mathcal{N}\left(0, \boldsymbol{m}_i k_B \tilde{T} \, \mathbb{I}_3\right), \tag{24}$$

where $\boldsymbol{m}_i$ is the mass of the $i$-th particle, and $k_B$ is the Boltzmann constant. This is justified for the *separable* Hamiltonians commonly assumed in MD, where the force depends only on $\boldsymbol{x}$ and is independent of $\boldsymbol{p}$. At inference time, we may similarly sample the initial momenta, but from the Maxwell–Boltzmann distribution at a fixed target temperature, i.e., $\sigma_T^2 = 0$.

**Training with zero drift, zero angular momentum and zero momentum.** For an $N$-particle system with positions $\{\boldsymbol{x}_i\}_{i=1}^N$, momenta $\{\boldsymbol{p}_i\}_{i=1}^N$, and masses $\{\boldsymbol{m}_i\}_{i=1}^N$, let $M = \sum_{i=1}^N \boldsymbol{m}_i$ denote the total mass of the system. To simplify the training objective, we ensure $\sum_{i=1}^N \boldsymbol{p}_i = \boldsymbol{0}$ by removing the center-of-mass drift $\boldsymbol{v}^{(\mathrm{com})} = \frac{1}{M} \sum_{i=1}^N \boldsymbol{p}_i$ from the momenta:

$$\boldsymbol{p}_i \leftarrow \boldsymbol{p}_i - \boldsymbol{m}_i \boldsymbol{v}^{(\mathrm{com})}. \tag{25}$$

This transformation corresponds to a transfer to a different inertial system and therefore can be done without changing the physics of the system. During simulation, we ensure that the system is always in the same inertial state as during training via a lightweight inference filter. As the kinetic energy may change under the drift removal, we rescale the momenta,

$$\boldsymbol{p}_i \leftarrow \sqrt{\frac{T_{\mathrm{tgt}}}{T_{\mathrm{cur}}}} \boldsymbol{p}_i, \tag{26}$$

where $T_{\mathrm{cur}}$ denotes the temperature after the drift removal and $T_{\mathrm{tgt}}$ denotes either (i) the temperature before the drift removal during training or (ii) the target temperature of the simulation during inference. We compute temperatures $T_{\mathrm{cur}}$ (and $T_{\mathrm{tgt}}$ during training) based on $3N - 3$ degrees of freedom, where we have accounted for the zero center-of-mass drift.

A fundamental property of Hamiltonian mechanics is the conservation of total angular momentum for an isolated system. The total angular momentum is defined as $\mathbf{L} = \sum_{i=1}^N \boldsymbol{r}_i \times \boldsymbol{p}_i$ with $\boldsymbol{r}_i = \boldsymbol{x}_i - \boldsymbol{x}^{(\mathrm{com})}$ and $\boldsymbol{x}^{(\mathrm{com})} = \frac{1}{M} \sum_{i=1}^N \boldsymbol{m}_i \boldsymbol{x}_i$. However, total angular momentum, unlike total linear momentum, changes the dynamics of the system, and therefore cannot be simply removed. Still, we found it beneficial in our early experiments to occasionally present the edge case of zero total angular momentum to the model. Therefore, with probability $q_{\mathbf{L}=\mathbf{0}}$ we train with zero angular momentum by removing the global rotational component from the momenta,

$$\boldsymbol{p}_i \leftarrow \boldsymbol{p}_i - \boldsymbol{m}_i \left(\boldsymbol{\omega} \times \boldsymbol{r}_i\right), \tag{27}$$

where $\boldsymbol{\omega}$ is obtained by solving $\mathbf{I}\boldsymbol{\omega} = \mathbf{L}$ with the inertia tensor $\mathbf{I} = \sum_{i=1}^N \boldsymbol{m}_i \left(\|\boldsymbol{r}_i\|^2 \mathbb{I}_3 - \boldsymbol{r}_i \boldsymbol{r}_i^\top\right)$.

Another interesting edge case arises if the momenta for the initial state in phase space are entirely zero, where the dynamics are fully determined by the forces of the system. Therefore, with probability $q_{\boldsymbol{p}=\boldsymbol{0}}$ we train with zero momenta:

$$\boldsymbol{p}_i \leftarrow \boldsymbol{0}. \tag{28}$$

Note that for larger $\Delta t$ the momenta still accumulate and have to be taken into account by the model accordingly.

**Training hyperparameters.** We use the Adam optimizer (Kingma & Ba, 2015) with $\beta_1 = 0.9$, $\beta_2 = 0.95$. We linearly warm up the learning rate over the first 1% of training steps from $\mu_0 = 10^{-6}$ to $\mu_{\max} = 10^{-4}$, followed by a cosine decay to $\mu_{\min} = 10^{-8}$. We do not use weight decay and clip the global gradient norm to 5. We apply equal loss weighting, i.e., $\lambda_{\boldsymbol{v}} = 1$ and $\lambda_{\boldsymbol{f}} = 1$ and use the mixture distribution as the base distribution $q(\tau)$ in our experiments. For all 3D systems, we set $q_{\mathbf{L}=\mathbf{0}} = 0.25$, $q_{\boldsymbol{p}=\boldsymbol{0}} = 0.25$, and $q_{\Delta t=0} = 0.75$. For all other systems, we set $q_{\mathbf{L}=\mathbf{0}} = 0$, $q_{\boldsymbol{p}=\boldsymbol{0}} = 0$, and $q_{\Delta t=\mathbf{0}} = 0$.

## B.2. Inference with Hamiltonian Flow Maps

The direct prediction of forces using a non-equivariant model architecture can introduce artifacts during simulation, including violations of rotational equivariance, momentum drift, and energy instabilities. To mitigate these effects, we can optionally apply a set of filters at inference time, similar to TrajCast (Thiemann et al., 2026) and FlashMD (Bigi et al., 2025a).

The inference algorithm for a single step is summarized in Algorithm 3, where $\mathrm{ApplyUpdate}$ refers to the HFM update in Equation (12). Starting from an initial state $(\boldsymbol{x}_0, \boldsymbol{p}_0)$ at physical time $t = 0$, we iteratively apply Algorithm 3 with fixed $\Delta t$, until the desired simulation time is reached. The initial state $(\boldsymbol{x}_0, \boldsymbol{p}_0)$ consists of the positions $\boldsymbol{x}_0$ of a relaxed structure and momenta $\boldsymbol{p}_0$ sampled from the Maxwell–Boltzmann distribution at target temperature $T$. Similar to training, we ensure zero net linear momentum and then rescale $\boldsymbol{p}_0$ to match the target temperature $T$ (see Equation (25) and Equation (26)).

---

**Algorithm 3** Inference with Hamiltonian Flow Maps

**Require:** Neural network $\bar{u}^{\boldsymbol{\theta}}$, current state $(\boldsymbol{x}_t, \boldsymbol{p}_t)$, timestep $\Delta t$, rotation matrix $\boldsymbol{R} \sim \mathrm{SO}(3)$
1: $\boldsymbol{x}_t \quad , \boldsymbol{p}_t \quad \leftarrow \mathrm{RandomRotationFilter}(\boldsymbol{R}, \boldsymbol{x}_t, \boldsymbol{p}_t)$
2: $\boldsymbol{x}_{t+\Delta t}, \boldsymbol{p}_{t+\Delta t} \leftarrow \mathrm{ApplyUpdate}(\bar{u}^{\boldsymbol{\theta}}, \boldsymbol{x}_t, \boldsymbol{p}_t, \Delta t)$
3: $\boldsymbol{x}_{t+\Delta t}, \boldsymbol{p}_{t+\Delta t} \leftarrow \mathrm{RandomRotationFilter}(\boldsymbol{R}^{-1}, \boldsymbol{x}_{t+\Delta t}, \boldsymbol{p}_{t+\Delta t})$
4: $\boldsymbol{x}_{t+\Delta t}, \boldsymbol{p}_{t+\Delta t} \leftarrow \mathrm{RemoveDriftFilter}(\boldsymbol{x}_{t+\Delta t}, \boldsymbol{p}_{t+\Delta t})$
5: $\qquad\quad \boldsymbol{p}_{t+\Delta t} \leftarrow \mathrm{CoupledConservationFilter}(\boldsymbol{x}_{t+\Delta t}, \boldsymbol{p}_{t+\Delta t})$
6: $\qquad\quad \boldsymbol{p}_{t+\Delta t} \leftarrow \mathrm{Thermostat}(\boldsymbol{p}_{t+\Delta t})$

---

In the following, we discuss the remove drift and random rotation filter from FlashMD. Our proposed coupled energy and angular momentum conservation filter and the used thermostats are discussed in Appendices B.3 and B.4, respectively.

**Remove drift filter.** For an $N$-particle system at time $t$ with positions $\{\boldsymbol{x}_{t,i}\}_{i=1}^N$, momenta $\{\boldsymbol{p}_{t,i}\}_{i=1}^N$, and masses $\{m_i\}_{i=1}^N$, let $M = \sum_{i=1}^N m_i$ denote the total mass. We aim to achieve the conservation of total linear momentum of the system, i.e.,

$$\sum_{i=1}^N \boldsymbol{p}_{t+\Delta t,i} - \sum_{i=1}^N \boldsymbol{p}_{t,i} = 0. \tag{29}$$

One can show that this condition is satisfied by applying a *local* translation to the updated momenta

$$\boldsymbol{p}_{t+\Delta t,i} \leftarrow \boldsymbol{p}_{t+\Delta t,i} + m_i \left( -\boldsymbol{v}_{t+\Delta t}^{(\mathrm{com})} + \boldsymbol{v}_t^{(\mathrm{com})} \right), \tag{30}$$

where

$$\boldsymbol{v}_{t+\Delta t}^{(\mathrm{com})} = \frac{1}{M} \sum_{i=1}^N \boldsymbol{p}_{t+\Delta t,i}, \qquad \boldsymbol{v}_t^{(\mathrm{com})} = \frac{1}{M} \sum_{i=1}^N \boldsymbol{p}_{t,i}, \tag{31}$$

denote the center-of-mass for the current velocities $\boldsymbol{v}_t$, and updated velocities $\boldsymbol{v}_{t+\Delta t}$, respectively.

Since the total momentum is constant, the center-of-mass of the system follows a uniform linear motion, i.e.,

$$\sum_{i=1}^N m_i \boldsymbol{x}_{t+\Delta t,i} - \sum_{i=1}^N m_i \boldsymbol{x}_{t,i} = \Delta t \sum_{i=1}^N \boldsymbol{p}_{t,i}. \tag{32}$$

One can show that this condition is satisfied by applying a *global* translation to the updated positions

$$\boldsymbol{x}_{t+\Delta t,i} \leftarrow \boldsymbol{x}_{t+\Delta t,i} - \boldsymbol{x}_{t+\Delta t}^{\text{(com)}} + \boldsymbol{x}_t^{\text{(com)}} + \Delta t \boldsymbol{v}_t^{\text{(com)}}, \tag{33}$$

where

$$\boldsymbol{x}_t^{\text{(com)}} = \frac{1}{M} \sum_{i=1}^N \boldsymbol{m}_i \boldsymbol{x}_{t,i}, \quad \boldsymbol{x}_{t+\Delta t}^{\text{(com)}} = \frac{1}{M} \sum_{i=1}^N \boldsymbol{m}_i \boldsymbol{x}_{t+\Delta t,i}, \quad \boldsymbol{v}_t^{\text{(com)}} = \frac{1}{M} \sum_{i=1}^N \boldsymbol{p}_{t,i}, \tag{34}$$

denote the center-of-mass for the current positions $\boldsymbol{x}_t$, updated positions $\boldsymbol{x}_{t+\Delta t}$, and current velocities $\boldsymbol{v}_t$, respectively.

**Random rotation filter.** During simulation, we apply random rotations to states to average out artifacts introduced due to violations of rotational equivariance. Notice that the rotation has to be applied to the centered positions, as otherwise this operation may shift the frame center. We summarize the RandomRotationFilter function in Algorithm 4, which applies a random rotation $\boldsymbol{R}$ to the current state $(\boldsymbol{x}_t, \boldsymbol{p}_t)$ and its inverse $\boldsymbol{R}^{-1}$ to the updated state $(\boldsymbol{x}_{t+\Delta t}, \boldsymbol{p}_{t+\Delta t})$ (see Figure 3 and Algorithm 3). The rotations $\boldsymbol{R}, \boldsymbol{R}^{-1}$ are applied based on the ApplyRotation function in Equation (20).

---

**Algorithm 4** RandomRotationFilter

---

**Require:** Neural network $\bar{u}^{\boldsymbol{\theta}}$, state $(\boldsymbol{x}, \boldsymbol{p})$, rotation matrix $\boldsymbol{R} \sim \text{SO}(3)$
1: $\boldsymbol{x} \leftarrow \text{ApplyRotation}(\boldsymbol{R}, \boldsymbol{x} - \boldsymbol{x}_{\text{mean}}) + \boldsymbol{x}_{\text{mean}}$
2: $\boldsymbol{p} \leftarrow \text{ApplyRotation}(\boldsymbol{R}, \boldsymbol{p})$

---

### B.3. Coupled Conservation Filter for Angular Momentum and Energy

In this subsection, we describe our proposed coupled conservation filter for angular momentum and energy. Compared to sequential update schemes, this filter solves a constrained optimization problem in closed form to minimally adjust the momenta after each update step, such that total energy and angular momentum are conserved simultaneously.

**Problem formulation.** For an $N$-particle system with updated positions $\{\boldsymbol{x}_{t+\Delta t,i}\}_{i=1}^N$, updated momenta $\{\boldsymbol{p}_{t+\Delta t,i}\}_{i=1}^N$, and masses $\{\boldsymbol{m}_i\}_{i=1}^N$, let $M = \sum_{i=1}^N \boldsymbol{m}_i$ denote the total mass of the system, and $\boldsymbol{r}_{t+\Delta t,i} = \boldsymbol{x}_{t+\Delta t,i} - \boldsymbol{x}_{t+\Delta t}^{\text{(com)}}$ denote the updated position of the $i$-particle expressed in the center-of-mass frame with $\boldsymbol{x}_{t+\Delta t}^{\text{(com)}} = \frac{1}{M} \sum_{i=1}^N \boldsymbol{m}_i \boldsymbol{x}_{t+\Delta t,i}$.

We seek corrected updated momenta $\{\boldsymbol{p}'_{t+\Delta t,i}\}_{i=1}^N$ such that

$$\sum_{i=1}^N \frac{\|\boldsymbol{p}'_{t+\Delta t,i}\|^2}{2\boldsymbol{m}_i} = K_{\text{tgt}}, \tag{35}$$

$$\sum_{i=1}^N \boldsymbol{r}_{t+\Delta t,i} \times \boldsymbol{p}'_{t+\Delta t,i} = \mathbf{L}_{\text{tgt}}, \tag{36}$$

while minimizing the mass-weighted momentum change

$$\min_{\boldsymbol{p}'} \sum_{i=1}^N \frac{1}{2\boldsymbol{m}_i} \|\boldsymbol{p}'_{t+\Delta t,i} - \boldsymbol{p}_{t+\Delta t,i}\|^2. \tag{37}$$

**Target definitions.** In Equation (36), the target $\mathbf{L}_{\text{tgt}}$ is the total angular momentum before the simulation step:

$$\mathbf{L}_{\text{tgt}} = \sum_{i=1}^N \boldsymbol{r}_{t,i} \times \boldsymbol{p}_{t,i}. \tag{38}$$

In Equation (35), the target kinetic energy $K_{\text{tgt}}$ refers to

$$K_{\text{tgt}} = K_{\text{cur}} - \Delta E_{\text{tot}}. \tag{39}$$

The total energy decomposes into kinetic and potential energy. We need to evaluate a potential $V$ to compute the potential energy before and after the simulation step. For this we can either apply (i) a separately trained MLFF or (ii) attach a separate head for predicting the potential energy to the HFM model. In this work, we experiment with both strategies (see Appendix F). Notice that the computation of the potential energy can be reused for the next simulation step, such that the computational overhead is a single potential evaluation per simulation step. The kinetic energy can be obtained directly from the momenta. We use both to compute the total energy before and after the simulation step. The difference in total energy, denoted as $\Delta E_{\text{tot}}$, needs to be accounted for by correction of the momenta after the simulation step.

**Lagrangian.** Introduce a scalar Lagrange multiplier $\alpha \in \mathbb{R}$ for the kinetic-energy constraint (35) and a vector Lagrange multiplier $\boldsymbol{\beta} \in \mathbb{R}^3$ for the angular-momentum constraint (36). The Lagrangian is

$$\mathcal{L}(p', \alpha, \boldsymbol{\beta}) = \sum_{i=1}^{N} \frac{1}{2\boldsymbol{m}_i} \|\boldsymbol{p}'_{t+\Delta t,i} - \boldsymbol{p}_{t+\Delta t,i}\|^2 + \alpha\left(\sum_{i=1}^{N} \frac{\|\boldsymbol{p}'_{t+\Delta t,i}\|^2}{2\boldsymbol{m}_i} - K_{\text{tgt}}\right) + \boldsymbol{\beta} \cdot \left(\sum_{i=1}^{N} \boldsymbol{r}_{t+\Delta t,i} \times \boldsymbol{p}'_{t+\Delta t,i} - \mathbf{L}_{\text{tgt}}\right). \quad (40)$$

Taking the derivative of $\mathcal{L}$ with respect to $\boldsymbol{p}'_{t+\Delta t,i}$ and setting it to zero yields

$$\frac{1}{\boldsymbol{m}_i}(\boldsymbol{p}'_{t+\Delta t,i} - \boldsymbol{p}_{t+\Delta t,i}) + \alpha \frac{\boldsymbol{p}'_{t+\Delta t,i}}{\boldsymbol{m}_i} + \boldsymbol{\beta} \times \boldsymbol{r}_{t+\Delta t,i} = 0. \quad (41)$$

Multiplying by $\boldsymbol{m}_i$ and rearranging gives

$$\boldsymbol{p}'_{t+\Delta t,i} = \frac{1}{1+\alpha}\left(\boldsymbol{p}_{t+\Delta t,i} - \boldsymbol{m}_i(\boldsymbol{\beta} \times \boldsymbol{r}_{t+\Delta t,i})\right). \quad (42)$$

**Angular momentum constraint.** Inserting Equation (42) into the angular-momentum constraint (36) yields

$$\frac{1}{1+\alpha}\left(\mathbf{L} - \mathbf{I}\boldsymbol{\beta}\right) = \mathbf{L}_{\text{tgt}} \quad (43)$$

with inertia $\mathbf{I} = \sum_{i=1}^{N} \boldsymbol{m}_i\left(\|\boldsymbol{r}_{t+\Delta t,i}\|^2 \mathbb{I}_3 - \boldsymbol{r}_{t+\Delta t,i}\boldsymbol{r}_{t+\Delta t,i}^{\top}\right)$ and angular momentum $\mathbf{L} = \sum_{i=1}^{N} \boldsymbol{r}_{t+\Delta t,i} \times \boldsymbol{p}_{t+\Delta t,i}$. Solving for $\boldsymbol{\beta}$ gives

$$\boldsymbol{\beta}(\alpha) = \mathbf{I}^{-1}\mathbf{L} - (1+\alpha)\mathbf{I}^{-1}\mathbf{L}_{\text{tgt}}. \quad (44)$$

**Momentum decomposition.** For ease of notation, let's define:

$$\lambda := 1 + \alpha, \qquad \boldsymbol{p}^{(0)}_{t+\Delta t,i} := \boldsymbol{p}_{t+\Delta t,i} - \boldsymbol{m}_i\left((\mathbf{I}^{-1}\mathbf{L}) \times \boldsymbol{r}_{t+\Delta t,i}\right), \qquad \boldsymbol{p}^{(1)}_{t+\Delta t,i} := \boldsymbol{m}_i\left((\mathbf{I}^{-1}\mathbf{L}_{\text{tgt}}) \times \boldsymbol{r}_{t+\Delta t,i}\right). \quad (45)$$

Using the expression for $\boldsymbol{\beta}(\lambda)$, we obtain

$$\boldsymbol{p}_{t+\Delta t,i} - \boldsymbol{m}_i(\boldsymbol{\beta}(\lambda) \times \boldsymbol{r}_{t+\Delta t,i}) = \boldsymbol{p}^{(0)}_{t+\Delta t,i} + \lambda\boldsymbol{p}^{(1)}_{t+\Delta t,i}. \quad (46)$$

And for the corrected momenta

$$\boldsymbol{p}'_{t+\Delta t,i}(\lambda) = \frac{1}{\lambda}\left(\boldsymbol{p}^{(0)}_{t+\Delta t} + \lambda\boldsymbol{p}^{(1)}_{t+\Delta t,i}\right). \quad (47)$$

**Kinetic energy as a function of $\lambda$.** The kinetic energy of the corrected momenta is

$$K(\lambda) = \sum_{i=1}^{N} \frac{\|\boldsymbol{p}'_{t+\Delta t,i}(\lambda)\|^2}{2\boldsymbol{m}_i} = \frac{1}{\lambda^2}\sum_{i=1}^{N} \frac{\|\boldsymbol{p}^{(0)}_{t+\Delta t,i} + \lambda\boldsymbol{p}^{(1)}_{t+\Delta t,i}\|^2}{2\boldsymbol{m}_i}. \quad (48)$$

Introduce the scalar quantities

$$A = \sum_{i=1}^{N} \frac{\|\boldsymbol{p}^{(0)}_{t+\Delta t,i}\|^2}{2\boldsymbol{m}_i}, \qquad B = \sum_{i=1}^{N} \frac{\boldsymbol{p}^{(0)}_{t+\Delta t,i} \cdot \boldsymbol{p}^{(1)}_{t+\Delta t,i}}{\boldsymbol{m}_i}, \qquad C = \sum_{i=1}^{N} \frac{\|\boldsymbol{p}^{(1)}_{t+\Delta t,i}\|^2}{2\boldsymbol{m}_i}. \quad (49)$$

With these definitions, we have

$$K(\lambda) = \frac{1}{\lambda^2}\left(A + B\lambda + C\lambda^2\right). \quad (50)$$

**Quadratic equation for** $\lambda$**.** Imposing the energy constraint (35), $K(\lambda) = K_{\text{tgt}}$, leads to a quadratic equation:

$$(C - K_{\text{tgt}})\lambda^2 + B\lambda + A = 0. \tag{51}$$

Once $\lambda$ is determined, the energy multiplier is recovered as

$$\alpha = \lambda - 1. \tag{52}$$

**Final corrected momenta.** The corrected momenta satisfying both constraints are

$$\boldsymbol{p}'_{t+\Delta t,i} = \frac{1}{1+\alpha}\Big(\boldsymbol{p}_{t+\Delta t,i} - \boldsymbol{m}_i\big(\boldsymbol{\beta}(\alpha) \times \boldsymbol{r}_{t+\Delta t,i}\big)\Big), \qquad \boldsymbol{\beta}(\alpha) = \mathbf{I}^{-1}\mathbf{L} - (1+\alpha)\mathbf{I}^{-1}\mathbf{L}_{\text{tgt}}. \tag{53}$$

We note that this optimization problem can be efficiently solved as a quadratic equation, however it does not represent the direct projection onto the correct manifold preserving all relevant quantities, as we do not correct predicted positions. This is mainly motivated by inference speed, since updating positions as well would lead to a non-linear problem, which must be solved iteratively and requires evaluating the MLFF potential multiple times, which is computationally expensive.

### B.4. Thermostats

We perform NVT simulations based on three different thermostats throughout this work:

**Langevin thermostat.** The Langevin thermostat (Bussi & Parrinello, 2007) applies *local* stochastic corrections to momenta, allowing to sample the constant-temperature (NVT) ensemble. This makes it the most robust choice for our experiments. During simulations, we apply all our inference filters (see Section 4.3), i.e., drift removal, coupled conservation of total energy and total angular momentum, as well as random rotations.

**CSVR thermostat.** The CSVR thermostat (Bussi et al., 2007) applies *global* stochastic velocity rescaling, to sample from the NVT ensemble with the desired target temperature. We find that global velocity rescaling in some cases leads to interference with our global correction for total energy, and therefore use only the drift removal filter, random rotation filter, and an alternative filter that only preserves angular momentum.

**Nosé-Hoover thermostat.** The Nosé-Hoover thermostat (Martyna et al., 1996) applies *global* corrections in a fully deterministic and reversible way. We note that the lack of stochasticity amplifies model errors, which are caused by the non-symplectic nature of our map. In some of our experiments this leads to noticeable artifacts. Therefore, we use only the drift removal filter, random rotation filter, and an alternative filter that only preserves angular momentum for this thermostat.

### B.5. Units

We use the default `ase` (Larsen et al., 2017) units throughout our molecular experiments, i.e., energy in eV, distances in Å, forces in eV/Å.

### B.6. Compute Infrastructure

We perform training and simulation of our models on a single NVIDIA A100 80GB GPU or NVIDIA H100 80GB GPU, depending on availability. For fair comparison, the timings were always measured on a single NVIDIA A100 80GB GPU.

### B.7. Software Licences

In this work, we use `jax` (Bradbury et al., 2018) (Apache-2.0) and the its machine learning library `flax` (Heek et al., 2024) (Apache-2.0). We adopt the transformer architecture from Frank et al. (2025) (MIT), which is implemented on top of `e3x` (Unke & Maennel, 2024) (Apache-2.0). For simulation utilities, we use `ase` (Larsen et al., 2017) (GNU LGPLv2.1).

## C. Sufficiency of the Proposed Loss

In Equations (7)–(9), we derive a necessary condition that every Hamiltonian flow map needs to fulfill. In the following, we will derive that this is also a sufficient condition following ideas from Geng et al. (2025a).

While Geng et al. (2025a) derive a consistency condition by differentiating the average velocity field with respect to the end time (a backward consistency condition), our method relies on differentiating with respect to the start time (a forward consistency condition). Geng et al. (2025a, Appendix B.3) show the sufficiency of their loss, and we show here that our forward formulation provides an equally sufficient condition for learning correct Hamiltonian dynamics. We start from Equation (7), which is equivalent to the loss term used in Equation (10), as we show in Section 4. Once we revert the derivative with respect to an integration by $t$, an integration constant appears on both sides such that

$$\frac{\mathrm{d}}{\mathrm{d}t}\left[\Delta t \bar{u}(\boldsymbol{x}_t, \boldsymbol{p}_t, \Delta t)\right] = \frac{\mathrm{d}}{\mathrm{d}t}\int_t^{t^*}\binom{\boldsymbol{v}_\tau}{\boldsymbol{f}_\tau}\mathrm{d}\tau \implies \Delta t \bar{u}(\boldsymbol{x}_t, \boldsymbol{p}_t, \Delta t) + C_1 = \int_t^{t^*}\binom{\boldsymbol{v}_\tau}{\boldsymbol{f}_\tau}\mathrm{d}\tau + C_2. \tag{54}$$

For the special case of $t = t^* \to \Delta t = 0$, we have $\int_t^{t^*}\binom{\boldsymbol{v}_\tau}{\boldsymbol{f}_\tau}\mathrm{d}\tau = 0$ and therefore $C_1 = C_2$.

As both integration constants are equal, this shows that Equation (6)$\implies$ Equation (7) and therefore that our loss is sufficient to learn the correct integral. Crucially, for Hamiltonian systems, this implies that if the training loss is perfectly minimized, the predictions of the network will model the true dynamics of the system. The only systematic source of error would be a bias in the given force labels, or a generalization error introduced by the model.

## D. Comparison with Few-Step Diffusion Models

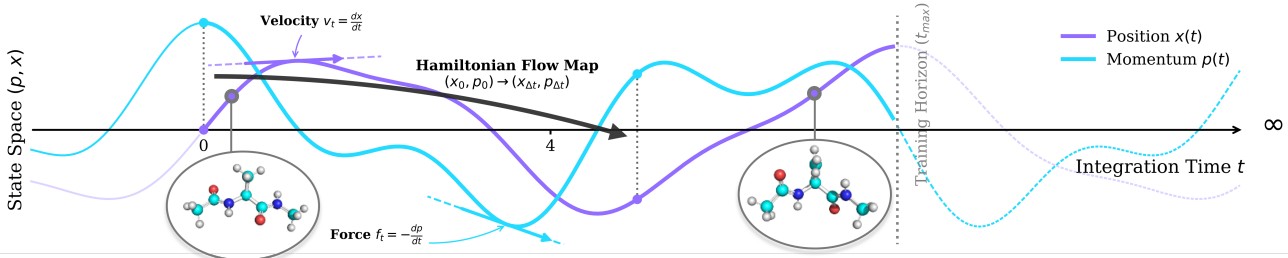

*Figure 11.* Our loss formulation can be understood in a similar way like few-step generative models. Instead of integrating the probability flow ODE, we integrate Hamiltonian dynamics. In our setting, the forward evolution of momenta and positions must be considered as a coupled integration problem, i.e., the Hamiltonian Flow Map (HFM) predicts a future state in phase space.

**Generative sampling vs. deterministic dynamics.** Our approach shares a conceptual foundation with accelerated sampling in diffusion and flow-matching models (Sohl-Dickstein et al., 2015; Ho et al., 2020; Song et al., 2021; Albergo & Vanden-Eijnden, 2023; Lipman et al., 2023). Recent advances in this field employ few-step generation techniques, such as consistency models or rectified flows, to bypass the expensive numerical integration of the probability flow-ODE (Song et al., 2023; Song & Dhariwal, 2024; Lu & Song, 2025; Liu et al., 2023). We visualize our training target in Figure 11.

As described in Section 4.2, our learning process can be viewed as a form of self-distillation across time. Zhang et al. (2026) provide a rigorous analysis of this mechanism in the context of generative modeling, demonstrating that the *Mean Flow* loss (Geng et al., 2025a) analytically decomposes into two distinct gradient components: *Trajectory Flow Matching* and *Trajectory Consistency*. In our context, the loss decomposes into a force matching and an integral fitting part.

Unlike generative approaches that utilize noise injection and denoising scores to construct a transport map between distributions, our training objective is entirely deterministic. Because classical Hamiltonian dynamics dictates a precise, causal evolution for every point in phase space, our objective can be seen as a physics-informed loss that directly penalizes deviations from the integrated equations of motion.

**Probability flow-ODE vs. Hamiltonian dynamics.** Generative flow models typically rely on a probability flow-ODE designed to transport a prior distribution to a target data distribution (and thus always requires data pairs). The vector field

governing this flow is often a design choice, and can be constructed to be straight or smooth to facilitate easy integration (Liu et al., 2023), which has a major impact on the required number of integration steps. In contrast, the vector field in our setting is dictated by physical laws, specifically Hamilton's equations. Unlike diffusive dynamics often found in generative models, Hamiltonian dynamics respect the conservation of energy and symplecticity. Consequently, our objective is not to find any flow that transports samples between distributions, but to learn the specific physical flow, i.e., the true dynamics of the system, which are unknown a priori.

**Finite interval vs. unbounded time.**  Generative flows operate on a fixed, finite time interval, typically $t \in [0, 1]$, solving a boundary value problem where the endpoints are known. Hamiltonian simulation, conversely, is an initial value problem over an effectively unbounded time horizon ($t \to \infty$). We do not aim to reach a specific target distribution (we do not even know the target during training), but rather to iterate the learned map indefinitely in an auto-regressive manner. This requires the learned map to be stable under repeated composition, crucially evoking the need to avoid catastrophic accumulation of errors, a constraint that is generally less important in single-shot generative tasks.

**The Lyapunov limit.**  For generative flow models, the complexity of few-shot maps is determined by the interplay of source and target distributions, as well as the choice of the imposed vector field or interpolant. In Hamiltonian mechanics, the difficulty is governed by the system's chaoticity, quantified by the maximal Lyapunov exponent: Even for simple physical systems that might seem easy to integrate at first, the divergence of trajectories typically grows exponentially, imposing a hard physical limit beyond which the flow map becomes effectively unlearnable by a smooth neural network. Generative flow models can sacrifice accuracy for speed to a certain extent by smoothing out complex transport paths. In contrast, our method must faithfully reconstruct the evolution of phase space up to some $\Delta t$ that lies within the chaotic limit.

## E. Model Architecture

In this section, we describe the three model architectures used throughout this work. However, we want to emphasize that our proposed framework is largely architecture-agnostic: any existing architecture, which is suitable for the underlying system (e.g. architectures for atomistic systems from the generative modeling or machine learning force field literature) can be used to learn Hamiltonian Flow Maps (HFMs) with minimal architectural changes. We provide a list of all hyperparameters and additional implementation details in the corresponding subsections of each experiment in Appendix F.

### E.1. Multi-Layer Perceptron

For the toy datasets, we learn Hamiltonian Flow Maps (HFMs) using a simple Multi-Layer Perceptron.

**Embeddings.**  We obtain the initial embedding of the $i$-th particle as the combination of features based on the time, position and momentum information using three separate two-layer MLPs,

$$\boldsymbol{h}_i = \mathbf{MLP}(\mathbf{Embed}_t(\Delta t)) + \mathbf{MLP}(\boldsymbol{x}_i) + \mathbf{MLP}(\boldsymbol{p}_i) \in \mathbb{R}^H, \tag{55}$$

where $\mathbf{Embed}_t(\Delta t) \in \mathbb{R}^H$ denotes Gaussian random Fourier features (Tancik et al., 2020) of the time $\Delta t \in [0, \Delta t_{\max}]$. As we encode absolute positions $\boldsymbol{x}_i \in \mathbb{R}^3$, these embeddings do not obey any form of Euclidean symmetries: they do not transform equivariantly under rotation of the input positions, and they are also not invariant to translations of the system.

**Refinement layers.**  The initial embedding is further refined using a three-layer MLP,

$$\boldsymbol{h}_i = \mathbf{MLP}(\boldsymbol{h}_i) \in \mathbb{R}^H, \tag{56}$$

which produces the final latent representation shared by all prediction heads.

**Prediction heads.**  Finally, we predict the mean velocities and forces for each particle using two separate prediction heads,

$$\bar{\boldsymbol{v}}_i = \mathbf{MLP}(\boldsymbol{h}_i) \in \mathbb{R}^3, \tag{57}$$
$$\bar{\boldsymbol{f}}_i = \mathbf{MLP}(\boldsymbol{h}_i) \in \mathbb{R}^3, \tag{58}$$

where each prediction head consists of a two-layer MLP.

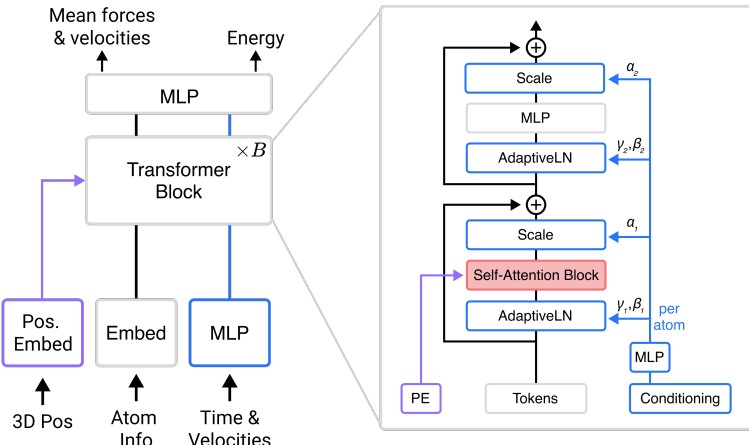

*Figure 12.* Translation-invariant transformer architecture adapted from Frank et al. (2025). PE: Relative positional embeddings are used for scaling attention coefficients in the transformer. Conditioning: Time and Velocity inputs are passed into each transformer block via adaptive layer norms.

### E.2. Translation-invariant Transformer

For the gravitational N-body particle system and the molecular systems, we adopt the translation-invariant Transformer architecture for 3D geometries proposed by Frank et al. (2025), which operates on pairwise displacement vectors rather than the absolute particle positions. To adapt this architecture to our setting, we introduce two modifications:

(1) particle-wise conditioning on time, velocities, and masses, and

(2) multiple prediction heads with task-specific adaptive layer normalizations.

For convenience, we summarize the components of the original architecture below and describe our modifications in detail. We provide an overview of the model architecture in Figure 12.

**Token embeddings.** The initial token representation of the $i$-th particle is obtained via

$$\boldsymbol{h}_i = \mathbf{Embed}_z(z_i) \in \mathbb{R}^H, \tag{59}$$

where $\mathbf{Embed}_z(z_i) \in \mathbb{R}^H$ denotes a learnable embedding based on the particle type $z_i \in \mathbb{N}_+$.

**Positional embeddings.** We construct translation-invariant relative positional embeddings (rPE) based on the distance $d_{ij} = \|\boldsymbol{x}_i - \boldsymbol{x}_j\|_2$ and the and normalized displacement vector $\hat{\boldsymbol{r}}_{ij} = (\boldsymbol{x}_i - \boldsymbol{x}_j)/\|\boldsymbol{x}_i - \boldsymbol{x}_j\|_2$ between particle $i$ and $j$ using a two-layer MLP,

$$\boldsymbol{e}_{ij}^{\mathrm{rPE}} = \mathbf{MLP}([\boldsymbol{\phi}(d_{ij}); \hat{\boldsymbol{r}}_{ij}] \in \mathbb{R}^H, \tag{60}$$

where $\boldsymbol{\phi} : \mathbb{R} \to \mathbb{R}^{M_{\mathrm{RBF}}}$ is a radial basis function, $[\cdot; \cdot]$ denotes concatenation, and $\odot$ denotes element-wise multiplication. As the relative positional embeddings are based on distance and normalized displacement vector between pairs of particles, these embeddings are translation-invariant.

**Attention mechanism.** The self-attention update is defined as

$$\boldsymbol{h}_i = \boldsymbol{h}_i + \sum_{j=1}^{N} \mathbf{softmax}_j \left( \frac{(\boldsymbol{W}_Q \boldsymbol{h}_i)^\top (\boldsymbol{W}_K \boldsymbol{h}_j \odot \boldsymbol{W}_K^E \boldsymbol{e}_{ij}^{\mathrm{rPE}})}{\sqrt{H}} \right) \left( \boldsymbol{W}_V \boldsymbol{h}_j \odot \boldsymbol{W}_V^E \boldsymbol{e}_{ij}^{\mathrm{rPE}} \right), \tag{61}$$

where $\boldsymbol{W}_Q, \boldsymbol{W}_K, \boldsymbol{W}_V, \boldsymbol{W}_K^E, \boldsymbol{W}_V^E \in \mathbb{R}^{H \times H}$ are learnable weight matrices and $\odot$ denotes element-wise multiplication. For ease of notation, we present the self-attention update with a single attention head, but apply multi-head attention with $n_{\mathrm{heads}}$ (Vaswani et al., 2017).

**Conditioning strategy.** We perform particle-wise conditioning based on time $\Delta t \in [0, t_{\max}]$, as well as the mass $\boldsymbol{m}_i > 0$ and velocity $\boldsymbol{v}_i \in \mathbb{R}^3$ of the $i$-th particle. First, we construct the time-based conditioning token based on a two-layer MLP,

$$\boldsymbol{c}^t = \mathbf{MLP}(\mathbf{Embed}_t(\Delta t)) \in \mathbb{R}^H, \tag{62}$$

where $\mathbf{Embed}_{\Delta t}(\Delta t) \in \mathbb{R}^H$ denotes Gaussian random Fourier features (Tancik et al., 2020) of the time $\Delta t$. Next, we encode the magnitude of the velocity $\boldsymbol{v}_i$ of the $i$-th particle using a Gaussian basis expansion, similar to Thiemann et al. (2026), followed by an element-specific linear transformation,

$$\mathbf{Embed}_{\boldsymbol{v}}(\boldsymbol{v}_i) = \boldsymbol{W}_{z_i} \psi(\|\boldsymbol{v}_i\|_2) \in \mathbb{R}^{M_{\text{Gaussian}}}, \tag{63}$$

where $\psi(\cdot) = [\psi_1(\cdot), \ldots, \psi_{M_{\text{Gaussian}}}(\cdot)]^\top \in \mathbb{R}^M$ is a set of $M_{\text{Gaussian}}$ Gaussian basis functions, $\boldsymbol{W}_{z_i} \in \mathbb{R}^{M_{\text{Gaussian}} \times M_{\text{Gaussian}}}$ is a learnable weight matrix, and $z_i$ is the type of the $i$-th particle. The velocity $\boldsymbol{v}_i$ is then encoded using a two-layer MLP,

$$\boldsymbol{c}_i^{\boldsymbol{v}} = \mathbf{MLP}([\mathbf{Embed}_{\boldsymbol{v}}(\boldsymbol{v}_i); \boldsymbol{v}_i]) \in \mathbb{R}^H, \tag{64}$$

where $[\cdot; \cdot]$ denotes concatenation. Similarly, the mass $\boldsymbol{m}_i$ of the $i$-th particle is encoded by another two-layer MLP

$$\boldsymbol{c}_i^{\boldsymbol{m}} = \mathbf{MLP}(\boldsymbol{m}_i) \in \mathbb{R}^H. \tag{65}$$

Finally, the particle-specific conditioning token $\boldsymbol{c}_i$ is obtained via concatenation followed by a two-layer MLP,

$$\boldsymbol{c}_i = \mathbf{MLP}([\boldsymbol{c}^t; \boldsymbol{c}_i^{\boldsymbol{v}}; \boldsymbol{c}_i^{\boldsymbol{m}}]) \in \mathbb{R}^H. \tag{66}$$

Given the particle-specific conditioning token $\boldsymbol{c}_i$, we can compute scale, shift and gating parameters via

$$[\boldsymbol{\gamma}_i; \boldsymbol{\beta}_i; \boldsymbol{\alpha}_i] = \boldsymbol{W}(\text{SiLU}(\boldsymbol{c}_i)) \tag{67}$$

with weight matrix $\boldsymbol{W} \in \mathbb{R}^{3H \times H}$ initialized to all zeros and $\boldsymbol{\gamma}_i, \boldsymbol{\beta}_i, \boldsymbol{\alpha}_i \in \mathbb{R}^H$. These parameters are used for conditioning based on adaptive layer norm (AdaLN) and adaptive scale (AdaScale) throughout the transformer blocks, defined as

$$\text{AdaLN}(\boldsymbol{h}, \boldsymbol{\gamma}, \boldsymbol{\beta}) = \text{LN}(\boldsymbol{h}) \odot (1 + \boldsymbol{\gamma}) + \boldsymbol{\beta}, \tag{68}$$

and

$$\text{AdaScale}(\boldsymbol{h}, \boldsymbol{\alpha}) = \boldsymbol{h} \odot \boldsymbol{\alpha}, \tag{69}$$

where LN is layer normalization (Ba et al., 2016) without scale and shift parameters and $\odot$ is element-wise multiplication.

**Prediction heads.** Given the token representation $\boldsymbol{h}_i^{[B]} \in \mathbb{R}^H$ for the $i$-th particle after $B$ transformer blocks, we predict the mean velocities and mean forces using two separate prediction heads with task-specific adaptive layer normalizations. We calculate task-specific scale and shift parameters for both tasks based on the particle-specific conditioning token $\boldsymbol{c}_i$,

$$\left[\boldsymbol{\gamma}_i^{\boldsymbol{v}}; \boldsymbol{\beta}_i^{\boldsymbol{v}}; \boldsymbol{\gamma}_i^{\boldsymbol{f}}; \boldsymbol{\beta}_i^{\boldsymbol{f}}\right] = \boldsymbol{W}(\text{SiLU}(\boldsymbol{c}_i)), \tag{70}$$

where $\boldsymbol{\gamma}_i^{\boldsymbol{v}}, \boldsymbol{\beta}_i^{\boldsymbol{v}}, \boldsymbol{\gamma}_i^{\boldsymbol{f}}, \boldsymbol{\beta}_i^{\boldsymbol{f}} \in \mathbb{R}^H$ and $\boldsymbol{W} \in \mathbb{R}^{4H \times H}$ initialized to all zeros. Each head applies AdaLN followed by a two-layer MLP:

$$\bar{\boldsymbol{v}}_i = \mathbf{MLP}(\text{AdaLN}(\boldsymbol{h}_i^{[B]}, \boldsymbol{\gamma}_i^{\boldsymbol{v}}, \boldsymbol{\beta}_i^{\boldsymbol{v}})) \in \mathbb{R}^3, \tag{71}$$

$$\bar{\boldsymbol{f}}_i = \mathbf{MLP}(\text{AdaLN}(\boldsymbol{h}_i^{[B]}, \boldsymbol{\gamma}_i^{\boldsymbol{f}}, \boldsymbol{\beta}_i^{\boldsymbol{f}})) \in \mathbb{R}^3. \tag{72}$$

We optionally attach an additional prediction head to predict the particle-wise energy contributions,

$$E_i = \mathbf{MLP}(\text{AdaLN}(\boldsymbol{h}_i^{[B]}, \boldsymbol{\gamma}_i^E, \boldsymbol{\beta}_i^E)) \in \mathbb{R}, \tag{73}$$

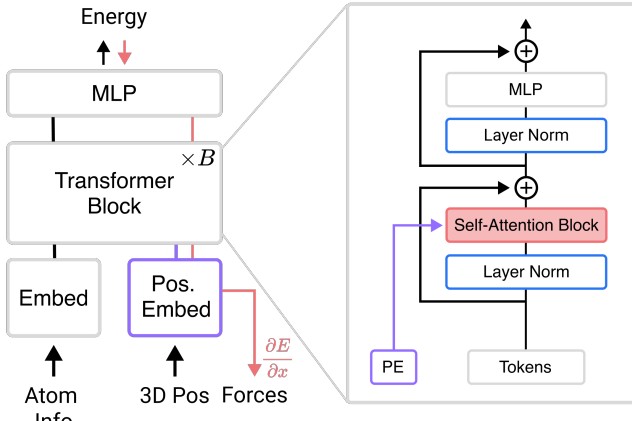

*Figure 13.* Equivariant MLFF Transformer architecture adapted from Frank et al. (2025). PE: Spherical harmonics embeddings are used for scaling attention coefficients in the transformer. We maintain rotational equivariant representations throughout the network.

where the scale and shift parameters $\gamma_i^E, \beta_i^E \in \mathbb{R}^H$ for the energy predictions are obtained as in Equation (70). Based on these predictions, we can then compute the potential energy of the system as the sum of energy contributions per particle,

$$E_{\text{pot}} = \sum_{i=1}^{N} E_i, \tag{74}$$

and use this prediction in our energy and angular momentum conservation filter during inference (see Appendix B.3).

### E.3. $\mathrm{SO}(3)$-Equivariant Transformer

We adopt the $\mathrm{SO}(3)$-equivariant Transformer architecture for 3D geometries proposed by Frank et al. (2025), because of the similarities with the translation-invariant Transformer architecture we use for learning Hamiltonian Flow Maps (HFMs). As we employ the architecture as conservative machine learning force field (MLFF), we remove the time-based conditioning branch from the model. The model uses equivariant formulations for activation functions (Weiler et al., 2018), dense layers (Unke & Maennel, 2024), and layer normalizations (Liao et al., 2024). We give an overview in Figure 13.

Following the notation in Unke & Maennel (2024), the model operates on $\mathrm{SO}(3)$-equivariant tokens $\boldsymbol{h} \in \mathbb{R}^{(L+1)^2 \times H}$, where $L$ denotes the maximal degree of the spherical harmonics. We denote the features corresponding to the $\ell$-th degree as $\boldsymbol{h}^{(\ell)} \in \mathbb{R}^{(2\ell+1) \times H}$, where the $(2\ell+1)$ entries corresponds to the orders $m = -\ell, \ldots, +\ell$ per degree $\ell$. The maximum degree $L$ is chosen to ensure high fidelity at a reasonable computational cost. In this work, we use $L = 1$, similar to PaiNN (Schütt et al., 2021) or TorchMDNet (Thölke & Fabritiis, 2022), which restricts the representation to scalars and vectors. We refer interested readers to Duval et al. (2023) and Tang (2025) for further discussion.

**Token embeddings.** The initial token representation of the $i$-th particle is obtained as a concatenation of $L+1$ components

$$\boldsymbol{h}_i = \bigoplus_{\ell=0}^{L} \boldsymbol{h}_i^{(\ell)} \in \mathbb{R}^{(L+1)^2 \times H}, \qquad \boldsymbol{h}_i^{(\ell)} = \begin{cases} \mathbf{Embed}_z(z_i), & l = 0, \\ \boldsymbol{0}^{(\ell)}, & l > 0, \end{cases} \in \mathbb{R}^{(2\ell+1) \times H}. \tag{75}$$

Therefore, the features of degree $\ell = 0$ are given by a learnable embedding $\mathbf{Embed}_z(z_i) \in \mathbb{R}^{1 \times H}$ based on the particle type $z_i \in \mathbb{N}_+$ and the features of degree $\ell > 0$ are initialized with the zero matrix $\boldsymbol{0}^{(\ell)} \in \mathbb{R}^{(2l+1) \times H}$, i.e., all zeros.

**Positional embeddings.** We construct $\mathrm{SO}(3)$-equivariant Euclidean positional embeddings (PE(3)) based on the distance $d_{ij} = \|\boldsymbol{x}_i - \boldsymbol{x}_j\|_2$ and normalized displacement vector $\hat{\boldsymbol{r}}_{ij} = (\boldsymbol{x}_i - \boldsymbol{x}_j)/\|\boldsymbol{x}_i - \boldsymbol{x}_j\|_2$ between particle $i$ and $j$ as a

concatenation of $L + 1$ components

$$e_{ij}^{\text{PE}(3)} = \bigoplus_{\ell=0}^{L} \phi_\ell(d_{ij}) \odot Y_\ell(\hat{r}_{ij}) \in \mathbb{R}^{(L+1)^2 \times M_{\text{RBF}}}, \tag{76}$$

where $\phi_\ell : \mathbb{R} \mapsto \mathbb{R}^{1 \times M_{\text{RBF}}}$ is a radial basis function, $Y_\ell \in \mathbb{R}^{(2\ell+1)\times 1}$ are spherical harmonics of degree $\ell = 0, \ldots, L$, and $\odot$ is element-wise multiplication with implied broadcasting along the feature dimension s.t. $\phi_\ell(d_{ij}) \odot Y_\ell(\hat{r}_{ij}) \in \mathbb{R}^{(2\ell+1)\times B}$.

As discussed in Frank et al. (2025), these positional embeddings transform equivariantly under rotation of the input positions. Moreover, these positional embeddings are also invariant to translations as displacement vectors are used as inputs.

**Attention mechanism.** In the following, we will make use of equivariant dense layers (Unke & Maennel, 2024), which act on a feature vector $h \in \mathbb{R}^{(L+1)^2 \times H_{\text{in}}}$ per degree channel $\ell$ with $h^{(\ell)} \in \mathbb{R}^{(2\ell+1)\times H_{\text{in}}}$ as

$$Wh = \bigoplus_{\ell=0}^{L} \tilde{h}^{(\ell)}, \qquad \tilde{h}^{(\ell)} = \begin{cases} h^{(\ell)} W_{(\ell)} + \mathbf{b}, & \ell = 0, \\ h^{(\ell)} W_{(\ell)}, & \ell > 0, \end{cases} \tag{77}$$

where $\tilde{h}^{(\ell)} \in \mathbb{R}^{(2\ell+1)\times H_{\text{out}}}$ is a $\ell$-th degree feature component, $W_{(\ell)} \in \mathbb{R}^{H_{\text{in}} \times H_{\text{out}}}$ is a learnable weight matrix and $\mathbf{b} \in \mathbb{R}^{1 \times H_{\text{out}}}$ is a bias applied only to the scalar channel. First, we define a pairwise gating mechanism $E \in \mathbb{R}^{(L+1)^2 \times H}$ as

$$E_{ij}^{(\ell,m)} = W_K^E e_{ij}^{\text{PE}(3)(\ell=0)} \quad \text{for all } \ell = 0, \ldots, L \text{ and } m = -\ell, \ldots, \ell, \tag{78}$$

i.e., the scalar gate gets replicated across all $(\ell, m)$ channels. The self-attention update is then defined as

$$h_i = h_i + \sum_{j=1}^{N} \text{softmax}_j \left( \frac{\langle W_Q h_i, W_K h_j \odot E_{ij} \rangle_F}{\sqrt{(L+1)^2 H}} \right) \left( W_V h_j \otimes W_V^E e_{ij}^{\text{PE}(3)} \right), \tag{79}$$

where $W_Q, W_K, W_V, W_V^E$ are learnable equivariant dense layers, $\langle \cdot \rangle_F$ is the Frobenius inner product, and $\otimes$ is the Clebsch-Gordan tensor product (Thomas et al., 2018; Fuchs et al., 2020; Frank et al., 2022). For ease of notation, we present the self-attention update with a single attention head, but apply multi-head attention with $n_{\text{heads}}$ (Vaswani et al., 2017).

**Prediction head.** Given the token representation $h_i^{[B]} \in \mathbb{R}^{(L+1)^2 \times H}$ for the $i$-th particle after $B$ transformer blocks, we predict energy contributions per particle based on the features $h_i^{[B](\ell=0)} \in \mathbb{R}^{1 \times H}$ of degree $\ell = 0$ using a two-layer MLP,

$$E_i = \text{MLP}(h_i^{[B](\ell=0)}) \in \mathbb{R}. \tag{80}$$

Based on the predictions, we compute the potential energy of the system as the sum of energy contributions over all particles,

$$E_{\text{pot}} = \sum_{i=1}^{N} E_i + E_{\text{rep}}, \tag{81}$$

where $E_{\text{rep}}$ denotes an empirical correction term inspired by the Ziegler–Biersack–Littmark (ZBL) potential (Ziegler & Biersack, 1985), which augments the energy prediction with physical knowledge about nuclear repulsion (Unke et al., 2021a). Finally, the forces are computed as the negative gradient of the total potential energy with respect to the positions,

$$f_i = -\nabla_{x_i} E_{\text{pot}}, \tag{82}$$

using automatic differentiation in `jax` (Bradbury et al., 2018).

# F. Experimental Setup

## F.1. Single Particle Systems

The potential energies are defined as $V_{\text{s}}$ (Spring pendulum) and $V_{\text{b}}$ (Barbanis potential):

$$V_{\text{s}}(\mathbf{x}) = mgy + \frac{1}{2}k \left( \sqrt{x^2 + y^2} - l_0 \right)^2 \tag{83}$$

$$V_{\mathrm{b}}(\mathbf{x}) = \frac{1}{2}(\omega_x^2 x^2 + \omega_y^2 y^2) + \lambda x^2 y^2 \tag{84}$$

To simplify the training of our neural networks, we use a self-consistent dimensionless coordinate system without physical units for these experiments. For our experiments, we set $\lambda = 10.0$ and $g = 9.81$. We train a Hamiltonian Flow map in the conserved energy setting with $E_{tot} = 1.5$ (Barbanis), $E_{tot} = -5$ (Spring pendulum).

**Dataset.** We generate training data by uniformly sampling positions and rejecting them if the potential energy exceeds the desired total energy. We compute force labels using the ground truth potentials. Afterwards, we uniformly sample the direction of momenta and adjust the norm such that the total energy matches exactly. Note that all samples are drawn independently and there is no temporal or trajectory information in the datasets. We sample 100k decorrelated samples in total for each dataset.

**Architecture.** We learn Hamiltonian Flow Maps (HFMs) using the Multi-Layer Perceptron-based network architecture with hidden dimensionality $H = 1024$.

**Hyperparameters.** Both HFM models are trained for 1000 epochs with batch size 512. For both single particle systems, we set $t_{\max} = 2.5$ as the maximum timestep.

**Simulation.** We start simulations from random initial conditions with the correct total energy, generated via rejection sampling and adjustment of momenta magnitude as explained above. We do not use any filters for 1D systems. Ground truth trajectories are generated with Velocity Verlet integration of the true potential using a small timestep of $\Delta t = 0.01$.

### F.2. Gravitational N-Body System

The potential is given as a gravitational interaction without boundary conditions:

$$V_{\mathrm{g}}(\mathbf{x}) = -\sum_i \sum_{i<j} \frac{G\boldsymbol{m}_i \boldsymbol{m}_j}{\|\boldsymbol{x}_i - \boldsymbol{x}_j\|} \tag{85}$$

**Dataset.** The dataset creation is described in Brandstetter et al. (2022), where the ground truth data is generated using Velocity Verlet integration with small timesteps using natural units. The training set consists of 10.000 trajectories, and validation and test set of 2.000 trajectories each, while each trajectory contains 5.000 integration steps. For training with our loss, we disregard the time information in the dataset and train our model using decorrelated (randomly sampled) forces and geometries from the train dataset.

**Architecture.** We learn Hamiltonian Flow Maps (HFMs) using a translation-invariant Transformer architecture. Model hyperparameters are given in Table 3. Since masses are all the same, we don't use mass embeddings. We use a larger cutoff (40) with more radial basis functions (64), since distances between interacting particles can be far.

*Table 3.* Architectural details for the translation-invariant transformer on gravitational $N$-body system. $B$ is the number of blocks, $n_{\mathrm{heads}}$ is the number of attention heads, $H$ is the hidden dimension, and $M_{\mathrm{RBF}}$ and $M_{\mathrm{Gaussian}}$ are the number of radial basis and Gaussian basis functions, respectively.

| Model | $B$ | $n_{\mathrm{heads}}$ | $H$ | $M_{\mathrm{RBF}}$ | $M_{\mathrm{Gaussian}}$ |
|---|---|---|---|---|---|
| Transformer | 6 | 8 | 256 | 64 | 8 |

**Hyperparameters.** We set $\Delta t_{\max} = 0.1$ as the maximum timestep and train with batch size 64 for 300 epochs.

**Simulation.** We use all filters for our HFM model: drift removal, random rotation and coupled energy and angular momentum correction with the ground truth potential. Ground truth is generated with Velocity Verlet integration of the true potential using a small timestep of $\Delta t = 0.001$. The Velocity Verlet integrator in Figure 5 uses the ground truth analytic potential and therefore produces results that are indistinguishable from the ground truth for small timesteps.

### F.3. Small Molecules

**Dataset.** We take four small organic molecules from the MD17 dataset (Chmiela et al., 2017), which is widely used as a force prediction benchmark for MLFFs. We use the split from Fu et al. (2023), i.e., 9,500 configurations for training, 500 configurations for validation, and 10,000 configurations for testing. We sample momenta $\boldsymbol{p}$ from the Maxwell–Boltzmann distribution with mean $\mu_T = 500\,\mathrm{K}$ and standard deviation $\sigma_T = 150\,\mathrm{K}$ as described in Appendix B.

**Architecture.** We learn Hamiltonian Flow Maps (HFMs) using a translation-invariant Transformer architecture. Additionally, we employ an SO(3)-equivariant Transformer as a MLFF baseline. Model hyperparameters are given in Table 4.

*Table 4.* Architectural details for the Translation-invariant and SO(3)-equivariant Transformer on small molecules. $B$ is the number of blocks, $n_{\mathrm{heads}}$ is the number of attention heads, $H$ is the hidden dimension, and $M_{\mathrm{RBF}}$ and $M_{\mathrm{Gaussian}}$ are the number of radial basis and Gaussian basis functions, respectively. Since the SO(3)-Transformer does not have a conditioning branch, we write '–' for $M_{\mathrm{Gaussian}}$.

| **Model** | $B$ | $n_{\mathrm{heads}}$ | $H$ | $M_{\mathrm{RBF}}$ | $M_{\mathrm{Gaussian}}$ |
|---|---|---|---|---|---|
| Transformer | 6 | 8 | 256 | 10 | 8 |
| SO(3)-Transformer | 3 | 4 | 128 | 10 | — |

**Hyperparameters.** We use batch size of 500 and 50,000 epochs for the HFM models, corresponding to a total training time of up to 1d11h on a single NVIDIA A100 80BG GPU. We set $\Delta t_{\mathrm{max}} = 10\mathrm{fs}$ for all systems. For the SO(3)-equivariant Transformer, we train for 5,000 epochs with a batch size of 50 corresponding to a total training time of up to 16h on a single NVIDIA A100 80BG GPU. For MLFF training, we follow standard practice (Batzner et al., 2022) and use as our training objective a weighted sum of energy and force loss terms, i.e., $\mathcal{L} = \lambda_E \mathcal{L}_E + \lambda_F \mathcal{L}_F$, and set $\lambda_E = 0.01$ and $\lambda_F = 0.99$.

**Simulation.** We perform 300 ps of MD simulation in the NVT ensemble using a Langevin thermostat. We set the target temperature to $T = 500\,\mathrm{K}$ and apply a friction of $\gamma = 1/100\,\mathrm{fs}$ in all experiments. We run multiple simulations in parallel and report averaged quantities with standard deviations. Similar to training, we start the simulation with zero net linear momentum, and apply inference-time filters during simulation: drift removal, random rotation and coupled energy and angular momentum correction. We use our own trained MLFFs as potentials for the energy correction filter.

**Evaluation.** Following Fu et al. (2023), we calculate the interatomic distance distribution $h(r)$ averaged over frames from equilibrated trajectories and calculate the MAE w.r.t. the reference data. Let $r$ be the distance, $N$ be the total number of particles, $i, j$ indicate the pairs of atoms that contribute to the distance statistics, and $\delta$ the Dirac Delta function to extract value distributions. For a given configuration $\boldsymbol{x}$, the distribution of interatomic distances $h(r)$ is defined as

$$h(r) = \frac{1}{N(N-1)} \sum_{i=1}^{N} \sum_{j=1, j \neq i}^{N} \delta\left(r - \|\boldsymbol{x}_i - \boldsymbol{x}_j\|_2\right). \tag{86}$$

Finally, we calculate the MAE as

$$\mathrm{MAE}_{h(r)} = \int_{r=0}^{\infty} \left| \langle h(r) \rangle - \langle \hat{h}(r) \rangle \right| dr, \tag{87}$$

where $\langle \cdot \rangle$ denotes the average over frames of an equilibrated trajectories, i.e., $\langle h(r) \rangle := \frac{1}{T} \sum_{t=1}^{T} h_t(r)$ and $\langle \hat{h}(r) \rangle := \frac{1}{T} \sum_{t=1}^{T} \hat{h}_t(r)$, with $\langle h(r) \rangle$ computed from the reference and $\langle \hat{h}(r) \rangle$ computed from the generated trajectory by the model.

### F.4. Paracetamol

**Dataset.** We adopt the MD17 dataset (Chmiela et al., 2017) for Paracetamol with 106,490 samples. We use 80% of the data ($\approx$ 85k configurations) for training. We randomly select 256 training samples to simulate the low data regime. We sample momenta $\boldsymbol{p}$ from the Maxwell–Boltzmann distribution with mean $\mu_T = 500\,\mathrm{K}$ and standard deviation $\sigma_T = 150\,\mathrm{K}$ as described in Appendix B.

**Architecture.** We learn Hamiltonian Flow Maps (HFMs) using a translation-invariant Transformer architecture. Additionally, we employ an SO(3)-equivariant Transformer as an MLFF baseline. Hyperparameters are given in Table 5.

*Table 5.* Architectural details for the Translation-invariant and SO(3)-equivariant Transformer on paracetamol. $B$ is the number of blocks, $n_{\text{heads}}$ is the number of attention heads, $H$ is the hidden dimension, and $M_{\text{RBF}}$ and $M_{\text{Gaussian}}$ are the number of radial basis and Gaussian basis functions, respectively. Since the SO(3)-Transformer does not have a conditioning branch, we write '–' for $M_{\text{Gaussian}}$.

| Model | $B$ | $n_{\text{heads}}$ | $H$ | $M_{\text{RBF}}$ | $M_{\text{Gaussian}}$ |
|---|---|---|---|---|---|
| Transformer | 6 | 8 | 256 | 10 | 8 |
| SO(3)-Transformer | 3 | 4 | 128 | 10 | — |

**Hyperparameters.** We use a batch size of 128 and 2,000 epochs for the HFM models, resulting in a total training time of up to 22h on a single NVIDIA A100 80BG GPU. We use 640,000 epochs for the HFM model trained on 256 samples, amounting to 1d14h training time. In both cases, we set $\Delta t_{\max} = 10$fs. For the SO(3)-equivariant Transformer, we train for 500 epochs with a batch size of 64, corresponding to a total training time of up to 13h. We train the MLFF for 160,000 epochs with batch size 64, which amounts to 1d13h training time. Following (Batzner et al., 2022), we use as our training objective a weighted sum of energy and force loss terms, i.e., $\mathcal{L} = \lambda_E \mathcal{L}_E + \lambda_F \mathcal{L}_F$, and set $\lambda_E = 0.01$ and $\lambda_F = 0.99$.

**Simulation.** We perform 3 ns of MD simulation in the NVT ensemble using a Langevin thermostat. We set the target temperature to $T = 500$ K and apply a friction of $\gamma = 1/100$ fs in all experiments. We also perform 3 ns of MD simulation in the NVE ensemble. Similar to training, we start the simulation with zero net linear momentum, and apply inference-time filters during simulation: drift removal, random rotation and coupled energy and angular momentum correction. We use our own trained MLFFs as the potential for the energy correction.

**ScoreMD.** We benchmark against the recent diffusion model from Plainer et al. (2025), which learns a diffusion process on Boltzmann-distributed samples, can generate iid samples via the standard denoising diffusion process, and also enables force estimation for simulation. To ensure a fair comparison, we use the graph transformer architecture from their work. We train the model with 128 hidden units, 3 layers, and 16-dimensional feature embeddings. For the dataset with 256 samples, we use a batch size of 256; for the dataset with $\approx 85$k samples, we use a batch size of 1024. To stabilize force estimation, we evaluate the network at diffusion time $t = 0.05$. After training, we run Langevin simulations with the same setup as in our framework, using 25 parallel simulations to speed up sampling. We did not use any filters when evaluating ScoreMD.

### F.5. Alanine Dipeptide

**Dataset.** We use the dataset provided Köhler et al. (2021), which contains data from a 1 $\mu$s simulation in implicit solvent of which we use 80% for training, and 10% for validation. We sample momenta $\boldsymbol{p}$ from the Maxwell–Boltzmann distribution with mean $\mu_T = 500$ K and standard deviation $\sigma_T = 150$ K as described in Appendix B.

**Architecture.** We learn Hamiltonian Flow Maps (HFMs) using a translation-invariant Transformer architecture. Model hyperparameters are given in Table 6. In contrast to the other experiments, we do not train a separate MLFFs for the coupled energy and angular momentum conservation filter, but instead just add an additional output head to predict the energy.

*Table 6.* Architectural details for the translation-invariant transformer on alanine dipeptide. $B$ is the number of blocks, $n_{\text{heads}}$ is the number of attention heads, $H$ is the hidden dimension, and $M_{\text{RBF}}$ and $M_{\text{Gaussian}}$ are the number of radial basis and Gaussian basis functions, respectively.

| Model | $B$ | $n_{\text{heads}}$ | $H$ | $M_{\text{RBF}}$ | $M_{\text{Gaussian}}$ |
|---|---|---|---|---|---|
| Transformer | 6 | 8 | 256 | 10 | 8 |

**Hyperparameters.** We use a batch size of 512 and 2,000 epochs for the HFM models, resulting in a total training time of up to 3d12h on a single NVIDIA H100 80BG GPU, or 4d22h on an A100 80GB GPU. The energy loss weight is 0.01.

**Simulation.** We perform 10 parallel simulations of 100 ns of MD simulation in the NVT ensemble using a Langevin thermostat, totalling to 1 $\mu$s. We set the target temperature to $T = 300$ K and apply a friction of $\gamma = 1/1000$ fs in all experiments, which is the same as in the reference data (Köhler et al., 2021). Similar to training, we start the simulation with zero net linear momentum, and apply inference-time filters during simulation: drift removal, random rotation and coupled

energy and angular momentum correction. We use the same HFM model with a separate output head to also predict the potential energy for the energy correction filter.

## F.6. Ac-Ala3-NHMe

**Dataset.** We adopt the MD22 dataset (Chmiela et al., 2023) for Ac-Ala3-NHMe with 85,109 samples. We use 80% of the data ($\approx 68$k configurations) for training. We sample momenta $\boldsymbol{p}$ from the Maxwell–Boltzmann distribution with mean $\mu_T = 500\,\mathrm{K}$ and standard deviation $\sigma_T = 150\,\mathrm{K}$ as described in Appendix B.

**Architecture.** We learn Hamiltonian Flow Maps (HFMs) using a translation-invariant Transformer architecture. Additionally, we employ an $\mathrm{SO}(3)$-equivariant Transformer as an MLFF baseline. Hyperparameters are given in Table 7.

*Table 7.* Architectural details for the Translation-invariant and $\mathrm{SO}(3)$-equivariant Transformer on Ac-Ala3-NHMe. $B$ is the number of blocks, $n_{\text{heads}}$ is the number of attention heads, $H$ is the hidden dimension, and $M_{\text{RBF}}$ and $M_{\text{Gaussian}}$ are the number of radial basis and Gaussian basis functions, respectively. Since the $\mathrm{SO}(3)$-Transformer does not have a conditioning branch, we write '–' for $M_{\text{Gaussian}}$.

| Model | $B$ | $n_{\text{heads}}$ | $H$ | $M_{\text{RBF}}$ | $M_{\text{Gaussian}}$ |
|---|---|---|---|---|---|
| Transformer | 6 | 8 | 256 | 10 | 8 |
| $\mathrm{SO}(3)$-Transformer | 3 | 4 | 128 | 10 | — |

**Hyperparameters.** We use a batch size of 128 and 5,000 epochs for the HFM model, resulting in a total training time of up to 40h on a single NVIDIA H100 80BG GPU. We set $\Delta t_{\max} = 10$fs. For the $\mathrm{SO}(3)$-equivariant Transformer, we train for 500 epochs with a batch size of 64, corresponding to a total training time of up to 10h on a single NVIDIA H100 80BG GPU. Following (Batzner et al., 2022), we use as our training objective a weighted sum of energy and force loss terms, i.e., $\mathcal{L} = \lambda_E \mathcal{L}_E + \lambda_F \mathcal{L}_F$, and set $\lambda_E = 0.01$ and $\lambda_F = 0.99$.

**Simulation.** We perform 3 ns of MD simulation in the NVT ensemble using a Langevin thermostat. We set the target temperature to $T = 500\,\mathrm{K}$ and apply a friction of $\gamma = 1/100\,\mathrm{fs}$ in all experiments. Similar to training, we start the simulation with zero net linear momentum, and apply inference-time filters during simulation: drift removal, random rotation and coupled energy and angular momentum correction. We use our own trained MLFFs as the potential for the energy correction.

## F.7. Chignolin and BBA

**Dataset.** In this setup, we use the pre-trained coarse-grained generative force field ScoreMD (Plainer et al., 2025) and distill it. We take the data from Lindorff-Larsen et al. (2011) and use the same training split as in Plainer et al. (2025), reduce it to $C_\alpha$ atoms, and use ScoreMD to recompute coarse-grained force and energy labels for these training positions. As such, this allows us to train directly on coarse-grained samples where energy or forces are typically unavailable. Chignolin was simulated with a temperature of 340K and BBA with 325K.

**Architecture.** We learn Hamiltonian Flow Maps (HFMs) using a translation-invariant Transformer architecture. Model hyperparameters are given in Table 8. Similarly to the setup for alanine dipeptide, we do not train a separate MLFFs for the coupled energy and angular momentum conservation filter, but instead just add output head to predict the energy.

*Table 8.* Architectural details for the translation-invariant transformer and MLFF on Chignolin and BBA. $B$ is the number of blocks, $n_{\text{heads}}$ is the number of attention heads, $H$ is the hidden dimension, and $M_{\text{RBF}}$ and $M_{\text{Gaussian}}$ are the number of radial basis and Gaussian basis functions, respectively.

| System | Model | $B$ | $n_{\text{heads}}$ | $H$ | $M_{\text{RBF}}$ | $M_{\text{Gaussian}}$ | Cutoff in Å |
|---|---|---|---|---|---|---|---|
| Chignolin | Transformer | 6 | 8 | 256 | 10 | 8 | 15 |
| Chignolin | $\mathrm{SO}(3)$-Transformer | 3 | 4 | 128 | 10 | — | 15 |
| BBA | Transformer | 6 | 8 | 256 | 10 | 8 | 20 |
| BBA | $\mathrm{SO}(3)$-Transformer | 3 | 4 | 128 | 10 | — | 20 |

**Hyperparameters.** We use a batch size of 512 and 2,000 epochs for the Chignolin HFM models, resulting in a total training time of up to 1d on a single NVIDIA H100 80BG GPU. As for BBA, we use a batch size of 512 with 400 epochs and a training time of up to 1d4h on a single NVIDIA H100 80BG GPU.

## G. Additional Experiments and Ablations

### G.1. Analysis of Conservation Laws in the Microcanonical (NVE) Ensemble

Figure 15 provides the temporal evolution for total energy and angular momentum corresponding to the experiment in Figure 6. Our setup with filters maintains the conservation of both quantities over long simulation runs (500 ps).

Given identical starting conditions, our HFM model should theoretically trace the same phase-space trajectories as a reference MLFF integrated via Velocity Verlet, up to a certain time limit related to the divergence in chaotic systems. We compare HFM and Velocity Verlet (using our MLFF) trajectories starting from the same initial conditions for paracetamol in the NVE ensemble in Figure 15. Following Thiemann et al. (2026), we monitor selected interatomic distances and angles.

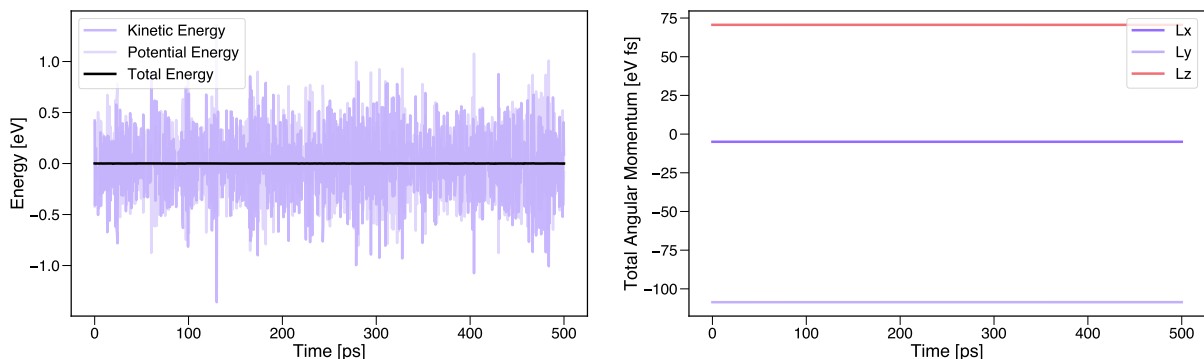

*Figure 14.* Conservation of total energy and total angular momentum in the NVE ensemble for paracetamol. **Left:** Temporal evolution of kinetic, potential and total energy over time. **Right:** Temporal evolution of angular momentum components over time.

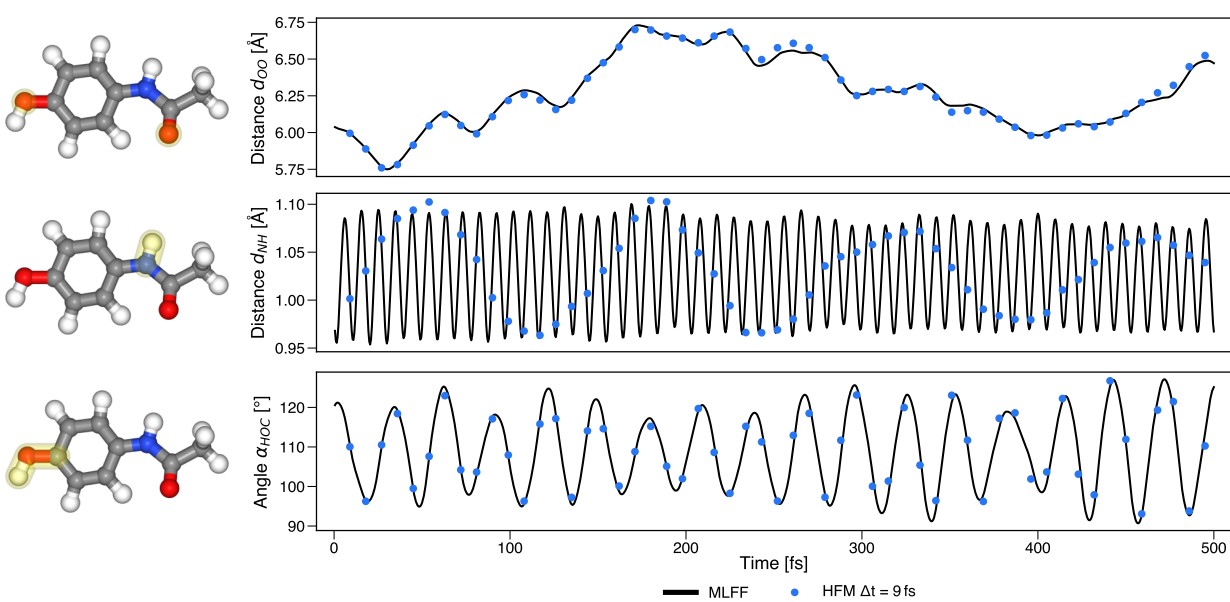

*Figure 15.* Short NVE simulations for paracetamol. We integrate our HFM model with $\Delta t = 9$ fs steps in an NVE setting, and compare to integration using Velocity Verlet. The figure is inspired by Thiemann et al. (2026). **First row:** oxygen-oxygen distance. **Second row:** nitrogen-hydrogen distance. **Third row:** hydroxyl group angle.

## G.2. Wall Clock Time Comparison for Simulation of Aspirin

In Table 9, we perform a wall clock time comparison between our model and MLFFs for aspirin. We measure the time for a single inference step, and also use this information to compute the total simulation length in ns for a single GPU day. We evaluate the speed of our own MLFF architecture, as well as two representative baselines from the literature: SchNet (Schütt et al., 2018), a very fast and lightweight invariant MLFF with limited expressivity, as well as NequIP (Batzner et al., 2022), a standard E(3)-equivariant MLFF. We re-implement both baseline MLFFs in JAX for a fair framework-independent runtime comparison. We use a single NVIDIA A100 80GB GPU for all timings, and report the average over 5000 measurements.

To ensure energy conservation, our inference pipeline includes an auxiliary MLFF call to compute potential energy corrections. We must consider the cost for an additional MLFF forward pass to compute the potential energy per simulation step. In this case, our own MLFF is used for the total energy correction, while using the HFM model to predict the energy would also be possible. The timings for HFM in Table 9 already account for this fact: The 2.1ms total inference time decomposes into 0.9ms for the MLFF forward pass (note that we don't need an additional backward pass to compute the conservative forces, since we are only interested in the potential energy) and 1.2ms for the translation-invariant Transformer that predicts mean forces and velocities. Consequently, even with this overhead, HFM-9fs achieves a simulation throughput of 378.5 ns/day, significantly outperforming classical integrators, even if an extremely lightweight model like SchNet with limited expressivity is employed.

*Table 9.* Time per forward pass (ms/step) and resulting simulation length (ns/day) for representative invariant (Schütt et al., 2018) and E(3)-equivariant (Batzner et al., 2022) MLFFs and for our HFM with increasing integration timestep. MLFFs use $\Delta t = 0.5$ fs. For fair comparison, all models are implemented in JAX. Timings are measured for aspirin on NVIDIA A100 80GB GPU.

| | SchNet | NequIP | Our MLFF | HFM-0.5fs | HFM-1fs | HFM-5fs | HFM-9fs |
|---|---|---|---|---|---|---|---|
| $\frac{\text{ms}}{\text{step}}$ | 0.3 | 2.1 | 1.4 | 2.1 | 2.1 | 2.1 | 2.1 |
| $\frac{\text{ns}}{\text{day}}$ | 148.9 | 20.6 | 31.7 | 21.0 | 42.1 | 210.3 | 378.5 |

## G.3. Analysis of Force Error for Small Molecules

In Table 10, we evaluate the instantaneous force prediction error of our model. It is important to note that HFM is trained to model integrated flow maps, rather than minimizing instantaneous force errors alone. Consequently, as the mixing parameter $q_{\tau=0}$ increases, the model prioritizes force matching over time consistency, resulting in a lower Force MAE. Compared to a dedicated MLFF trained using force matching alone, our models still achieve decent Force MAE. Following Geng et al. (2025a), we use $q_{\Delta t=0} = 0.75$ in all our experiments to strike balance between force matching and time consistency.

*Table 10.* Force MAE in units of [meV/Å] for different $q_{\Delta t=0}$. Sampling $\Delta t = 0$ more frequently yields better Force MAE. Setting $q_{\Delta t=0} = 0.75$ strikes a good balance between force matching and time consistency in our experiments.

| Method | Aspirin | Ethanol | Naphthalene | Salicylic Acid |
|---|---|---|---|---|
| MLFF | 3.8 | 1.5 | 1.5 | 2.2 |
| HFM ($\Delta t = 0$, $q_{\Delta t=0} = 0.75$) | 4.4 | 2.4 | 2.2 | 6.0 |
| HFM ($\Delta t = 0$, $q_{\Delta t=0} = 0.50$) | 4.5 | 3.0 | 2.8 | 6.3 |
| HFM ($\Delta t = 0$, $q_{\Delta t=0} = 0.25$) | 7.0 | 2.8 | 3.0 | 4.7 |

## G.4. Analysis of Structural Observables for Other Thermostats

To asses the structural accuracy of our simulatios, we follow Fu et al. (2023) and calculate the distribution $h(r)$ over interatomic distances $r$ and report the MAE w.r.t. *ab-initio* reference data. We investigate the impact of thermostat choice within the NVT ensemble.

We observe that a global and deterministic Nosé-Hoover thermostat (Martyna et al., 1996) can lead to simulation artifacts in our framework causing larger deviations in MAE of $h(r)$ histograms (Table 12). Similar effects have been observed in other long-timestep models (Bigi et al., 2025a) and are attributed to a failure of the thermostat to maintain kinetic energy equipartition across atom types. This is likely caused by non-conservative force predictions, as similar effects are observed using direct force prediction in MLFFs (Bigi et al., 2025b). Our results using the stochastic CSVR thermostat (Bussi et al., 2007) in Table 11 are comparable to those using the Langevin thermostat (Bussi & Parrinello, 2007) in Table 1.

Furthermore, during early experiments, we found that global momenta rescaling in global thermostats is generally less robust than local Langevin dynamics, as local stochasticity seems to be helpful for dampening model errors. Consequently, we avoid explicit energy rescaling in our filters when using global thermostats (CSVR or Nosé-Hoover in Tables 11 and 12), as both mechanisms interfere with one another and degrade stability. Instead, we use an alternative filter that only preserves angular momentum.

*Table 11.* Interatomic distances $h(r)$ MAE [unitless] for 300 ps of MD simulation in the NVT ensemble using a **CSVR thermostat** w.r.t. the reference data from *ab-initio* calculations. Results are averaged over 5 simulation runs with standard deviations shown in parentheses. Lower values indicate better performance.

| Dataset | MLFF | Hamiltonian Flow Map | | | | | |
|---|---|---|---|---|---|---|---|
| | 0.5 fs | 0.5 fs | 1 fs | 3 fs | 5 fs | 7 fs | 9 fs |
| Aspirin | 0.026 (0.003) | 0.099 (0.003) | 0.024 (0.003) | 0.050 (0.002) | 0.070 (0.003) | 0.070 (0.005) | 0.064 (0.010) |
| Ethanol | 0.074 (0.005) | 0.119 (0.003) | 0.057 (0.006) | 0.079 (0.002) | 0.113 (0.004) | 0.099 (0.005) | 0.125 (0.001) |
| Naphthalene | 0.043 (0.005) | 0.053 (0.002) | 0.037 (0.001) | 0.041 (0.004) | 0.040 (0.001) | 0.046 (0.001) | 0.051 (0.001) |
| Salicylic Acid | 0.038 (0.003) | 0.025 (0.001) | 0.021 (0.001) | 0.064 (0.005) | 0.073 (0.004) | 0.054 (0.006) | 0.059 (0.003) |

*Table 12.* Interatomic distances $h(r)$ MAE [unitless] for 300 ps of MD simulation in the NVT ensemble using a *Nosé-Hoover* **thermostat** w.r.t. the reference data from *ab-initio* calculations. Results are averaged over 5 simulation runs with standard deviations shown in parentheses. Lower values indicate better performance.

| Dataset | MLFF | Hamiltonian Flow Map | | | | | |
|---|---|---|---|---|---|---|---|
| | 0.5 fs | 0.5 fs | 1 fs | 3 fs | 5 fs | 7 fs | 9 fs |
| Aspirin | 0.020 (0.003) | 0.236 (0.054) | 0.221 (0.008) | 0.048 (0.006) | 0.088 (0.014) | 0.094 (0.023) | 0.272 (0.002) |
| Ethanol | 0.067 (0.003) | 0.716 (0.017) | 0.695 (0.045) | 0.057 (0.002) | 0.367 (0.064) | 0.246 (0.153) | 0.244 (0.035) |
| Naphthalene | 0.036 (0.004) | 0.085 (0.003) | 0.077 (0.002) | 0.176 (0.043) | 0.172 (0.053) | 0.121 (0.011) | 0.501 (0.006) |
| Salicylic Acid | 0.024 (0.002) | 0.159 (0.062) | 0.102 (0.087) | 0.210 (0.020) | 0.154 (0.067) | 0.058 (0.004) | 0.062 (0.003) |

## G.5. Analysis of Simulation Stability for Small Molecules

We further validate the robustness of HFM by assessing simulation stability within the NVT ensemble. We adopt the stability metric by Fu et al. (2023). We compare runs from five different initial conditions and three thermostats. Tables 13–15 report the mean trajectory duration before collapse for Langevin, CSVR, and Nosé-Hoover thermostats. A mean of 300 ps refers to no simulation collapse. The results confirm that HFM maintains high stability even for large timesteps.

*Table 13.* Simulation stability under the **Langevin thermostat** with a filter for explicit coupled conservation of total energy and angular momentum. We report the mean stable trajectory duration in ps (max 300). Values are presented as: Mean (Standard Deviation) [Number of collapsed runs].

| Dataset | MLFF | Hamiltonian Flow Map | | | | |
|---|---|---|---|---|---|---|
| | 0.5 fs | 1 fs | 3 fs | 5 fs | 7 fs | 9 fs |
| Aspirin | 300.0 (000.0) [0] | 300.0 (000.0) [0] | 300.0 (000.0) [0] | 300.0 (000.0) [0] | 300.0 (000.0) [0] | 300.0 (000.0) [0] |
| Ethanol | 300.0 (000.0) [0] | 300.0 (000.0) [0] | 300.0 (000.0) [0] | 300.0 (000.0) [0] | 300.0 (000.0) [0] | 300.0 (000.0) [0] |
| Naphthalene | 300.0 (000.0) [0] | 300.0 (000.0) [0] | 300.0 (000.0) [0] | 300.0 (000.0) [0] | 300.0 (000.0) [0] | 300.0 (000.0) [0] |
| Salicylic Acid | 300.0 (000.0) [0] | 300.0 (000.0) [0] | 300.0 (000.0) [0] | 254.1 (091.9) [1] | 200.5 (039.6) [5] | 196.7 (127.7) [2] |

*Table 14.* Simulation stability under the **CSVR thermostat** without explicit energy correction filter. We report the mean stable trajectory duration in ps (max 300). Values are presented as: Mean (Standard Deviation) [Number of collapsed runs].

| Dataset | MLFF | Hamiltonian Flow Map | | | | |
|---|---|---|---|---|---|---|
| | 0.5 fs | 1 fs | 3 fs | 5 fs | 7 fs | 9 fs |
| Aspirin | 300.0 (000.0) [0] | 300.0 (000.0) [0] | 300.0 (000.0) [0] | 300.0 (000.0) [0] | 300.0 (000.0) [0] | 300.0 (000.0) [0] |
| Ethanol | 300.0 (000.0) [0] | 300.0 (000.0) [0] | 300.0 (000.0) [0] | 300.0 (000.0) [0] | 300.0 (000.0) [0] | 300.0 (000.0) [0] |
| Naphthalene | 300.0 (000.0) [0] | 300.0 (000.0) [0] | 300.0 (000.0) [0] | 300.0 (000.0) [0] | 300.0 (000.0) [0] | 300.0 (000.0) [0] |
| Salicylic Acid | 300.0 (000.0) [0] | 240.4 (077.2) [2] | 300.0 (000.0) [0] | 300.0 (000.0) [0] | 260.1 (079.9) [1] | 221.3 (102.7) [2] |

*Table 15.* Simulation stability under the **Nosé-Hoover thermostat** without explicit energy correction filter. We report the mean stable trajectory duration in ps (max 300). Values are presented as: Mean (Standard Deviation) [Number of collapsed runs].

| Dataset | MLFF | Hamiltonian Flow Map | | | | |
|---|---|---|---|---|---|---|
| | 0.5 fs | 1 fs | 3 fs | 5 fs | 7 fs | 9 fs |
| Aspirin | 300.0 (000.0) [0] | 300.0 (000.0) [0] | 300.0 (000.0) [0] | 300.0 (000.0) [0] | 300.0 (000.0) [0] | 271.4 (057.3) [1] |
| Ethanol | 300.0 (000.0) [0] | 300.0 (000.0) [0] | 300.0 (000.0) [0] | 300.0 (000.0) [0] | 300.0 (000.0) [0] | 300.0 (000.0) [0] |
| Naphthalene | 300.0 (000.0) [0] | 294.1 (011.8) [1] | 300.0 (000.0) [0] | 300.0 (000.0) [0] | 300.0 (000.0) [0] | 300.0 (000.0) [0] |
| Salicylic Acid | 300.0 (000.0) [0] | 178.9 (105.4) [3] | 300.0 (000.0) [0] | 300.0 (000.0) [0] | 287.3 (025.3) [1] | 300.0 (000.0) [0] |

### G.6. Ablation for $\tau$ Sampling Distribution on Small Molecules

In this subsection, we ablate how different base distributions $q(\tau)$ for the timestep $\Delta t$ affect the integration accuracy on several small molecules. In this work, we experiment with three different choices of $q(\tau)$ (see Appendix B). Therefore, for each molecule, we train three models that only differ in $q(\tau)$, while keeping all other hyperparameters fixed.

To quantify the model's ability to recover the correct Hamiltonian flow, we compute the Root Mean Square Deviation (RMSD) between the final state $(\boldsymbol{x}_{\mathrm{HFM}}, \boldsymbol{p}_{\mathrm{HFM}})$ predicted by our HFM model in a single step and a reference state $(\boldsymbol{x}_{\mathrm{ref}}, \boldsymbol{p}_{\mathrm{ref}})$ obtained via integrating the baseline MLFF with Velocity Verlet over the same time interval $\Delta t$. To account for the inherent diffusivity of the system, where particles naturally drift further apart over longer timescales, we normalize this prediction error by the accumulated RMSD of the reference trajectory. The normalized metric for positions at timestep $t$ is defined as:

$$\text{Normalized RMSD}_{\boldsymbol{x}}(t) = \frac{\mathrm{RMSD}(\boldsymbol{x}_{\mathrm{HFM}}(t), \boldsymbol{x}_{\mathrm{ref}}(t))}{\sum_{k=0}^{N-1} \mathrm{RMSD}(\boldsymbol{x}_{\mathrm{ref}}(\frac{(k+1)t}{N}), \boldsymbol{x}_{\mathrm{ref}}(\frac{kt}{N}))}, \tag{88}$$

where the denominator approximates the path length of the ground truth trajectory. We employ the same formula for momenta as well. This normalization ensures that errors at larger $\Delta t$ are not inflated due to larger total displacements, providing a fair comparison of integral error across different timescales.

The results in Figure 16 indicate that our proposed mixture distribution allows for consistently low integration error within the interval $[0, \Delta t_{\max}]$ on all systems. In contrast, the uniform distribution has consistently low integration error on aspirin and naphthalene, but higher integration error on ethanol and salicylic acid. We argue that this is due to instabilities we have observed while using the uniform distribution during training, which underlines that the choice of $q(\tau)$ has a large impact on the structure of the loss landscape (Zhang et al., 2026). Finally, the logit-normal distribution yields low integration error for small $\Delta t$, but the integration error increases significantly as $\Delta t \to \Delta t_{\max}$, since these timesteps were rarely sampled during training. Hence, we use our mixture distribution in all our experiments.

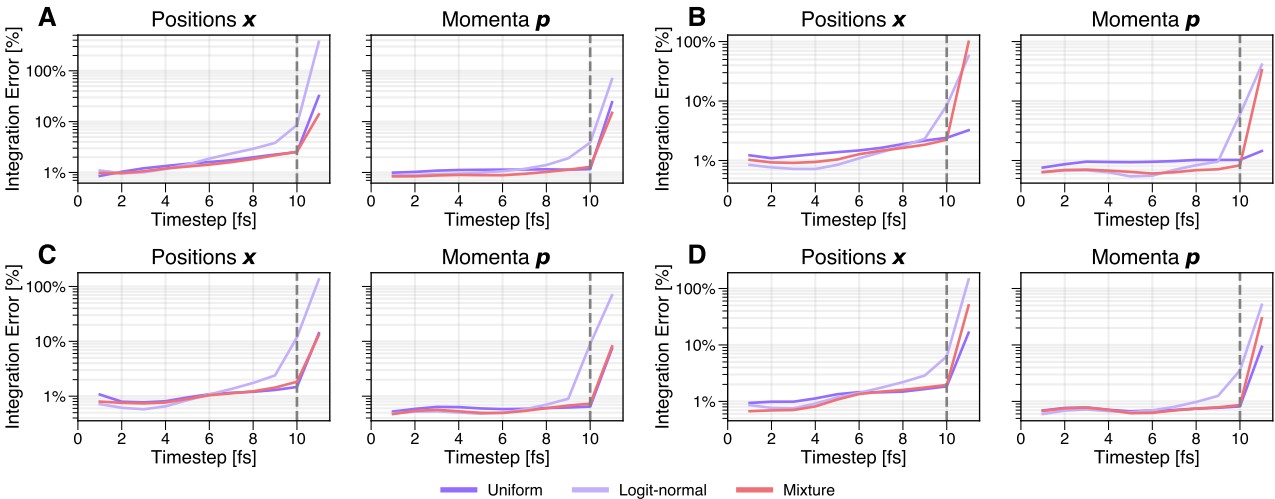

*Figure 16.* We show the normalized integration error (RMSD) for different sampling distributions $q(\tau)$ on small molecules. (**A**) Aspirin. (**B**) Ethanol. (**C**) Naphthalene. (**D**) Salicylic Acid. The dashed vertical line at $\Delta t_{\max} = 10\,\mathrm{fs}$ marks the training limit.

### G.7. Analysis of Training Dynamics on Paracetamol

In Figure 17, we analyze the convergence of the HFM model by monitoring the trajectory-level deviation from Velocity Verlet integration across different training checkpoints using the metric from Appendix G.6. We observe a systematic reduction in normalized integration error as training progresses, indicating that the model successfully refines its approximation of the underlying Hamiltonian flow. Interestingly, the integration error decreases for all timesteps over the course of the training, hinting at the fact that the model learns time consistency and force accuracy simultaneously and not separately.

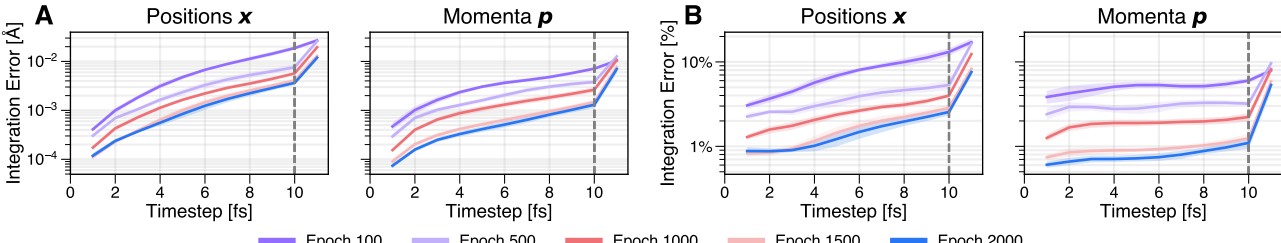

*Figure 17.* Evolution of integration error during training on paracetmol. (**A**) Absolute integration error. (**B**) Normalized integration error. The dashed vertical line at $\Delta t_{\max} = 10$ fs marks the training limit.

Furthermore, we investigate the impact of the training horizon $\Delta t_{\max}$ in Figure 18. The results highlight a trade-off between long-range stability and short-range precision: while a larger $\Delta t_{\max}$ enhances the model's predictive capability at large timesteps, it possibly leads to a degradation in accuracy for smaller timesteps (see Figure 18, right panel around 5fs).

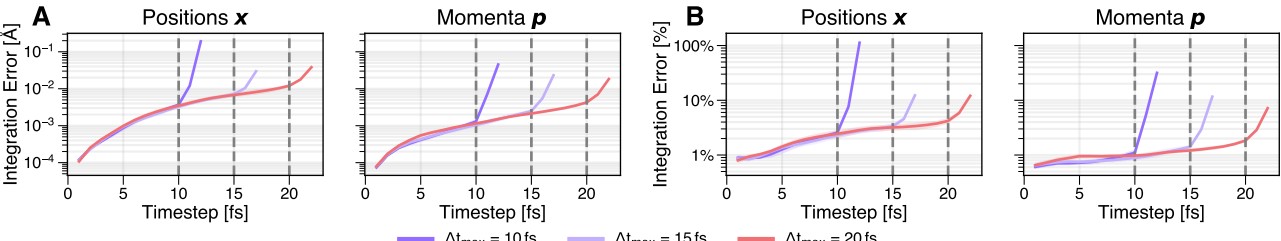

*Figure 18.* Ablation of maximum training timestep $\Delta t_{\max}$ on paracetamol. (**A**) Absolute integration error. (**B**) Normalized integration error. The dashed vertical line at $\Delta t_{\max} = 10$ fs marks the training limit.

### G.8. Analysis of Time Consistency on Paracetamol

In Figure 19, we evaluate the internal consistency of the learned Hamiltonian Flow Maps. A physically valid time-evolution operator must satisfy the semi-group property of the flow (Boffi et al., 2025a). In our notation, where $u_{\Delta t}$ denotes the flow map advancing the system state $z_t = (x_t, p_t)$ by time $\Delta t$, this condition implies that a single forward step should be equivalent to two sequential half-steps:

$$u_{\Delta t}(z_t) \approx u_{\Delta t/2}\big(u_{\Delta t/2}(z_t)\big). \tag{89}$$

To quantify violations of the semi-group property, we compute the consistency error as the RMSD between the state obtained from one full step $\Delta t$ and the state obtained from two composed steps of $\Delta t/2$. Analogous to the integration error analysis in Appendix G.6, we report both the absolute RMSD and the normalized RMSD, where the latter is scaled by the accumulated displacement of the composed trajectory to account for the natural scale of motion.

Figure 19 displays these metrics for an HFM trained with $\Delta t_{\max} = 10$ fs. As a reference, the dashed grey line shows the consistency error of a standard Velocity Verlet integrator based on our baseline MLFF, comparing a single 0.5 fs step against two 0.25 fs steps. The results demonstrate that our model maintains a comparable degree of internal consistency, up to around 4 fs steps. However, as shown in the other experiments, larger timesteps can still be used effectively in simulations.

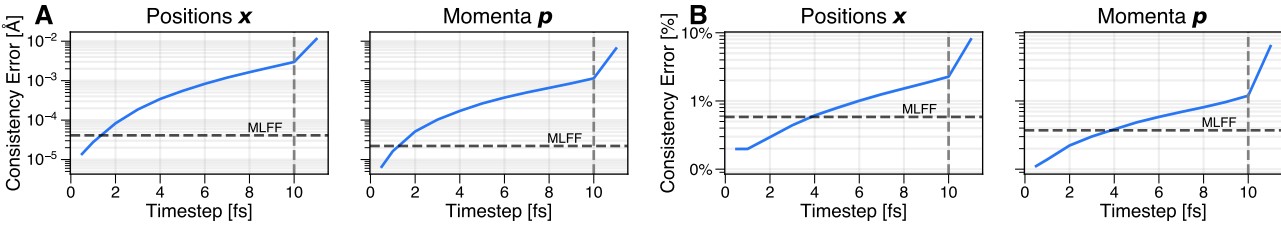

*Figure 19.* Analysis of time consistency on Paracetamol. We measure the deviation between a single predicted step of size $\Delta t$ and two composed steps of size $\Delta t/2$. (**A**) Absolute consistency RMSD error. (**B**) Normalized consistency RMSD error. The dashed vertical line at $\Delta t_{\max} = 10$ fs marks the training limit. The dashed horizontal line indicates the consistency error of Velocity Verlet with $0.5$ fs steps.

## G.9. Analysis of Vibrational Density of States on Paracetamol

In Figure 20, we perform a granular analysis of the vibrational density of states to assess the spectral fidelity of our HFM model for different training regimes. We employ the HFM trained with $\Delta t_{\max} = 10$ fs and evaluated in simulation for $\Delta t = 9$ fs consistent with Figure 7 in the main text. The plots display the per-atom power spectra, computed by taking the Fourier transform of the velocity autocorrelation function for each atom type in the paracetamol molecule. This allows us to disentangle the contributions of different chemical elements (C, H, N, O) to the vibrational dynamics.

We compare the spectra generated by HFM (blue) against those obtained from integrating a reference MLFF baseline with Velocity Verlet (black) for different training set sizes ($\approx 85k$ or $256$ training samples). The results demonstrate that our model recovers the correct frequency modes even in the data-scarce regime, where we train on $256$ decorrelated samples.

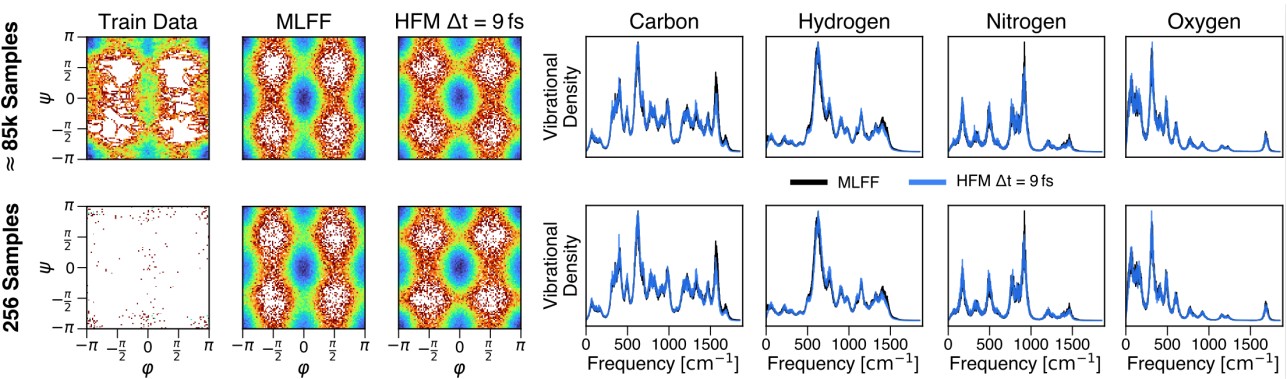

*Figure 20.* Data efficiency and spectral fidelity in NVT simulations ($3$ ns) for paracetamol. We compare the per-atom vibrational spectra of paracetamol generated by integrating the reference MLFF with Velocity Verlet (black) and our HFM model with $\Delta t = 9$ fs (blue). Rows correspond to different training set sizes.

## G.10. Analysis of Temperature Robustness on Paracetamol

In Figure 21, we test the temperature robustness of a single HFM model on paracetamol. Specifically, we report results for $T \in \{300, 350, 400, 450, 550, 600, 650, 700\}$ K, and compare the HFM model evaluated at $\Delta t = 9$ fs with our MLFF baseline. Across these temperatures, our HFM model closely matches the distribution of dihedral angles $(\varphi, \psi)$ and vibrational spectra obtained by the MLFF baseline. These results suggest high temperature robustness of our HFM model.

## G.11. Ablation: Effect of Inference Filters on Observables

Our inference pipeline employs projection filters to strictly enforce the conservation of total angular momentum and total energy. Here, we analyze the impact of these filters on thermodynamic consistency and dynamic as well as static observables. We further investigate whether using conservation filters might artificially stabilize invalid trajectories, e.g., those generated by Velocity Verlet integration with larger timesteps.

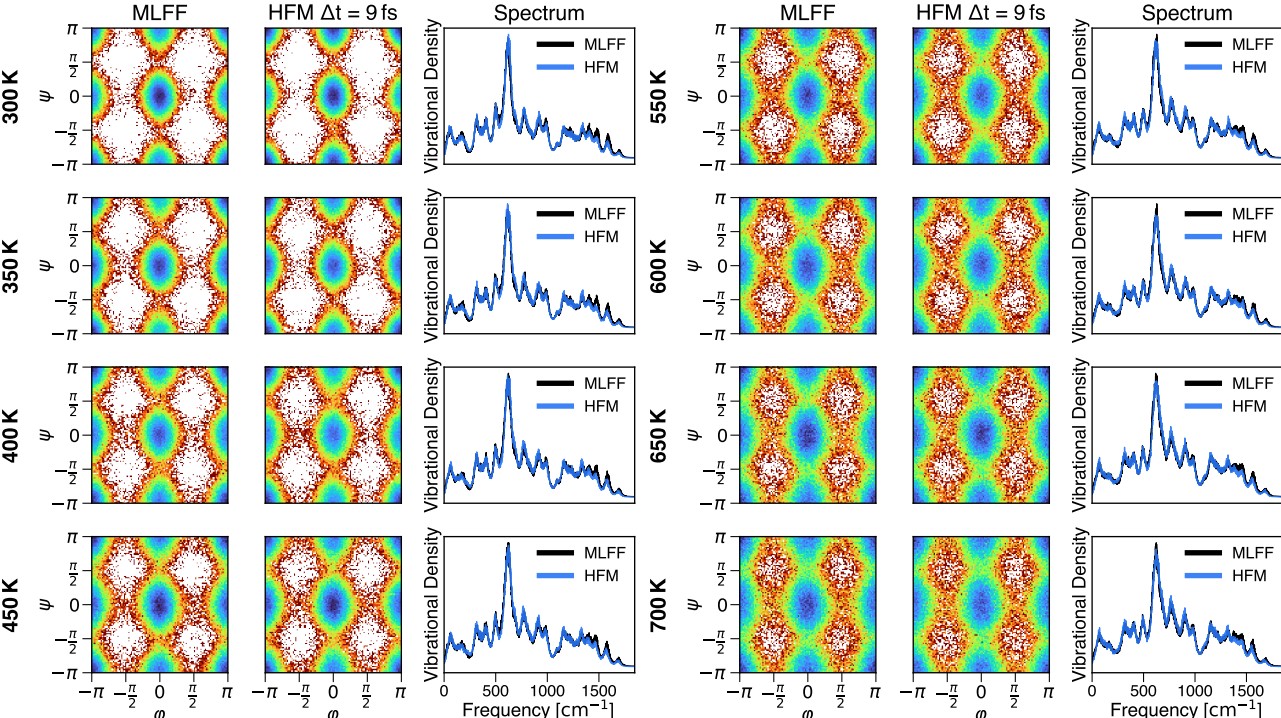

*Figure 21.* Temperature robustness in NVT simulations (3 ns) for paracetamol. We compare the ramachandran plot over dihedral angles $\varphi, \psi$ and vibrational spectra of paracetamol generated by integrating the reference MLFF with Velocity Verlet (black) and our HFM model with $\Delta t = 9$ fs (blue). We report results for a wide range of temperatures $T \in \{300, 350, 400, 450, 550, 600, 650, 700\}$ K.

First, we examine element-wise kinetic energy distributions, using the Langevin thermostat to simulate paracetamol in the NVT ensemble. In Table 16, we report the deviation of the per-element kinetic temperature from the target thermostat temperature. Without conservation filters (left), we observe significant deviations, particularly for hydrogen atoms at small timesteps, indicating a failure to maintain equipartition. Enabling the conservation filters (right) restores thermodynamic consistency, keeping temperature deviations minimal across all atom types and timesteps.

*Table 16.* Kinetic energy deviation ($\Delta T$) from the target thermostat temperature for paracetamol simulations using the Langevin thermostat. We compare integration without (left) and with (right) energy and angular momentum conservation filters. Values are reported as mean (standard deviation) in Kelvin. The filters effectively restore equipartition.

| Timestep [fs] | Without energy / angular momentum conservation | | | | | With energy / angular momentum conservation | | | | |
|---|---|---|---|---|---|---|---|---|---|---|
| | $\Delta T_{\text{all}}$ | $\Delta T_{\text{H}}$ | $\Delta T_{\text{C}}$ | $\Delta T_{\text{N}}$ | $\Delta T_{\text{O}}$ | $\Delta T_{\text{all}}$ | $\Delta T_{\text{H}}$ | $\Delta T_{\text{C}}$ | $\Delta T_{\text{N}}$ | $\Delta T_{\text{O}}$ |
| $\Delta t = 1$ | -13.7 (0.0) | -22.6 (0.0) | -6.1 (0.0) | -10.0 (0.0) | -6.0 (0.0) | -0.8 (0.0) | -8.8 (0.0) | 6.2 (0.0) | 3.2 (0.0) | 4.9 (0.0) |
| $\Delta t = 3$ | -2.1 (0.2) | -3.6 (0.2) | -0.7 (0.5) | -1.7 (1.0) | -1.1 (0.4) | -0.7 (0.1) | -2.3 (0.3) | 0.8 (0.3) | -0.5 (0.8) | 0.3 (0.4) |
| $\Delta t = 5$ | 1.9 (0.4) | 0.9 (0.6) | 3.6 (0.5) | 1.6 (1.5) | -0.4 (0.8) | -0.0 (0.4) | -0.8 (0.6) | 1.4 (0.5) | -0.2 (1.2) | -2.4 (1.2) |
| $\Delta t = 7$ | 1.3 (0.5) | -0.3 (0.7) | 2.9 (0.4) | 5.1 (0.8) | -0.3 (1.5) | 0.5 (0.3) | -1.2 (0.4) | 2.2 (0.4) | 3.7 (1.4) | -0.3 (0.6) |
| $\Delta t = 9$ | 2.6 (0.2) | 1.4 (0.6) | 4.0 (0.5) | 6.9 (0.8) | -0.2 (1.0) | 0.4 (0.4) | -1.0 (0.5) | 2.0 (0.5) | 4.4 (1.6) | -2.0 (0.9) |

Second, we assess whether the structural dynamics predicted by our model rely on these filters. Figure 22 displays the statistics and vibrational spectra generated without explicit energy or angular momentum conservation. The spectra remain well aligned with the ground truth MLFF results (see Figure 20 for comparison). This suggests that the HFM model learns valid dynamics intrinsic to the Hamiltonian flow, while the filters serve to correct thermodynamic drifts.

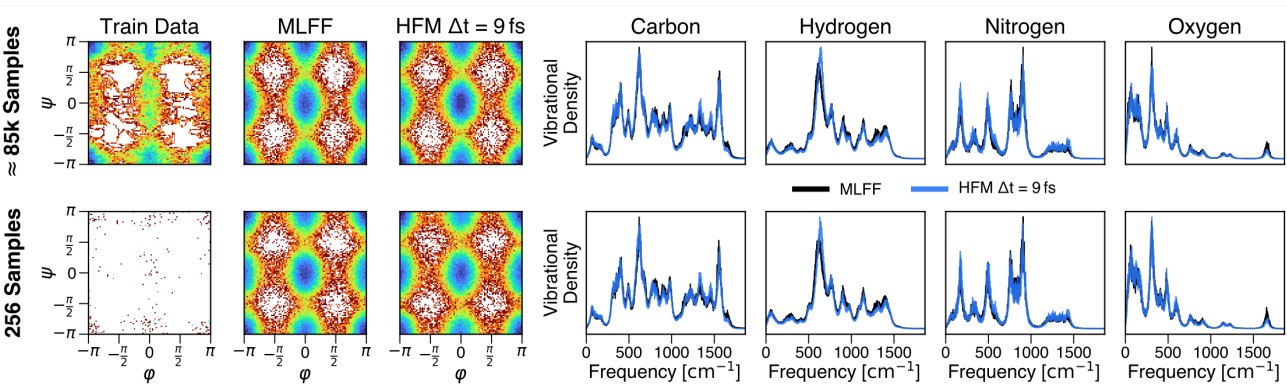

*Figure 22.* Robustness of vibrational spectra **without filters**. We show the per-atom vibrational density of states for paracetamol ($\Delta t = 9$ fs) generated **without** energy and angular momentum conservation filter. Comparing this to Figure 20 confirms that the structural dynamics are robustly captured by the model even in the absence of explicit constraints.

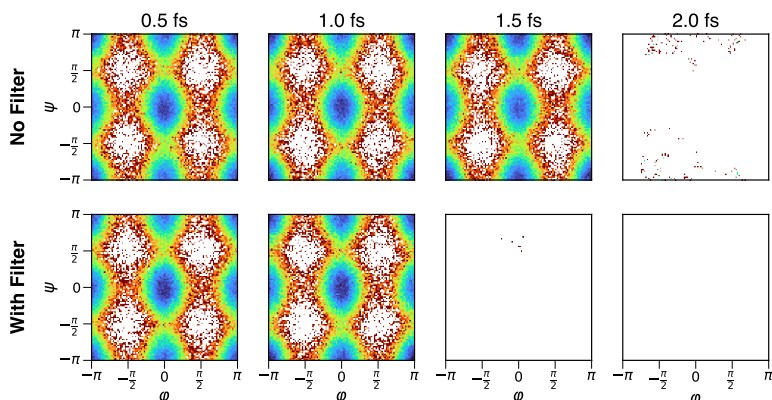

*Figure 23.* Effect of inference filters on Velocity Verlet stability: We attempt to stabilize integration of our MLFF using Velocity Verlet using inference filters. **Top**: The unfiltered baseline becomes unstable between timesteps of 1.5 and 2.0 fs. **Bottom**: Applying filters does not rescue the simulation but rather causes earlier divergence (for smaller timestep), confirming that filters do not in general artificially stabilize incorrect integration steps.

Finally, we address the question whether our filters might stabilize arbitrary invalid trajectories and thereby enable larger timesteps. To test this, we apply our filters to a standard Velocity Verlet integration of the reference MLFF with different larger timesteps. As shown in Figure 23, the filters do not enable larger timesteps for Velocity Verlet. In contrast, applying rigid energy constraints to the erroneous steps of Velocity Verlet causes the simulation to diverge earlier (instability onset $< 1.5$ fs) than for the unfiltered baseline. This confirms that the large-timestep stability of HFM stems from the learned flow map itself, not from the post-hoc application of inference filters.

### G.12. Ablation: Effect of Integration Timestep on Free Energy Surface for Alanine Dipeptide

**Free energy surfaces.** In this section, we complement the main paper and report additional Ramachandran plots for an HFM trained with our objective up to $\Delta t_{\max}$ and evaluated at different timesteps $\Delta t$. The corresponding plots are shown in Figure 24, with numerical results in Table 17. Overall, models evaluated up to $\Delta t = 13$ fs recover the low-probability mode and match the reference distribution well, except for $\Delta t \approx 10$ fs, where we observe a systematic deviation. Since this behavior is consistent across hyperparameters and training setups for this system, we attribute it to an inherent instability at this timescale. As $\Delta t$ approaches $\Delta t_{\max}$, the agreement degrades further: the simulations no longer recover all basins and, in some cases, evaluations at $\Delta t = \Delta t_{\max}$ diverge.

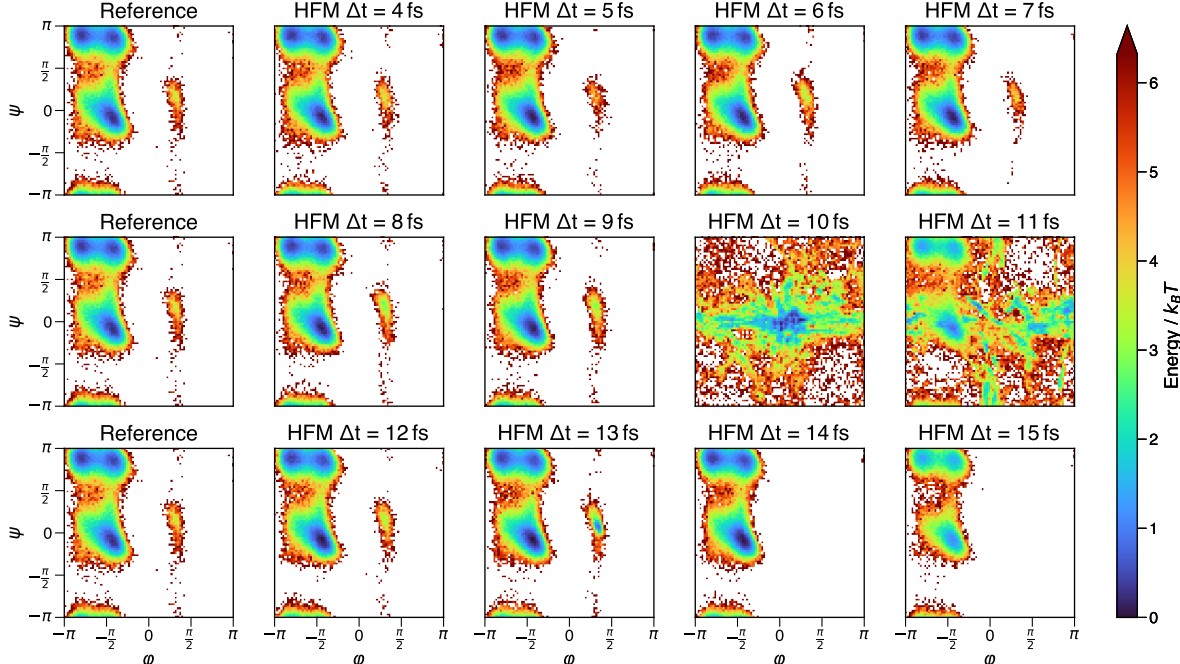

*Figure 24.* Comparing the free energy surfaces projected onto the dihedral angles $\varphi, \psi$ for an HFM evaluated at different step sizes $\Delta t$. Each simulation was initialized from 10 starting points, totaling $1\,\mu s$ of simulation time, and was downsampled to 100k samples for visualization.

To analyze the deviation of our HFM model for $\Delta t = 10\,\text{fs}$ and $\Delta t = 10\,\text{fs}$, we conduct additional reference simulations using the Amber ff99SB-ILDN force field in `OpenMM` (Eastman et al., 2017), which was also used to generate the ground truth data for training (Köhler et al., 2021). From those simulations, we compute the element-wise spectra to analyze vibrational modes (see Figure 25).

We find that the fast hydrogen-X vibrations have periods of $10\,\text{fs}$ and $11\,\text{fs}$, which coincide exactly with the integration timesteps where our HFM model becomes unstable in our experiments. We argue that as we hit the period of the hydrogen vibrations exactly with our integration timestep, it is much harder for the model to make an accurate prediction, since it needs to learn that large intra-step oscillations cancel out exactly for this timestep. Especially if our model makes a relatively small frequency error, i.e., it predicts a slightly larger or smaller time step, the relative error in phase space may accumulate quickly. For larger integration timesteps, this phenomenon vanishes and rollouts become stable again.

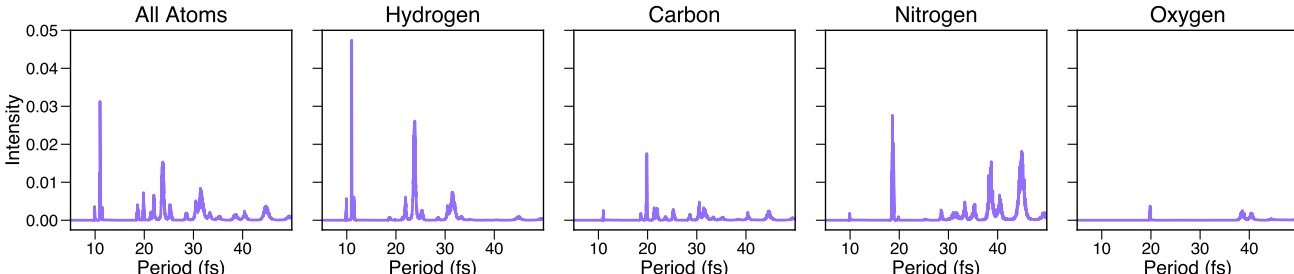

*Figure 25.* Vibrational frequencies for 1ns simulation of alanine dipeptide using the Amber ff99SB-ILDN force field in OpenMM.

**Accuracy vs. timestep for alanine dipeptide.** To quantify agreement between the predicted free energy surface and the reference distribution from Köhler et al. (2021), we report the mean Jensen–Shannon divergence (MJS) and the potential mean force (PMF) error following Durumeric et al. (2026) with the hyperparameters from (Plainer et al., 2025). Both metrics compare the 2D free energy surface projection, with the PMF error weighting low-probability regions more strongly due to the use of $\log$. In Table 17, we observe that the deviation from the reference increases with the step size $\Delta t$. For

$\Delta t \approx 10$ fs, simulations tend to become unstable and no longer yield meaningful results.

*Table 17.* Alanine dipeptide accuracy across different timesteps $\Delta t$ (fs). We report the PMF error and MJS. Lower values are better.

| Metric | 4 | 5 | 6 | 7 | 8 | 9 | 10 | 11 | 12 | 13 | 14 | 15 |
|---|---|---|---|---|---|---|---|---|---|---|---|---|
| PMF | 0.041 | 0.049 | 0.045 | 0.044 | 0.049 | 0.048 | 29.676 | 10.480 | 0.053 | 0.169 | 0.117 | 0.170 |
| MJS | 0.0039 | 0.0046 | 0.0042 | 0.0041 | 0.0048 | 0.0045 | 0.5380 | 0.2419 | 0.0051 | 0.0120 | 0.0067 | 0.0094 |

**3D conformations.** Figure 26 shows ten randomly selected snapshots from the alanine dipeptide trajectory generated by the HFM at $\Delta t = 12$ fs.

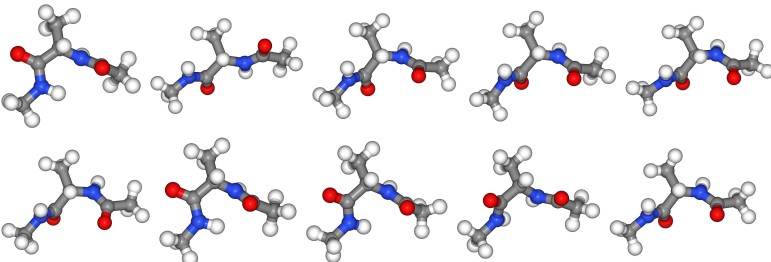

*Figure 26.* 3D renderings of ten randomly selected frames from the alanine dipeptide simulation using the HFM evaluated at $\Delta t = 12$ fs.

### G.13. Lyapunov Time Analysis

To evaluate whether the integrator preserves physical error accumulation without introducing spurious instabilities, we analyze the divergence of perturbed rollouts by measuring the system's characteristic error growth rate. A structurally stable integrator should match the intrinsic chaotic timescales of the underlying continuous physical system and not substantially exceed them. Following the methodology outlined in Bigi et al. (2025a) Appendix M, we quantify this behavior by calculating the empirical Lyapunov times.

As summarized in Table 18, the Lyapunov times obtained via HFM closely match the reference molecular dynamics (MD) baseline across the evaluated integration step sizes. This demonstrates that HFM accurately reproduces the true chaotic dynamics and error propagation of the system.

*Table 18.* Empirical Lyapunov times for Ethanol. We report the characteristic error growth rate in fs across different integration step sizes.

| Dataset | MLFF | Hamiltonian Flow Map | | | |
|---|---|---|---|---|---|
| | 0.5 fs | 0.5 fs | 1 fs | 2 fs | 5 fs |
| Ethanol | 199 | 186 | 186 | 192 | 211 |

### G.14. Comparison to Learning from Trajectory Data

To assess the *data efficiency* of our approach, we extend our evaluation by comparing our trajectory-free method against trajectory-based training under a fixed computational budget for generating training data. In molecular modeling, representative geometries are often relatively cheap to obtain (e.g., via semi-empirical methods), making expensive ground-truth force evaluations the primary bottleneck. While our approach uses these forces directly as training inputs for isolated states, trajectory-based models require explicit simulations, which demands iteratively computing forces and next states. We conduct experiments for MD17 Paracetamol (Chmiela et al., 2017) and MD22 Ac-Ala3-NHMe (Chmiela et al., 2023).

Following the setup of TrajCast (Thiemann et al., 2026), we generate baseline trajectories with 0.5fs steps until the force evaluation budget is exhausted by initializing NVE runs from 5 randomly subsampled geometries of the MD17/MD22 datasets. We use our pre-trained MLFF as proxy for computing ground truth forces. We extract all pairs of states with a temporal separation of exactly 7 fs. As shown in Table 19 and Table 20, our trajectory-free consistency objective consistently achieves lower errors for the same force labeling costs. This enhanced data efficiency is particularly pronounced in the low-data regime (Figure 27 and Figure 28). Even if trained with a smaller budget of ground truth force evaluations, our model remains significantly closer to the true physical distribution during sampling rollouts.

*Table 19.* Data efficiency comparison of our training objective (HFM) and the Trajectory Matching (TM) objective (Thiemann et al., 2026; Bigi et al., 2025a) across different training data budgets for Paracetamol in NVT simulations (1 ns, 10 replicas, 7fs steps). Training data budgets are measured in terms of the number of force evaluations for creating them. We compute the potential mean force (PMF) error and mean Jensen-Shannon (MJS) divergence to the free energy surface produced with an accurate small-stepsize MLFF.

| Force Evaluations | PMF (HFM) ↓ | MJS (HFM) ↓ | PMF (TM) ↓ | MJS (TM) ↓ |
|---|---|---|---|---|
| 128 | 0.136 | 0.0132 | ⚡ | ⚡ |
| 256 | 0.129 | 0.0121 | 5.772 | 0.2696 |
| 1024 | 0.127 | 0.0120 | 0.326 | 0.0312 |
| 85k | 0.124 | 0.0119 | 0.122 | 0.0115 |

*Table 20.* Data efficiency comparison of our training objective (HFM) and the Trajectory Matching (TM) objective (Thiemann et al., 2026; Bigi et al., 2025a) across different training data budgets for Ac-Ala3-NHMe in NVT simulations (3 ns, 10 replicas, 7fs steps). Training data budgets are measured in terms of the number of force evaluations for creating them. We compute the potential mean force (PMF) error and mean Jensen-Shannon (MJS) divergence to the free energy surface produced with an accurate small-stepsize MLFF.

| Training Data | PMF (HFM) ↓ | MJS (HFM) ↓ | PMF (TM) ↓ | MJS (TM) ↓ |
|---|---|---|---|---|
| 2048 | 0.258 | 0.0246 | 1.484 | 0.1054 |
| 85k | 0.184 | 0.0159 | 0.523 | 0.0496 |
| 256k | N/A | N/A | 0.185 | 0.0176 |

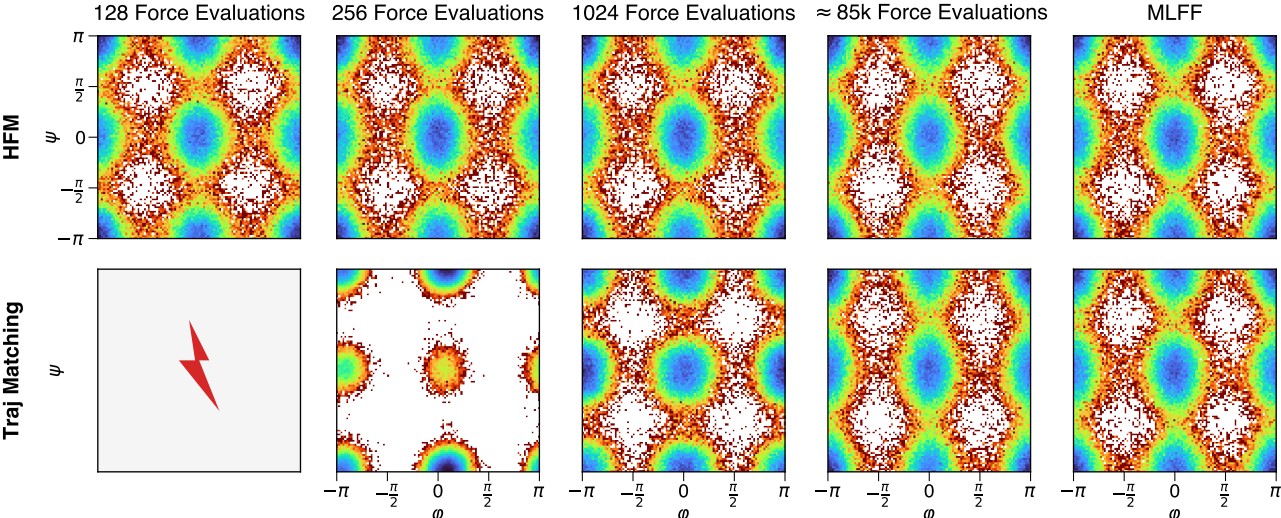

*Figure 27.* Ramachandran plots for Paracetamol in NVT simulations (1 ns, 10 replicas, 7fs steps) comparing our training objective (HFM) and the Trajectory Matching (TM) objective (Thiemann et al., 2026; Bigi et al., 2025a) across different training data budgets in terms of the number of force evaluations.

## G.15. Analysis of Transition Probabilities for Chignolin

For proteins, transition rates between metastable states are a critical time-dependent property. On Chignolin, we evaluate this by estimating the transition probabilities with a Markov chain following the definition in Plainer et al. (2025). As shown in Table 21, HFM obtains very low mean Jensen-Shannon divergence (JSD) values compared to the MLFF reference simulation. Even at a large timestep of $\Delta t = 25$ fs, we achieve a JSD of $8.11 \cdot 10^{-4}$, indicating accurate recovery of metastable transitions (see Figure 29). To test the limits of our model, we include results up to $\Delta t = 45$ fs.

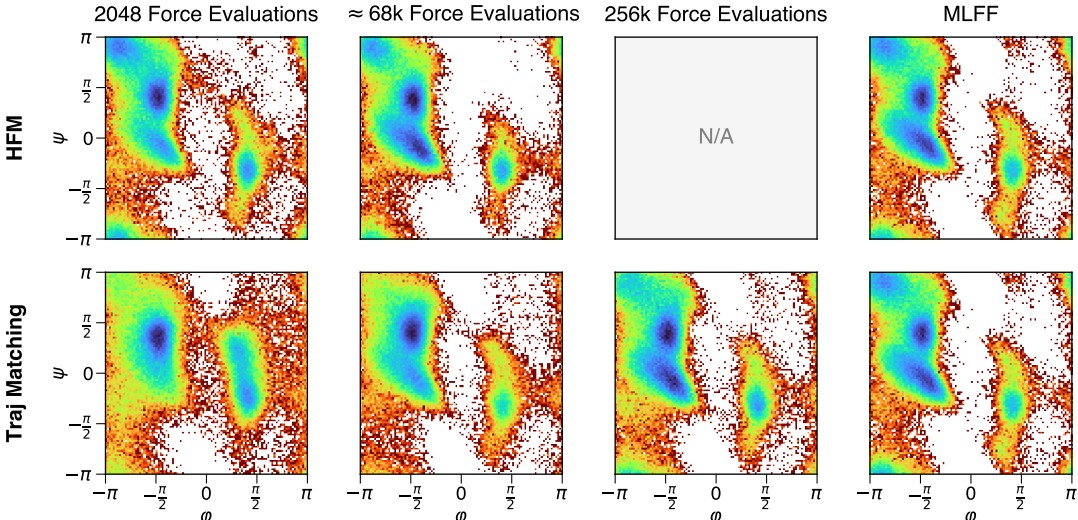

*Figure 28.* Ramachandran plots for Ac-Ala3-NHMe in NVT simulations (3 ns, 10 replicas, 7fs steps) comparing our training objective (HFM) and the Trajectory Matching (TM) objective (Thiemann et al., 2026; Bigi et al., 2025a) across different training data budgets in terms of the number of force evaluations.

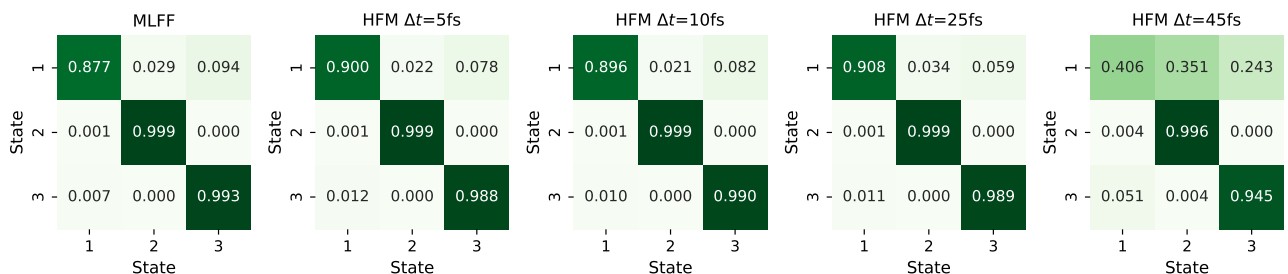

*Figure 29.* Comparison of transition probabilities for Chignolin. We assigned each 3D conformation to a single state and then estimated the transition probabilities with a Markov chain.

*Table 21.* Computing the mean Jensen-Shannon divergence (JSD) of transition rates of Chignolin comparing HFM to an MLFF reference simulation. We have assigned each 3D conformation of chignolin to one metastable state and estimate the transition probabilities with a Markov chain. We then compute the mean JSD of these transition probabilities.

| Method | Mean JSD |
|---|---|
| HFM $\Delta t = 5\,\text{fs}$ | $3.63 \cdot 10^{-4}$ |
| HFM $\Delta t = 10\,\text{fs}$ | $2.35 \cdot 10^{-4}$ |
| HFM $\Delta t = 25\,\text{fs}$ | $8.11 \cdot 10^{-4}$ |
| HFM $\Delta t = 45\,\text{fs}$ | $5.11 \cdot 10^{-2}$ |

### G.16. Detailed Analysis of Rollouts for BBA

We analyze the individual rollouts (per replica) for HFM and the CG MLFF to better understand the difference in sampling of the modes. Due to the larger time-steps, the HFM generates fewer samples per rollout but explores the distribution well (Figure 31). The CG MLFF seems less stable and gets stuck in some cases which leads to visible artifacts (Figure 30).

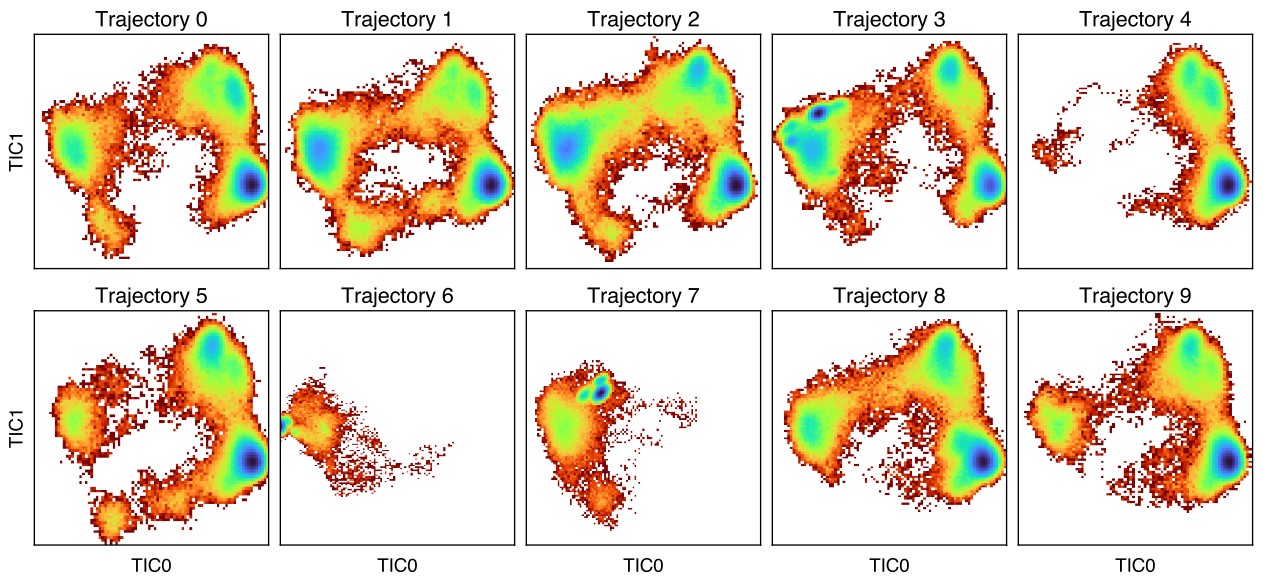

*Figure 30.* BBA rollouts using CG MLFF with 10 replicas.

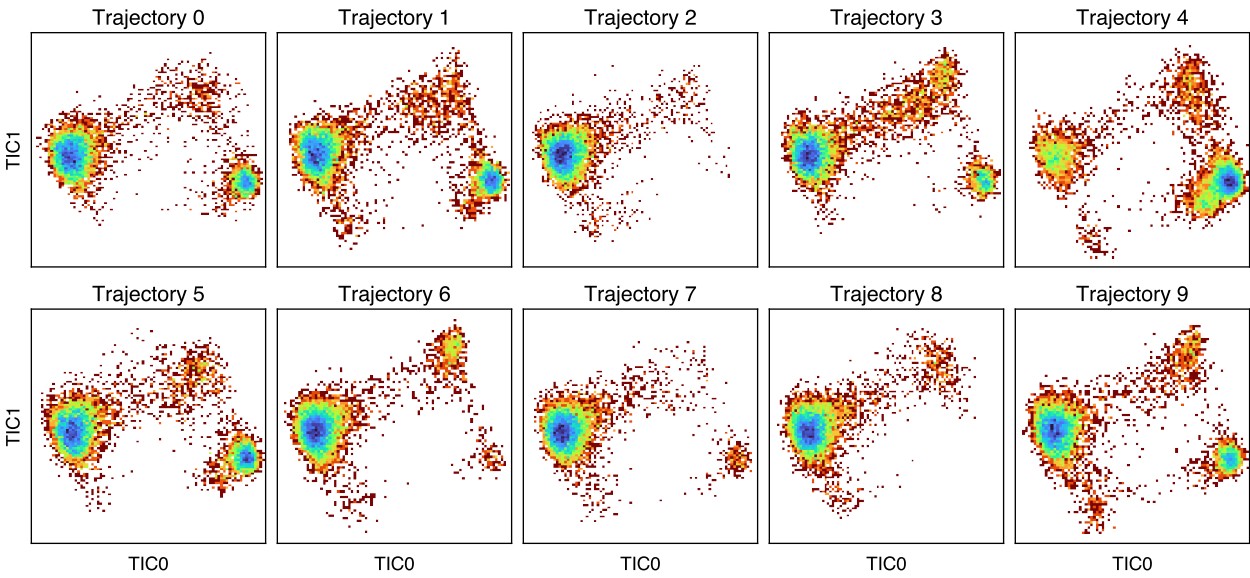

*Figure 31.* BBA rollouts using HFM with 10 replicas.

### G.17. 3D Conformations of BBA

In Figure 32, we verify the physical plausibility of the states our HFM model produces by plotting 3D renderings of random states of the BBA simulation. We can see that no atoms clash and the correct secondary structure is recovered.

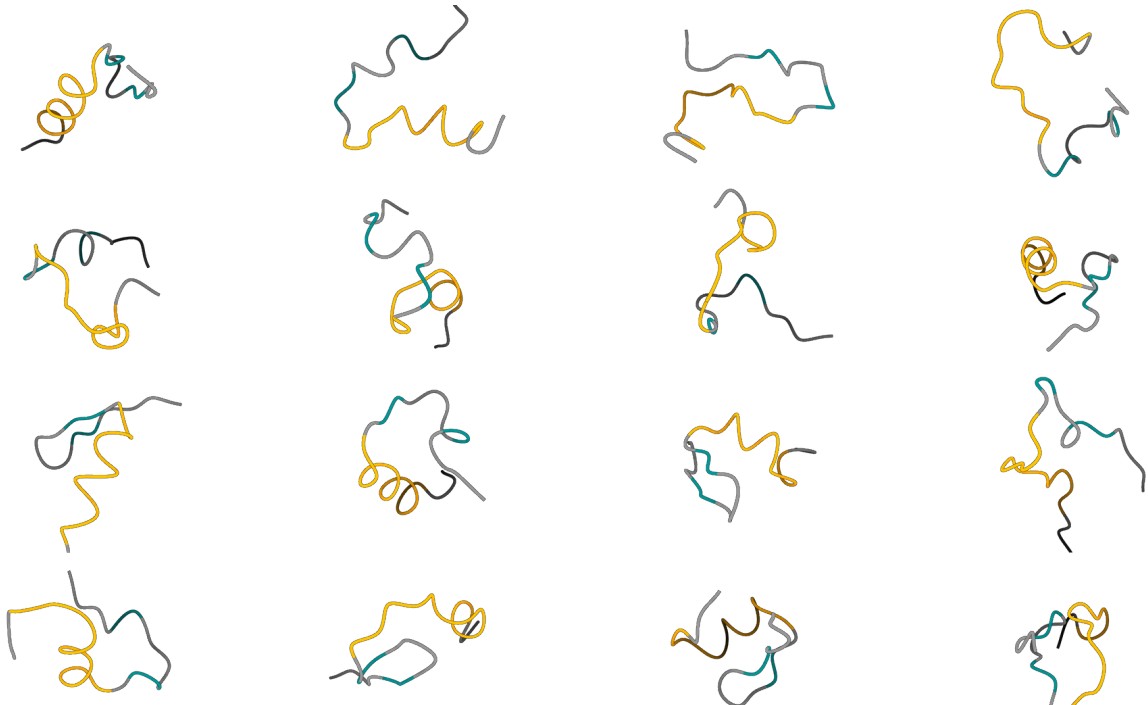

*Figure 32.* 3D renderings of 16 randomly selected frames from the BBA simulation using the HFM evaluated at $\Delta t = 15\,\text{fs}$.

