# OpenReview forum: "Learning Hamiltonian Flow Maps: Mean Flow Consistency for Large-Timestep Molecular Dynamics"
_ICML.cc/2026/Conference — ICML 2026 spotlight_

### Official Review · Reviewer_BFPw · 2026-03-07

**Soundness:** 2
**Presentation:** 3
**Significance:** 2
**Originality:** 2
**Overall Recommendation:** 4
**Confidence:** 4

**Summary:**

The paper proposes accelerating Hamiltonian dynamics by learning large-timestep updates using Hamiltonian flow maps directly from instantaneous phase-space information (positions, momenta, forces) without requiring trajectory data. instead of relying on classical numerical integration such as VV integrator. They introduces a mean-flow consistency objective that allows training using independent phase-space samples while enforcing consistency across different time horizons.

**Compliance With Llm Reviewing Policy:**

Affirmed.

**Final Justification:**

Acceptance as authors have addressed all my concerns during rebuttal.

**Key Questions For Authors:**

1) How sensitive is the method to the distribution used to sample momenta during training, since many datasets do not contain momentum labels?

2) How does the approach behave in strongly chaotic systems where small errors grow exponentially?

3) The model uses several inference filters; how much performance drop happens if these filters are removed?

4) It would be better if the author could compare the method with some more ML baselines like FlashMD, MDGen etc., as most of the methodology part is already known in the literature.

5) How does it scale to a very large system or more chaotic dynamics?

* Bigi, F., Chong, S., Kristiadi, A., & Ceriotti, M. (2025). FlashMD: long-stride, universal prediction of molecular dynamics. arXiv preprint arXiv:2505.19350.

* Jing, B., Stärk, H., Jaakkola, T., & Berger, B. (2024). Generative modeling of molecular dynamics trajectories. Advances in Neural Information Processing Systems, 37, 40534-40564.

**Limitations:**

Yes

**Strengths And Weaknesses:**

**Strength:**

1) new idea to learn large-timestep Hamiltonian flow maps directly from instantaneous supervision, avoiding trajectory generation which is expensive.

2) experimental results show improved stability at larger timesteps compared to Velocity Verlet integration which is practical benefit and scientific application.

**Weakness:**

1) theoretical guarantees about preservation of Hamiltonian structure (symplecticity, invariants) seem limited and mostly approximated using filters.

2) scalability to very large molecular systems or highly chaotic dynamics is not clearly shown.

3) proposed method seems to have limited novelty.

---

> ### Author Rebuttal · Authors · 2026-03-30
>
> We thank the reviewer and address all concerns below.
>
> > 1. How sensitive is the method to the distribution used to sample momenta during training, since many datasets do not contain momentum labels?
>
> To **train on existing MLFF datasets without momentum labels**, we sample momenta from the Maxwell-Boltzmann distribution. As seen in table R1 (and table S3, figure S4 in [supplement](https://drive.google.com/file/d/1snAAlZHVIedi2HRBTooHFoEdPpR2cRAV)), a model trained at 500K is highly robust and generalizes well across temperatures.
>
> *Table R1. Momenta Generalization on Paracetamol: We train our model at 500K, run inference at different temperatures (300-700K), and compare the free energy surfaces with an MLFF with PMF (Potential of Mean Force) errors and MJS (Mean Jensen-Shannon Divergence)*
>
> |Temp|PMF↓|MJS↓|
> |---|---|---|
> |300K|0.117|0.0104|
> |400K|0.120|0.0110|
> |500K|0.118|0.0111|
> |600K|0.138|0.0137|
> |700K|0.102|0.0108|
>
> > 2. How does the approach behave in strongly chaotic systems where small errors grow exponentially?
>
> > 5. How does it scale to a very large system or more chaotic dynamics?
>
> The considered molecular systems are already strongly chaotic, limiting all deterministic MD methods. Our approach extends the time horizon before divergence compared to classical integrators (App. G.1).
>
> Scalability is primarily determined by the underlying architecture rather than the training objective. We confirm robustness on **additional larger systems** including CG proteins (MD22, Chignolin, BBA), maintaining stable simulations and correct ensemble statistics at large timesteps. Notably, Chignolin in table R2 remains stable and accurate at a stepsize >20x larger than a normal MLFF. See figures S1,S6,S7 and tables S1,S7,S8 in our [supplement](https://drive.google.com/file/d/1snAAlZHVIedi2HRBTooHFoEdPpR2cRAV) for other systems.
>
> *Table R2. PMF and MJS errors over TICA components for Chignolin comparing HFM to an MLFF reference simulation*
>
> |Δt|PMF↓|MJS↓|
> |---|---|---|
> |15fs|0.166|0.0175|
> |25fs|0.122|0.0131|
> |35fs|0.073|0.0079|
> |45fs|0.314|0.0306|
>
> > 3. The model uses several inference filters; how much performance drop happens if these filters are removed?
>
> > theoretical guarantees seem limited and mostly approximated using filters
>
> We show stable NVT rollouts without filters in Table R3, though kinetic energy can deviate (Table 14, App. G.10). Any unconstrained approximation of the finite-time Hamiltonian flow breaks exact symplecticity. Strictly symplectic models exist but are not yet scalable [2]. Compare to last answer for reviewer oQ6j.
>
> *Table R3. Interatomic distances $h(r)$ MAE [unitless] extending Table 1, for HFM simulations without conservation filters.*
> |Dataset|MLFF 0.5fs|HFM 1fs|HFM 5fs|HFM 9fs|
> |---|---|---|---|---|
> |Aspirin|0.030|0.027|0.040|0.046|
> |Ethanol|0.073|0.076|0.094|0.112|
> |Naphthalene|0.043|0.044|0.050|0.060|
> |SalicylicAcid|0.035|0.037|0.051|0.057|
>
> > 4. It would be better if the author could compare the method with some more ML baselines
>
> MDGen targets a different regime relying on generative modeling to match distributions across very large jumps in time, consequently not recovering dynamic observables.
>
> We extend our evaluation with **additional baselines** and generate independent NVE training trajectories using a reference MLFF under a fixed budget of ground truth force evaluations following the setup of [1,3]. For each budget, we train a model using our consistency loss or **regression on trajectories**. We simulate under identical conditions (Table R4) finding that our method is much more data efficient. See figures S1,S3 and tables S1,S2 in our [supplement](https://drive.google.com/file/d/1snAAlZHVIedi2HRBTooHFoEdPpR2cRAV) for results on MD22.
>
> *Table R4. Comparison of HFM vs. trajectory matching [1,3] under the same budget of force evaluations for generating the training data. We state PMF and MJS errors over dihedral angles for Paracetamol compared to a MLFF reference. Consistency training (HFM) is more data efficient than trajectory matching.*
> |# Force Evaluations|HFM PMF↓|HFM MJS↓|Traj PMF↓|Traj MJS↓|
> |---|---|---|---|---|
> |128|0.136|0.0132|⚡|⚡|
> |256|0.129|0.0121|5.772|0.2696|
> |1024|0.127|0.0120|0.326|0.0312|
> |85k|0.124|0.0119|0.122|0.0115|
> > proposed method seems to have limited novelty
>
> We respectfully disagree:
> 1. We propose the first self-consistency objective for physical systems
> 2. We efficiently train a deterministic MD propagator that unifies multiple time strides into a single model; a task previously established highly challenging [1]
> 3. We are the first to train a large-timestep MD model entirely using existing MLFF datasets
> ---
> [1] Bigi et al. "FlashMD: long-stride, universal prediction of molecular dynamics" NeurIPS 2025
>
> [2] Bigi et al. "Learning the action for long-time-step simulations of molecular dynamics" arXiv:2508.01068 (2025)
>
> [3] Thiemann et al. "Force-free molecular dynamics through autoregressive equivariant networks" arXiv:2503.23794 (2025)

---

> > ### Author Rebuttal · Reviewer_BFPw · 2026-04-02
> >
> > Thanks for the rebuttal. I still have a few concise follow-up questions:
> >
> > > Momentum sampling
> >
> > How robust is the method if the training momentum distribution is misspecified or differs from test-time conditions?
> >
> > > Chaotic dynamics
> >
> > Can you quantify improvement in error growth (e.g., Lyapunov time) and preservation of time-dependent statistics?
> >
> > > Scalability
> >
> > What are the main bottlenecks when scaling to much larger systems (10×–100× atoms)?
> >
> > > Novelty
> >
> > How does your consistency objective fundamentally differ from prior multi-step or consistency-based training methods? Also, can you compare relevant work? Consider comparing against the following work as well:
> >
> > [1] JAMUN: Bridging Smoothed Molecular Dynamics and Score-Based Learning for Conformational Ensembles
> >
> > [2] Flow Matching for Accelerated Simulation of Atomic Transport in Crystalline Materials
> >
> >
> > > Failure cases
> >
> > In what scenarios does the method break or produce unphysical behavior?

---

> > > ### Author Response · Authors · 2026-04-04
> > >
> > > We would like to thank the reviewer for their detailed answer and for taking the time to engage with our previous response.
> > > > How robust is the method if the training momentum distribution [...] differs from test-time conditions?
> > >
> > > We agree that this is an important factor. The distribution is not a modeling choice but the Maxwell-Boltzmann distribution is the **correct equilibrium for a given temperature** and alternative distributions do not reflect meaningful variations.
> > >
> > > We address this concern by testing on unseen temperatures, which induce different (but still physically valid) momenta. To simulate severe test-time misspecification, we train at 300K and evaluate at significantly higher 700K, where our model remains stable and accurate.
> > >
> > > *Table R5. Generalization of HFM trained at 300K*
> > >
> > > |Temp|PMF↓|MJS↓|
> > > |---|---|---|
> > > |600 K|0.121|0.0120|
> > > |700 K|0.119|0.0124|
> > >
> > > > Can you quantify improvement in error growth (e.g., Lyapunov time)
> > >
> > > A stable integrator **matches the intrinsic Lyapunov time** of the dynamics and does not substantially exceed it. Following [1, App. M], HFM closely matches the ground-truth Lyapunov time for all timesteps (Table R6). This indicates that HFM preserves the correct error growth rate and does not introduce spurious instability.
> > >
> > > *Table R6. Lyapunov times for Ethanol*
> > > |MD 0.5fs|HFM 0.5fs|HFM 1.0fs|HFM 2.0fs|HFM 5fs|
> > > |---|---|---|---|---|
> > > |199|186|186|192|211|
> > >
> > > We believe that this strengthens our evaluation and will expand on it in the final revision.
> > >
> > > > preservation of time-dependent statistics?
> > >
> > > We analyze time-dependent dynamics via velocity autocorrelation in vibrational spectra (Fig 7,19,20), where HFM closely matches the reference simulation. For proteins, transition rates between metastable states are a critical time-dependent property. On Chignolin (25fs), we obtain a JSD of $8.1 \cdot 10^{-4}$, indicating accurate recovery of metastable transitions. See Fig S1,2 in [supplement 2](https://drive.google.com/file/d/1CjRzU66QCO23ujrc6xrcHks34N7dBT9b).
> > >
> > > > [...] bottlenecks when scaling to 10×–100× atoms?
> > >
> > >
> > > Our new results on large systems (MD22) show that our loss scales well to higher dimensions. The method itself adds on MLFFs and relies on a similar architecture where we naturally inherit the field’s bottlenecks for 10x-100x larger systems: 1) Local cutoffs allow for linear scaling in size but they do not capture long-range interactions needed for large systems 2) Generating ground-truth data becomes prohibitively expensive.
> > >
> > > These are challenges for the entire field, where we will benefit from ongoing efforts [4] and are compatible with existing strategies to bypass these limits. Our new results on CG proteins (BBA >500 atoms) demonstrate stable dynamics when combined with standard CG techniques.
> > >
> > > > Related work
> > >
> > > Thank you for pointing out these works. Our approach works in **a different regime than generative modeling**. Prior works use consistency losses to accelerate integrating probability flow ODEs. We instead apply a consistency objective to integrate Hamilton’s equations by learning a deterministic map in phase space.
> > >
> > > This distinction directly affects data requirements. Existing large-timestep methods [1,3] and the suggested LiFlow require trajectories to train, which are costly to generate. For perspective, FlashMD [1] spent 20,000 GPU hours generating trajectories vs 3,000 hours training.
> > >
> > > Both, JAMUN and LiFlow operate in the generative setting. JAMUN learns from trajectory-free equilibrium data, but breaks temporal consistency by introducing noise. We compare to ScoreMD [5], which uses the same diffusion objective but yields consistent dynamics. We show that we outperform it, especially under biased data (Fig. 7).
> > >
> > > We will highlight this distinction in a revision.
> > >
> > > > In what scenarios does the method break?
> > >
> > > Our objective complements MLFFs with a drop-in replacement to enable larger time steps and becomes equivalent to force matching for $\Delta t=0$. As such, we inherit the established robustness and favorable properties of MLFFs, as well as the well-understood failure modes outlined in [6].
> > >
> > > The unique failure modes of our method occur if $\Delta t$ exceeds what the learned map can resolve. This maximum viable timestep is system-dependent, but consistently exceeds classical integration limits (Tbl R2).
> > >
> > > ---
> > >
> > > Thank you again for your thoughtful and constructive feedback, which has helped us to improve the manuscript. We hope the scope, assumptions, and contribution of the work are clearer now, and we could assist in addressing the remaining concerns.
> > >
> > > [4] Bonneau et al "Breaking the Barriers of Molecular Dynamics With Deep-Learning: Opportunities, Pitfalls, and How to Navigate Them" WIREs Comp. Mol. Science 2026
> > >
> > > [5] Plainer et al "Consistent Sampling and Simulation: Molecular Dynamics with Energy-Based Diffusion Models" NeurIPS 2025
> > >
> > > [6] Poltavsky et al "Crash testing machine learning force fields for molecules, materials, and interfaces" Chemical science 2025

---

### Official Review · Reviewer_YX3a · 2026-03-10

**Soundness:** 2
**Presentation:** 3
**Significance:** 2
**Originality:** 3
**Overall Recommendation:** 4
**Confidence:** 4

**Summary:**

This work proposes a trajectory-free framework for learning large-timestep dynamics in Hamiltonian systems, with a particular focus on accelerating molecular dynamics simulations. The paper reframes MD acceleration as learning a large-timestep Hamiltonian flow map via a trajectory-free consistency objective, allowing efficient training on widely available MLFF datasets and enabling stable long-time simulations with much larger integration steps.

**Compliance With Llm Reviewing Policy:**

Affirmed.

**Final Justification:**

My key concerns are the motivation (data efficiency), experiment baselines, and datasets.
The authors addressed these concerns by explanation and additional experiments, which changed my evaluation from weak reject to weak accept.

**Key Questions For Authors:**

1. The method claims that instantaneous derivatives are sufficient to learn the Hamiltonian flow map. However, it is unclear whether learning purely from local information can reliably capture long-term nonlinear dynamics, especially in high-dimensional molecular systems. Could the authors provide a theoretical justification or analysis for long-horizon reliability?

2. The motivation is somewhat unclear. While the method avoids expensive trajectory generation, it requires momentum information, which increases the labeling burden. Could the authors compare the data efficiency with common settings, such as MLFF trained with energy and force labels, MLFF trained with only force labels, and trajectory-based models?

3. In Table 10, the proposed method appears to perform worse than the MLFF baseline not only in long-horizon simulations but also in the single-step setting (presumably 0.5 fs). Could the authors clarify this comparison and explain this behavior? Besides, in most tables, the proposed methods in 0.5 fs are missing. I understand the reason for examining long simulations, but the same setting for direct comparison is also needed.

4. Although the experiments on MD17 are thorough, MD17 is known to suffer from sampling bias and limited scalability. Would the method maintain its performance on more modern benchmarks such as rMD17 or MD22?

5. The motivation seems particularly relevant for large molecular systems where trajectory generation is expensive, yet experiments on larger-scale systems (in terms of atom count) are missing. Could the authors comment on the scalability of the method to larger systems?

6. It seems that the backbone is transformer-based, why not use equiformer or what so like to preserve the equivariance in the model?

7. I am confused about section E.3, is SO(3)-Equivariant Transformer part of the method (not discussed in the main text) or just the baseline (but in the section of Model Architecture)?

**Limitations:**

yes

**Strengths And Weaknesses:**

# Strengths
1. The work targets an important limitation in MD: small stable integration timesteps. Even with MLFFs, integration remains the dominant cost. Learning large-timestep flow maps is, therefore, a meaningful direction for accelerating long-time simulations.
2. The experiments involved simulations in NVE and NVT and analyzed the metastable states.
# Weaknesses
1. Although the authors claim that instantaneous derivatives are sufficient to learn the correct Hamiltonian flow map, it is unclear whether learning from local information alone can reliably capture long-time nonlinear dynamics, especially in high-dimensional molecular systems.
2. In my opinion, the motivation is quite questionable; the method is motivated by expensive trajectory generation; however, the proposed method relies on the momentum, which indeed increases the labeling burden. The author should compare the data efficiency with these training paradigms: MLFF with energy and force labels, MLFF with only force labels, and the trajectory models.
3. In Table 10, the performance of the proposed method not only performs worse on long-term simulations, but also in 1-timestep simulations compared to MLFF baselines (0.5 fs). (Table 10 is not complete, but I guess the baseline is 0.5 fs).
4. I appreciate the detailed experiments on MD17, but MD17 is somehow problematic due to the sampling bias and scalability; rMD17 or MD22 might be more suitable.
5. As MLFF and trajectory generative models develop, ML-based small-scale MD simulations have been deeply researched. I suppose the expensive trajectory data on large-scale datasets should be rather critical; however, the experiments on large-scale datasets in terms of the number of atoms are missing.

---

> ### Author Rebuttal · Authors · 2026-03-30
>
> We thank the reviewer for their thoughtful comments and address all concerns below
>
> > 1. The method claims that instantaneous derivatives are sufficient to learn the HFM. [...] it is unclear whether learning purely from local information can reliably capture long-term nonlinear dynamics [...] theoretical justification or analysis for long-horizon reliability?
>
> This is the central study of our work and is not unique to our method: standard MLFFs also rely on local-only information, with long-time dynamics emerging through the integrator. Our method follows the same principle but learns finite-time updates.
> Crucially, our objective is **not purely local** with Mean Flow consistency providing a global-in-time constraint. Empirically, we validate long-horizon reliability across dynamical and statistical observables (Figs. 6-8, Tbl. 1, App.G), closely matching reference simulations.
>
> > 2a. While the method avoids expensive trajectory generation, it requires momentum information, which increases the labeling burden.
>
> **Our method does not increase labeling cost**: momenta are sampled cheaply from the Maxwell-Boltzmann distribution (sec 4.2 and B.1) which allows direct training on widely-available MLFF datasets. This differentiates us from other approaches. E.g., FlashMD [1] spent 20,000 H200 GPU hours to generate trajectories and only 3,000 H100 hours for training.
>
> > 2b. Could the authors compare the data efficiency with common settings, such as MLFF trained with energy and force labels, MLFF trained with only force labels, and trajectory-based models?
>
> Our method yields similar data efficiency as MLFFs trained on energies and forces (Fig 7). Training MLFFs with only forces works similarly well [2]. We also added **comparisons with trajectory-based objectives** where HFMs are more data-efficient (see responses to Reviewer BFPw Q4 and oQ6j)
>
> > 3a. In Table 10, the proposed method appears to perform worse than the MLFF baseline not only in long-horizon simulations but also in the single-step setting (presumably 0.5fs).
>
> Non-symplectic large-timestep models and non-conservative forcefields are known to violate equipartition of energy in combination with global thermostats [1]. This effect is amplified by the deterministic Nosé-Hoover thermostat used in Table 10 *with the limitation not being specific to our method*. Our approach works well with local and global stochastic thermostats (Langevin, CSVR, Tables 1,9)
>
> > 3b. Missing 0.5fs comparison
>
> We agree and will include it. In general, we find that the $h(r)$ MAE of the HFM at 0.5fs is similar, but marginally lower than $h(r)$ MAE of the HFM at 1fs (Table R1).
>
> *Table R1: Interatomic distances $h(r)$ MAE [unitless] for HFM with 0.5fs step extending Table 1*
>
> |Aspirin|Ethanol|Naphthalene|Salicylic Acid|
> |---|---|---|---|
> |0.026|0.075|0.044|0.035|
>
> > 4. Would the method maintain its performance on more modern benchmarks such as rMD17 or MD22?
>
> > 5. Could the authors comment on the scalability of the method to larger systems?
>
> Scalability is primarily determined by the underlying architecture rather than our objective. Still, we provide **additional experiments on rMD17** (Table R2) where simulations match the MLFF baseline closely across all timesteps, and demonstrate scalability to **additional larger systems in MD22** (Table R3) and **fast-folding proteins** (BBA and Chignolin). See response to Reviewer BFPw Q2/Q5 and Figure S6,S7,S8 in [supplement](https://drive.google.com/file/d/1snAAlZHVIedi2HRBTooHFoEdPpR2cRAV)
>
> *Table R2: rMD17 $h(r)$ MAE*
>
> |Dataset|MLFF(0.5fs)|HFM(0.5fs)|1fs|3fs|5fs|7fs|9fs|
> |---|---|---|---|---|---|---|---|
> |Aspirin|0.056|0.056|0.060|0.059|0.060|0.060|0.062|
> |Ethanol|0.103|0.105|0.104|0.105|0.107|0.106|0.104|
> |Naphthalene|0.143|0.145|0.144|0.142|0.141|0.143|0.141|
> |SalicylicAcid|0.083|0.085|0.085|0.083|0.083|0.084|0.085|
>
> *Table R3: Results on Ac-Ala3-NHMe. Comparison of HFM free energy surfaces with an MLFF with PMF (Potential of Mean Force) error and MJS (Mean Jensen-Shannon Divergence)*
>
> |Timestep|PMF↓|MJS↓|
> |---|---|---|
> |7fs|0.184|0.0159|
> |9fs|0.234|0.0231|
>
> > 6. It seems that the backbone is transformer-based, why not use equiformer or what so like to preserve the equivariance in the model?
>
> Our method is **architecture agnostic** and compatible with equivariant models. Non-equivariant transformers have been shown to perform competitively [3], and hence we chose the backbone for efficiency.
>
> > 7. I am confused about section E.3, is SO(3)-Equivariant Transformer part of the method or just the baseline?
>
> The SO(3)-Equivariant Transformer is used for the MLFF baseline, not for HFM. We will clarify this in the final manuscript.
>
> ---
> [1] Bigi et al. "FlashMD: long-stride, universal prediction of molecular dynamics" NeurIPS 2025
>
> [2] Schütt et al. "Equivariant message passing for the prediction of tensorial properties and molecular spectra" ICML 2021
>
> [3] Langer et al. "Probing the effects of broken symmetries in machine learning" MLST 2024

---

> > ### Author Rebuttal · Reviewer_YX3a · 2026-04-03
> >
> > I appreciate the author's response, but I still have some concerns:
> > # Q2
> > IMHO, I do not think 'not increase labeling cost' equals 'cheap'. The author should still rigorously discuss and analyze the data efficiency, as this is the key contribution.
> >
> > # Q3a
> > The authors attribute the poor performance in Table 10 to the deterministic Nosé-Hoover thermostat. However, in my opinion, this may be a contradiction in robustness. If the method is intended as a general-purpose integrator, its sensitivity to the choice of thermostat is a significant practical weakness. The authors should include results using the stochastic thermostats (Langevin or CSVR) within Table 10 to prove that the performance drop is indeed due to the thermostat and not an inherent flaw in the HFM's ability to maintain the NVT ensemble at 0.5 fs.

---

> > > ### Author Response · Authors · 2026-04-04
> > >
> > > We would like to thank you again for engaging with our work. We appreciate your recognition that addressing the small timestep limitation in MD is meaningful, and your positive assessment of the experiments across ensembles. We fully agree that acceleration of long-time simulations must be both reliable and data-efficient and address your remaining concerns below.
> > >
> > > > I do not think 'not increase labeling cost' equals 'cheap'. The author should still rigorously discuss and analyze the data efficiency, as this is the key contribution.
> > >
> > > We thank the reviewer for this comment and would like to clarify, that the Maxwell-Boltzmann distribution is **a standard per-atom Gaussian** with the mass of the atom $m_i$, target temperature $T$, and Boltzmann constant $k_B$. We hence sample momenta from $\boldsymbol{p}_i \sim \mathcal{N}(0, m_i k_B T)$, which introduces negligible computational overhead during training.
> > >
> > > We agree that “not increasing labeling cost” does not imply data efficiency, which is a key factor, especially compared to existing large-timestep methods that train on trajectories. To directly address this, we extend our evaluation with controlled comparisons under a fixed budget of ground-truth force evaluations following [1]. For each budget, we train either (i) our trajectory-free consistency objective or (ii) trajectory-based models on generated NVE trajectories (Table R4).
> > >
> > > Across all budgets, we consistently observe that HFM achieves lower errors for the same force labeling costs, indicating higher data efficiency compared to trajectory-based training. This is particularly pronounced in the low-data regime. Full results (including additional results on MD22) are provided in Fig. S1,S3 and Tab. S1,S2 in our [supplement](https://drive.google.com/file/d/1snAAlZHVIedi2HRBTooHFoEdPpR2cRAV).
> > >
> > > *Table R4. Comparison of HFM vs. trajectory matching [1] under the same budget of force evaluations for generating the training data. We state PMF and MJS errors over dihedral angles for Paracetamol compared to a MLFF reference.*
> > > |# Force Evaluations|HFM PMF↓|HFM MJS↓|Traj PMF↓|Traj MJS↓|
> > > |---|---|---|---|---|
> > > |128|0.136|0.0132|⚡|⚡|
> > > |256|0.129|0.0121|5.772|0.2696|
> > > |1024|0.127|0.0120|0.326|0.0312|
> > > |85k|0.124|0.0119|0.122|0.0115|
> > >
> > > We will include the data-efficiency comparison, as well as a clarification of momenta sampling in the main text, and update Algorithm 1 to explicitly include the Gaussian sampling step.
> > > > Robustness wrt. choice of thermostat in NVT ensemble
> > >
> > > We agree that robustness across thermostats is an important consideration.
> > >
> > > As discussed in Appendix G.4, the performance degradation observed with the Nosé-Hoover thermostat is **not an inherent flaw** in our method’s ability to maintain the NVT ensemble but a **known artifact** of combining approximate, non-symplectic maps with global deterministic thermostats. This combination is known to cause failures in maintaining kinetic energy equipartition, an effect similarly observed in non-conservative MLFFs and other large-timestep models [1,4].
> > >
> > > We intentionally reported Table 10 to highlight this case transparently and believe that it will **increase practical applicability**, since it provides clear guidance on thermostat choice.
> > >
> > > Our method remains **highly robust** when paired with the **standard Langevin thermostat** used throughout the main text (Table 1) or **a global stochastic CSVR thermostat** (Table 9), neither of which exhibits the artifacts seen in Table 10. Our results in tables 1, 9 and 10 differ only in the choice of thermostat enabling a fair comparison under identical settings. We followed your suggestion to combine all results in a single table (see Tab. S1 in our [supplement 2](https://drive.google.com/file/d/1dF1I_34JakF7l0pN0s0cYkvB3D9PDuiM)).
> > >
> > > To directly address your concern regarding performance at 0.5 fs steps, we provide additional results for Nosé-Hoover thermostat at 0.5fs and compare it to the standard setting using the Langevin thermostat (Table R1) that we use throughout the main text. We will include a full table with all combinations of time steps and thermostats in the final manuscript.
> > >
> > > *Table R5: Interatomic distances $h(r)$ MAE [unitless] for HFM with 0.5fs step extending Tables 1 and 10*
> > > Thermostat|Aspirin|Ethanol|Naphthalene|Salicylic Acid|
> > > |---|---|---|---|---|
> > > |Langevin|0.026|0.075|0.044|0.035|
> > > |Nosé-Hoover|0.236|0.716|0.085|0.159|
> > >
> > >
> > > As demonstrated in all our experiments, our method remains a robust and highly efficient general-purpose integrator when paired with standard stochastic thermostats.
> > >
> > > ---
> > >
> > > Thank you again for your thoughtful and constructive feedback, which has helped us to improve the manuscript. We hope the scope, assumptions, and contribution of the work are clearer now, and we could assist in addressing the remaining concerns.
> > >
> > > [4] Bigi et al. "The dark side of the forces: assessing non-conservative force models for atomistic machine learning" ICML 2025

---

### Official Review · Reviewer_oQ6j · 2026-03-12

**Soundness:** 4
**Presentation:** 3
**Significance:** 2
**Originality:** 2
**Overall Recommendation:** 5
**Confidence:** 4

**Summary:**

This work proposes to learn the solution map to Hamiltonian dynamics by using a physics-informed neural network (PINN)-style loss that enforces the PDE satisfied by the solution map. The goal is to allow for larger time steps during molecular dynamics. They test on a few classical mechanics systems as well as 5 small benchmark molecules for molecular dynamics.

**Compliance With Llm Reviewing Policy:**

Affirmed.

**Final Justification:**

The authors clarified the data efficiency arguments, in particular around prior methods requiring trajectories. Existing datasets do not often provide accurate MLFF trajectories, so generation could still be expensive for a large dataset, even if we only need 10 time steps per sample. The authors also clarified the relationship with PINNs.

While the speed-up provided by this method is limited by chaos, as acknowledged by the authors, one order of magnitude is still a useful, though perhaps not groundbreaking advance. Ideally, we would need several orders of magnitude to reach useful time scales on many interesting problems (which is likely not possible deterministically). Whether it is worth giving up conservative forces for a 2-10x speedup is a question that should be further explored, especially if this method will be used in applications.

I have raised my score to reflect these considerations.

**Key Questions For Authors:**

1. How impactful is this approach when you are limited by the chaotic dynamics of the system?
2. How does this approach compare against other works for learning larger time steps?
3. Is simply correcting the conservation at each step sufficient or do you lose something by dropping explicit force fields? Have you tested this on more chaotic trajectories than shown in the appendix?

**Limitations:**

Generally, yes, but see above.

**Strengths And Weaknesses:**

**Soundness**
The paper appears to be technically sound. However, the authors perhaps overstate the usefulness of their approach to address various limitations of standard MD.

**Presentation**
The paper is generally well-written. Regarding prior work, the authors primarily focus on prior work in consistency models but do not cite relevant work on solving PDEs using PINNs and relevant theory, such as Koopman operator methods.

**Significance**
Accelerating MD simulations is certainly an important problem. It is complicated by the chaotic nature of MD over large time scales. As acknowledged by the authors, learning the solution operator for long times becomes increasingly difficult due to chaos, so taking larger steps in a deterministic fashion can only take you so far due to this theoretical limit. They appear to run into this barrier when increasing the size of the time step, and so are only able to achieve performance roughly 2-4x what is possible with standard MD. The proposed approach may be more useful in settings where the dynamics are less chaotic. This is why such approaches are useful for consistency models, for example, because those trajectories are designed to straight and easy to integrate.

Importantly, the authors cite but do not compare against any other works designed to learn larger time steps from trajectories. They argue that generating such trajectories for training the solution operator directly is prohibitively expensive. However, given that they only achieve step sizes ~10x the reference MD step size, this does not really hold up. It would be relatively straightforward to produce short MD trajectories of 10 steps at scale and in parallel. The authors should really be comparing against these methods and understanding the trade-offs. I would expect that directly learning from the trajectory would be easier to train versus the physics-informed loss proposed by the authors.

**Originality**
Regarding the claimed contributions:
1. Objective: The training objective is a standard PINN loss for the well-known PDE from dynamical systems that describes the evolution of the solution operator. The application in this domain may be novel.
2. Dataset: As addressed above, it is certainly possible to produce datasets of short MD trajectories and compare against other methods that were not tested. You could also bootstrap by using MLFF to produce such trajectories if needed.
3. Standard stability limits: Again, this is not necessarily novel when compared against other approaches for learning larger time steps.

---

> ### Author Rebuttal · Authors · 2026-03-30
>
> We thank the reviewer for the detailed feedback and address all concerns below.
>
> > Relation to PINNs
>
>
> We agree that our objective (Eq. 10) resembles a PINN loss minimizing the Liouville equation residual. However, since we derive this formulation through continuous-time generative consistency models, our framework differs from standard PINNs:
>
> 1. Our chosen consistency condition takes the form of a PDE residual but is only one formulation of flow map objectives [1]. We could naturally employ a semi-group consistency condition, not resembling PINNs.
>
> 2. PINNs learn a specific solution for fixed boundary and initial conditions. In contrast, our network learns an operator to advance *any* phase-space state by a variable $\Delta t$.
>
> 3. PINNs penalize a PDE residual against a fixed target of zero, whereas our network must bootstrap its own targets (appearing on both sides of the equation). This self-distillation is absent in PINN or Koopman literature.
>
> 4. Related PINN/Koopman approaches often require sequential data [2], whereas we use only instantaneous labels.
>
> We thank the reviewer for highlighting the connections to PINNs and we will make the relation clear in the revised manuscript.
>
> > Significance of our work and impact under chaotic dynamics
>
>
> Our work is particularly impactful in computationally expensive QM-accurate MD, where accessible timescales remain limited even with MLFFs. Our method provides an additional order-of-magnitude speedup without requiring new data.
> We further confirm robustness on **larger and highly chaotic systems**, including CG proteins (MD22, Chignolin, BBA), maintaining correct ensemble statistics at large timesteps. Notably, Chignolin in table R1 remains accurate at a stepsize >20x larger than normal MLFFs. See figures S1,S6,S7 and tables S1,S7,S8 in [supplement](https://drive.google.com/file/d/1snAAlZHVIedi2HRBTooHFoEdPpR2cRAV) for other systems
>
> *Table R1. PMF (Potential of Mean Force) and MJS (Mean Jensen-Shannon Divergence) errors over TICA components for Chignolin comparing HFM to an MLFF simulation*
>
> |Δt|PMF↓|MJS↓|
> |---|---|---|
> |5fs|0.175|0.0195|
> |15fs|0.166|0.0175|
> |25fs|0.122|0.0131|
> |35fs|0.073|0.0079|
> |45fs|0.314|0.0306|
>
> > Comparison to trajectory-based large-timestep models
>
> Our method trains on available single-point MLFF data. This eliminates teacher-model bias and maximizes information efficiency, as sequential trajectories are heavily correlated. Conversely, generating trajectories is expensive: FlashMD [3] spends 20,000 H200 GPU hours on trajectory generation vs. 3,000 H100 hours on training.
> To quantify our claims over **trajectory-based approaches**, we present **additional baselines** and generate independent NVE trajectories using a reference MLFF with a fixed budget of force evaluations following the setup of [3,4]. For each budget, we train a model using our consistency loss or regression on trajectories. We simulate under the same conditions (Table R2) finding that our method is more data efficient. See figures S1,S3 and tables S1,S2 in [supplement](https://drive.google.com/file/d/1snAAlZHVIedi2HRBTooHFoEdPpR2cRAV) for results on MD22.
>
> *Table R2. Comparison of HFM vs. trajectory matching [3,4] under the same budget of force evaluations for generating training data. We state PMF and MJS errors over dihedral angles for Paracetamol compared to a MLFF reference. In the limited data setting our model outperforms trajectory matching by a large margin*
>
> |# Force Evaluations|HFM PMF↓|HFM MJS↓|Traj PMF↓|Traj MJS↓|
> |---|---|---|---|---|
> |128|0.136|0.0132|⚡|⚡|
> |256|0.129|0.0121|5.772|0.2696|
> |1024|0.127|0.0120|0.326|0.0312|
> |85k|0.124|0.0119|0.122|0.0115|
>
> > do you lose something by dropping explicit (conservative) force fields? Have you tested this on more chaotic trajectories?
>
> Theoretically yes. Dropping this constraint sacrifices exact symplecticity and energy conservation. While symplectic models are preferable, they currently lack the scalability required for complex systems [5].
> We extend the experiments on conservation laws (App.G.1) to the more complex Ac-Ala3-NHMe. We simulate with a stepsize of 7fs for 3ns in NVE and find that total energy fluctuates with std of 0.46 mEV and the norm of total angular momentum with std 0.168 eV fs. This confirms that the empirical stability of our approach holds without explicit constraints. See figure S2 in [supplement](https://drive.google.com/file/d/1snAAlZHVIedi2HRBTooHFoEdPpR2cRAV) for details.
>
> ---
>
> [1] Boffi et al. “How to build a consistency model: Learning flow maps via self-distillation” NeurIPS 2025
>
> [2] Alford-Lago et al. "Deep learning enhanced dynamic mode decomposition" Chaos 2022
>
> [3] Bigi et al. "FlashMD: long-stride, universal prediction of molecular dynamics" NeurIPS 2025
>
> [4] Thiemann et al. "Force-free molecular dynamics through autoregressive equivariant networks" arXiv:2503.23794
>
> [5] Bigi et al. "Learning the action for long-time-step simulations of molecular dynamics" arXiv:2508.01068

---

> > ### Author Rebuttal · Reviewer_oQ6j · 2026-04-04
> >
> > Thank you for the additional clarifications and experiments. The point about efficiency is clearer now. My concerns have been addressed.

---

> > > ### Author Response · Authors · 2026-04-04
> > >
> > > We would like to thank the reviewer again for their careful and thoughtful assessment of our work. We are pleased to read that you found our work “technically sound” and addressing an “important problem”.
> > >
> > > We are happy that we were able to fully address your remaining concerns. In particular, we appreciate the opportunity to further clarify our contributions with respect to data efficiency. We also thank you for pointing us toward a more principled comparison to PINN-based approaches, which we found very insightful. We believe these clarifications and additions will significantly strengthen the final manuscript.
> > >
> > >
> > > We are glad that the reviewer agrees we have resolved all remaining concerns, and we hope this will positively inform their final rating of the submission.

---

### Official Review · Reviewer_WEXz · 2026-03-12

**Soundness:** 4
**Presentation:** 4
**Significance:** 4
**Originality:** 4
**Overall Recommendation:** 6
**Confidence:** 4

**Summary:**

The paper describes a method to train a Flow Map for Hamiltonian systems.

The work shows constructively how to define a loss from the mean-discrepancy field.

By training based on this loss and the istantaneus mean-discrepancy which recovers the force field, the trained model allows to simulate large time steps.

The paper experiments with a simple harmonic system to a larger system as protein.

While previous works, such as FlashMD, have already explored this direction, the paper provides a novel and powerful tool which has very bright potential.

I must admit, beautiful.

**Compliance With Llm Reviewing Policy:**

Affirmed.

**Ethical Review Concerns:**

no ethical concerns

**Final Justification:**

It would find it strange if this work is not accepted. The contribution is very clear and the evaluation shows the soundness of the approach.

**Key Questions For Authors:**

# Questions

## Q1
Why does the error in Figure 5 to not converge to the same error as n=512?
The current explanation is a bit loose.

## Q2
in the Molecular Dynamics section the time step is relative small (6,9,12fs). Why is not possible to increase further?

## Q3
How do you explain the case of Delta t = 10fs for the ADP simulations?

## Q4
It would be nice to understand what is the max time step and why is limited and what limits it.

**Limitations:**

The paper does not really clarify the limitations, but they can be understood from the content. There is a statement that the method does not substitute classical methods For the negative impact to society, there is a dedicated session.

**Strengths And Weaknesses:**

# soundness
The paper is very precise and provides all the necessary information.


# presentation
The presentation is clear and builds very well. All key information are in the main paper and the experiments are well design.

# significance,
The paper has the potential to impact the scientific community. it would be nice why the maximum time step is still relative small.

# originality.

---

> ### Author Rebuttal · Authors · 2026-03-30
>
> We sincerely thank the reviewer for the exceptionally positive feedback and for recognizing the potential of our method. We truly appreciate your careful reading of the manuscript and the highly insightful questions, which we address in detail below.
>
> > Q1. Why does the error in Figure 5 to not converge to the same error as n=512? The current explanation is a bit loose.
>
> In this example, the Velocity Verlet integrator uses the ground truth forces derived from the analytical potential, which was also used to generate the training and test data. As a result, for large $n$, the Velocity Verlet integrator nearly recovers the ground-truth trajectory, and its error is effectively limited only by machine precision. The approximation error of our *learned* HFM will be larger than the machine precision error, and its performance will eventually saturate for a large number of integration steps. For the target regime of fewer integration steps ($n < 256$), our learned HFM outperforms standard Velocity Verlet with exact analytical forces, which are typically unavailable in practice.
>
> > Q2. in the Molecular Dynamics section the time step is relative small (6,9,12fs). Why is not possible to increase further?
>
>
> We are happy to demonstrate that on the fast-folding CG protein Chignolin, our method remains stable and accurate at timesteps more than 20x larger than a standard MLFF integrator (see Table R1 and Figure S6 in our [supplement](https://drive.google.com/file/d/1snAAlZHVIedi2HRBTooHFoEdPpR2cRAV)).
>
> *Table R1. PMF (Potential of Mean Force) and MJS (Mean Jensen-Shannon Divergence) errors for free energies over TICA components for Chignolin comparing HFM to an MLFF reference simulation*
>
> |Δt|PMF↓|MJS↓|
> |---|---|---|
> |5fs|0.175|0.0195|
> |15fs|0.166|0.0175|
> |25fs|0.122|0.0131|
> |35fs|0.073|0.0079|
> |45fs|0.314|0.0306|
>
> We refer to Q4 for a discussion of the factors that ultimately limit the maximum timestep.
>
> > Q3. How do you explain the case of Delta t = 10fs for the ADP simulations?
>
> To analyze this phenomenon, we conducted additional reference simulations using the Amber ff99SB-ILDN force field in OpenMM, which was also used to generate the ground truth data for training. From those simulations, we compute the element-wise spectra to analyze vibrational modes (see Figure S5 in our [supplement](https://drive.google.com/file/d/1snAAlZHVIedi2HRBTooHFoEdPpR2cRAV)).
>
> We find that the fast hydrogen-X vibrations have periods of 10 and 11 fs, which coincide exactly with the integration timesteps where HFMs fail in our experiments. We argue that as we hit the period of the hydrogen vibrations exactly with our integration timestep, it is much harder for the model to make an accurate prediction, since it needs to learn that large intra-step oscillations cancel out exactly for this timestep. Especially if our model makes a relatively small frequency error, i.e., it predicts a slightly larger or smaller time step, the relative error in phase space may accumulate quickly. For larger integration timesteps, this phenomenon vanishes and rollouts become stable again. We thank the reviewer for this insightful question and will update the text accordingly.
>
> > Q4. It would be nice to understand what is the max time step and why is limited and what limits it.
>
> While we show empirically that our HFM enables integration timesteps about one order of magnitude larger than classical integrators, the maximum timestep remains limited by several factors:
> - The chaoticity of the system: Similar to other large-timestep models, our method is still limited by the chaoticity of the system. As we aim to learn a deterministic map in phase space, the optimization target becomes increasingly difficult to learn near the chaos limit, making the training unstable and imposing a hard physical limit on the maximum timestep (see Appendix D for discussion and Appendix G.7 for ablation studies).
> - The non-symplectic nature of large-timestep models: Symplecticity and energy conservation are both necessary and sufficient conditions for correct thermodynamic sampling. Our proposed inference filters can alleviate the energy drift during simulations but cannot enable larger timesteps directly (see Appendix G.10 for discussion). Strictly symplectic models exist but are not yet scalable [1].
> - The stability of our loss function: our loss formulation includes the computation of a Jacobian vector product, which may reduce the training stability [2] in particular for very large time steps.
>
> We believe that all of these points are interesting directions for future work to push integration timesteps even further, and we will make sure to include an extended discussion of those limits in the final version of our manuscript.
>
> ---
>
> [1] Bigi et al. "Learning the action for long-time-step simulations of molecular dynamics" arXiv:2508.01068 (2025)
>
> [2] Boffi et al. “How to build a consistency model: Learning flow maps via self-distillation” NeurIPS 2025

---

> > ### Author Rebuttal · Reviewer_WEXz · 2026-04-03
> >
> > Thank you for the very exceptional contribution and the clarification, yes, adding to the manuscript would be for sure improve the understading of the approach and its limitations.

---

> > > ### Author Response · Authors · 2026-04-04
> > >
> > > We would like to thank the reviewer again for their diligent and thoughtful assessment of our work. We are delighted to read that you found our work an “exceptional contribution”,  “novel and powerful” with “bright potential”.
> > >
> > > We are glad that we were able to fully address your remaining concerns and we sincerely appreciate your in-depth analysis of our experiments, as well as your suggestions regarding a more detailed investigation of the maximum timestep and additional results for the alanine dipeptide experiment. We believe these improvements will significantly strengthen the final manuscript.

---

### Decision · Program_Chairs · 2026-04-30

**Decision:**

Accept (spotlight)

**Comment:**

This paper is recommended for acceptance because it introduces a technically elegant and data efficient framework for accelerating molecular dynamics through 'Hamiltonian Flow Maps.' By replacing the standard, expensive trajectory-matching approach with a trajectory-free consistency loss, the authors enable a 20-fold integration speedup without the need for costly teacher simulations. This methodology enables existing single-point datasets for large-timestep simulations, representing a significant practical and theoretical advance over current models that require thousands of hours of pre-computed trajectory data.

Throughout the review process, the authors were exceptionally responsive, effectively distinguishing their operator-learning approach from traditional physics-informed neural networks and clarifying that their momentum sampling introduces no additional labeling costs. Their deep physical insight was particularly evident in the rebuttal's spectral analysis, which identified specific integration failure modes tied to molecular vibration periods. These additional evaluations on complex systems like the Chignolin protein proved that the method is both scalable and robust, effectively resolving initial reviewer concerns regarding data efficiency and chaotic dynamics.

In conclusion, the work is technically sound, rigorously evaluated, and addresses a major bottleneck in the AI for Science community. While the authors transparently acknowledge the physical limits imposed by system chaoticity, the empirical results across diverse benchmarks like rMD17 and MD22 are compelling. The paper provides a clear, actionable contribution that is highly useful to a broad audience, and I have incorporated the high quality of the authors' rebuttal and the subsequent reviewer consensus into this final decision to accept.